# A graphon-signal analysis of graph neural networks

**Ron Levie**
Faculty of Mathematics
Technion - Israel Institute of Technology
`levieron@technion.ac.il`

## Abstract

We present an approach for analyzing message passing graph neural networks (MPNNs) based on an extension of graphon analysis to a so called graphon-signal analysis. A MPNN is a function that takes a graph and a signal on the graph (a graph-signal) and returns some value. Since the input space of MPNNs is non-Euclidean, i.e., graphs can be of any size and topology, properties such as generalization are less well understood for MPNNs than for Euclidean neural networks. We claim that one important missing ingredient in past work is a meaningful notion of graph-signal similarity measure, that endows the space of inputs to MPNNs with a regular structure. We present such a similarity measure, called the graphon-signal cut distance, which makes the space of all graph-signals a dense subset of a compact metric space – the graphon-signal space. Informally, two deterministic graph-signals are close in cut distance if they "look like" they were sampled from the same random graph-signal model. Hence, our cut distance is a natural notion of graph-signal similarity, which allows comparing any pair of graph-signals of any size and topology. We prove that MPNNs are Lipschitz continuous functions over the graphon-signal metric space. We then give two applications of this result: 1) a generalization bound for MPNNs, and, 2) the stability of MPNNs to subsampling of graph-signals. Our results apply to any regular enough MPNN on any distribution of graph-signals, making the analysis rather universal.

## 1 Introduction

In recent years, the need to accommodate non-regular structures in data science has brought a boom in machine learning methods on graphs. Graph deep learning (GDL) has already made a significant impact on the applied sciences and industry, with ground-breaking achievements in computational biology [2, 10, 17, 28], and a wide adoption as a general-purpose tool in social media, e-commerce, and online marketing platforms, among others. These achievements pose exciting theoretical challenges: can the success of GDL models be grounded in solid mathematical frameworks? Since the input space of a GDL model is non-Euclidean, i.e., graphs can be of any size and any topology, less is known about GDL than standard neural networks. We claim that contemporary theories of GDL are missing an important ingredient: meaningful notions of metric on the input space, namely, graph similarity measures that are defined for *all graphs of any size*, which respect and describe in some sense the behavior of GDL models. In this paper, we aim at providing an analysis of GDL by introducing such appropriate metrics, using *graphon theory*.

A graphon is an extension of the notion of a graph, where the node set is parameterized by a probability space instead of a finite set. Graphons can be seen as limit objects of graphs, as the number of nodes increases to infinity, under an appropriate metric. One result from graphon theory (that reformulates Szemerédi's regularity lemma from discrete mathematics) states that any sufficiently large graph behaves as if it was randomly sampled from a stochastic block model with a fixed number of classes.

37th Conference on Neural Information Processing Systems (NeurIPS 2023).

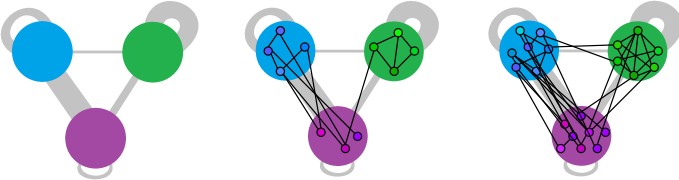

Figure 1: Illustration of the graph-signal cut distance. Left: a stochastic block model (SBM) with a signal. The color of the block represents the value of the signal at this block. The thickness of the edges between the blocks (including self-loops) represents the probability/density of edges between the blocks. Middle: a small graph-signal which looks like was sampled from the SMB. The color of the nodes represents the signal values. Right: a large graph-signal which looks like was sampled from the SMB. In graphon-signal cut distance, these two graph-signals are close to each other.

This result poses an "upper bound" on the complexity of graphs: while deterministic large graphs may appear to be complex and intricate, they are actually approximately regular and behave random-like.

In this paper we extend this regularity result to an appropriate setting for message passing neural networks (MPNNs), a popular GDL model. Since MPNNs take as input a graph with a signal defined over the nodes (a graph-signal), we extend graphon theory from a theory of graphs to a theory of graph-signals. We define a metric, called the *graph-signal cut distance* (see Figure 1 for illustration), and formalize regularity statements for MPNNs of the following sort.

> (1) Any deterministic graph-signal behaves as if it was randomly sampled from a stochastic block model, where the number of blocks only depends on how much we want the graph-signal to look random-like, and not on the graph-signal itself.
> (2) If two graph-signals behave as if they were sampled from the same stochastic block model, then any (regular enough) MPNN attains approximately the same value on both.

Formally, (1) is proven by extending Szemerédi's weak regularity lemma to graphon-signals. As a result of this new version of the regularity lemma, we show that the space of graph-signals is a dense subset of the space of graphon-signals, which is shown to be compact. Point (2) is formalized by proving that MPNNs with Lipschitz continuous message functions are Lipschitz continuous mappings from the space of graph-signals to an output space, in the graphon-signal cut distance.

We argue that the above regularity result is a powerful property of MPNNs. To illustrate this, we use the new regularity result to prove two corollaries. First, a generalization bound of MPNNs, showing that if the learned MPNN performs well on the training graph-signals, it is guaranteed to also perform well on test graph-signals. This is shown by first bounding the covering number of the graphon-signal space, and then using the Lipschitzness of MPNNs. Second, we prove that MPNNs are stable to graph-signal subsampling. This is done by first showing that randomly subsampling a graphon-signal produces a graph-signal which is close in cut distance to the graphon-signal, and then using the Lipschitzness of MPNNs.

As opposed to past works that analyze MPNNs using graphon analysis, we do not assume any generative model on the data. Our results apply to any regular enough MPNN on any distribution of graph-signals, making the analysis rather universal. We note that past works about generalization in GNNs [14, 23, 26, 30] consider special assumptions on the data distribution, and often on the MPNN model. Our work provides upper bounds under no assumptions on the data distribution, and only mild Lipschitz continuity assumptions on the message passing functions. Hence, our theory bounds the generalization error when all special assumptions (that are often simplistic) from other papers are not met. We show that when all assumptions fail, MPNNs still have generalization and sampling guarantees, albeit much slower ones. See Table 1. This is also true for past sampling theorems, e.g., [18, 22, 27, 31, 32].

**The problem with graph-signal domains.** Since the input space of MPNNs is non-Euclidean, results like universal approximation theorems and generalization bounds are less well developed for MPNNs than Euclidean deep learning models. For example, analysis like in [6] is limited to graphs of fixed sizes, seen as adjacency matrices. The graph metric induced by the Euclidean metric on

adjacency matrices is called *edit-distance*. This reduction of the graph problem to the Euclidean case does not describe the full complexity of the problem. Indeed, the edit-distance is defined for weighted graphs, and non-isomorphic simple graphs are always far apart in this metric. This is an unnatural description of the reality of machine learning on graphs, where different large non-isomorphic simple graphs can describe the same large-scale phenomenon and have similar outputs for the same MPNN.

Other papers that consider graphs of arbitrary but bounded size are based on taking the union of the Euclidean edit-distance spaces up to a certain graph size [3]. If one omits the assumption that all graphs are limited by a predefined size, the edit-metric becomes non-compact – a topology too fine to explain the behavior of real MPNNs. For example, two graphs with different number of nodes are always far apart in edit-distance, while most MPNN architectures in practice are not sensitive to the addition of one node to a large graph. In [19], the expressivity of GNNs is analyzed on spaces of graphons. It is assumed that graphons are Lipschitz continuous kernels. The metric on the graphon space is taken as the $L_\infty$ distance between graphons as functions. We claim that the Lipschitz continuity of the graphons in [19], the choice of the $L_\infty$ metric, and the choice of an arbitrary compact subset therein, are not justified as natural models for graphs, and are not grounded in theory. Note that graphon analysis is measure theoretic, and results like the regularity lemma are no longer true when requiring Lipschitz continuity for the graphons. Lastly, in papers like [18, 26, 27, 31], the data is assumed to be generated by one, or a few graphons, which limits the data distribution significantly. We claim that this discrepancy between theory and practice is an artifact of the inappropriate choices of the metric on the space of graphs, and the choice of a limiting generative model for graphs.

## 2 Background

For $n \in \mathbb{N}$, we denote $[n] = \{1, \ldots, n\}$. We denote the Lebesgue $p$ space over the measure space $\mathcal{X}$ by $\mathcal{L}^p(\mathcal{X})$, or, in short, $\mathcal{L}^p$. We denote by $\mu$ the standard Lebesgue measure on $[0, 1]$. A *partition* is a sequence $\mathcal{P}_k = \{P_1, \ldots, P_k\}$ of disjoint measurable subsets of $[0, 1]$ such that $\bigcup_{j=1}^k P_j = [0, 1]$. The partition is called *equipartition* if $\mu(P_i) = \mu(P_j)$ for every $i, j \in [k]$. We denote the indicator function of a set $S$ by $\mathbb{1}_S$. See Appendix A for more details. We summarize our notations in Appendix I.

### 2.1 Message passing neural networks

Most graph neural networks used in practice are special cases of MPNN (see [15] and [11] of a list of methods). MPNNs process graphs with node features, by repeatedly updating the feature at each node using the information from its neighbors. The information is sent between the different nodes along the edges of the graph, and hence, this process is called *message passing*. Each node merges all messages sent from its neighbors using an *aggregation scheme*, where typical choices is to sum, average or to take the coordinate-wise maximum of the messages. In this paper we focus on normalized sum aggregation (see Section 4.1). For more details on MPNNs we refer the reader to Appendix E.

### 2.2 Szemerédi weak regularity lemma

The following is taken from [13, 25]. Let $G = \{V, E\}$ be a simple graph with nodes $V$ and edges $E$. For any two subsets $U, S \subset V$, denote the number of edges with one end point at $U$ and the other at $S$ by $e_G(U, S)$. Let $\mathcal{P} = \{V_1, \ldots, V_k\}$ be a partition of $V$. The partition is called *equipartition* if $||V_i| - |V_j|| \leq 1$ for every $i, j \in [k]$. Given two node set $U, S \subset V$, if the edges between each pair of classes $V_i$ and $V_j$ were random, we would expect the number of edges of $G$ connecting $U$ and $S$ to be close to the expected value $e_{\mathcal{P}(U,S)} := \sum_{i=1}^k \sum_{j=1}^k \frac{e_G(V_i, V_j)}{|V_i||V_j|} |V_i \cap U| |V_j \cap S|$. Hence, the *irregularity*, that measures how non-random like the edges between $\{V_j\}_{j=1}^k$ are, is defined to be

$$\mathrm{irreg}_G(\mathcal{P}) = \max_{U, S \subset V} |e_G(U, S) - e_{\mathcal{P}}(U, S)| / |V|^2. \tag{1}$$

**Theorem 2.1** (Weak Regularity Lemma [13]). *For every $\epsilon > 0$ and every graph $G = (V, E)$, there is an equipartition $\mathcal{P} = \{V_1, \ldots, V_k\}$ of $V$ into $k \leq 2^{c/\epsilon^2}$ classes such that $\mathrm{irreg}_G(\mathcal{P}) \leq \epsilon$. Here, $c$ is a universal constant that does not depend on $G$ and $\epsilon$.*

[Theorem 2.1](#) asserts that we can represent any large graph $G$ by a smaller, coarse-grained version of it: the weighted graph $G^\epsilon$ with node set $V^\epsilon = \{V_1, \ldots, V_k\}$, where the edge weight between the nodes $V_i$ and $V_j$ is $\frac{e_G(V_i, V_j)}{|V_i|, |V_j|}$. The "large-scale" structure of $G$ is given by $G^\epsilon$, and the number of edges between any two subsets of nodes $U_i \subset V_i$ and $U_j \subset V_j$ is close to the "expected value" $e_{\mathcal{P}(U_i, U_j)}$. Hence, the deterministic graph $G$ "behaves" as if it was randomly sampled from $G^\epsilon$.

## 2.3 Graphon analysis

A graphon [4, 24] can be seen as a weighted graph with a "continuous" node set, or more accurately, the nodes are parameterized by an atomless standard probability space called the *graphon domain*. Since all such graphon domains are equivalent to $[0, 1]$ with the standard Lebesgue measure (up to a measure preserving bijection), we take $[0, 1]$ as the node set. The space of graphons $\mathcal{W}_0$ is defined to be the set of all measurable symmetric function $W : [0, 1]^2 \to [0, 1]$, $W(x, y) = W(y, x)$. The edge weight $W(x, y)$ of a graphon $W \in \mathcal{W}_0$ can be seen as the probability of having an edge between the nodes $x$ and $y$.

Graphs can be seen as special graphons. Let $\mathcal{I}_m = \{I_1, \ldots, I_m\}$ be an *interval equipartition*: a partition of $[0, 1]$ into intervals of equal length. The graph $G = \{V, E\}$ with adjacency matrix $A = \{a_{i,j}\}_{i,j=1}^m$ *induces* the graphon $W_G$, defined by $W_G(x, y) = a_{\lceil xm \rceil, \lceil ym \rceil}$ [1]. Note that $W_G$ is piecewise constant on the partition $\mathcal{I}_m$. We hence identify graphs with their induced graphons. A graphon can also be seen as a generative model of graphs. Given a graphon $W$, a corresponding random graph is generated by sampling i.i.d. nodes $\{X_n\}$ from he graphon domain, and connecting each pair $X_n, X_m$ in probability $W(X_n, X_m)$ to obtain the edges of the graph.

## 2.4 Regularity lemma for graphons

A simple way to formulate the regularity lemma in the graphon language is via stochastic block models (SBM). A SBM is a piecewise constant graphon, defined on a partition of the graphon domain $[0, 1]$. The *number of classes* of the SBM is defined to be the number of sets in the partition. A SBM is seen as a generative model for graphs, where graphs are randomly sampled from the graphon underlying the SBM, as explained above. Szemerédi weak regularity lemma asserts that for any error tolerance $\epsilon$, there is a number of classes $k$, such that any deterministic graph (of any size and topology) behaves as if it was randomly sampled from a SBM with $k$ classes, up to error $\epsilon$. Hence, in some sense, every graph is approximately *quasi-random*.

To write the weak regularity lemma in the graphon language, the notion of irregularity ([1](#)) is extended to graphons. For any measurable $W : [0, 1]^2 \to \mathbb{R}$ the *cut norm* is defined to be

$$\|W\|_\square = \sup_{U, S \subset [0,1]} \left| \int_{U \times S} W(x, y) dx dy \right|,$$

where $U, S \subset [0, 1]$ are measurable. It can be verified that the irregularity ([1](#)) is equal to the cut norm of a difference between graphons induced by adequate graphs. The *cut metric* between two graphons $W, V \in \mathcal{W}_0$ is defined to be $d_\square(W, V) = \|W - V\|_\square$. The *cut distance* is defined to be

$$\delta_\square(W, V) = \inf_{\phi \in S_{[0,1]}} \|W - V^\phi\|_\square,$$

where $S_{[0,1]}$ is the space of measure preserving bijections $[0, 1] \to [0, 1]$, and $V^\phi(x, y) = V(\phi(x), \phi(y))$ (see [Section 3.1](#) and [Appendix A.3](#) for more details). The cut distance is a pseudo metric on the space of graphons. By considering equivalence classes of graphons with zero cut distance, we can construct a metric space $\widetilde{\mathcal{W}_0}$ for which $\delta_\square$ is a metric. The following version of the weak regularity lemma is from [25, Lemma 7].

**Theorem 2.2.** *For every graphon $W \in \mathcal{W}_0$ and $\epsilon > 0$ there exists a step graphon $W' \in \mathcal{W}_0$ with respect to a partition of at most $\lceil 2^{c/\epsilon^2} \rceil$ sets such that $\delta_\square(W, W') \leq \epsilon$, for some universal constant c.*

The exact definition of a step graphon is given in [Definition 3.3](#). It is possible to show, using [Theorem 2.2](#), that $\widetilde{\mathcal{W}_0}$ is a compact metric space [25, Lemma 8]. Instead of recalling this construction here, we refer to [Section 3.4](#) for the extension of this construction to graphon-signals.

---

[1]In the definition of $W_G$, the convention is that $\lceil 0 \rceil = 1$.

# 3 Graphon-signal analysis

A graph-signal $(G, \mathbf{f})$ is a graph $G$, that may be weighted or simple, with node set $[n]$, and a signal $\mathbf{f} \in \mathbb{R}^{n \times k}$ that assigns the value $f_j \in \mathbb{R}^k$ for every node $j \in [n]$. A graphon-signal will be defined in Section 3.1 similarly to a graph-signal, but over the node set $[0, 1]$. In this section, we show how to extend classical results in graphon theory to a so called graphon-signal theory. All proofs are given in the appendix.

## 3.1 The graphon signal space

For any $r > 0$, define the *signal space*

$$\mathcal{L}_r^\infty[0, 1] := \big\{ f \in \mathcal{L}^\infty[0, 1] \bigm| \forall x \in [0, 1], \ |f(x)| \le r \big\}. \tag{2}$$

We define the following "norm" on $\mathcal{L}_r^\infty[0, 1]$ (which is not a vector space).

**Definition 3.1** (Cut norm of a signal)**.** *For a signal $f : [0, 1] \to \mathbb{R}$, the* cut norm $\|f\|_\square$ *is defined as*

$$\|f\|_\square := \sup_{S \subseteq [0,1]} \left| \int_S f(x) d\mu(x) \right|, \tag{3}$$

*where the supremum is taken over the measurable subsets $S \subset [0, 1]$.*

In Appendix A.2 we prove basic properties of signal cut norm. One important property is the equivalence of the signal cut norm to the $L_1$ norm

$$\forall f \in \mathcal{L}_r^\infty[0, 1], \quad \|f\|_\square \le \|f\|_1 \le 2\|f\|_\square.$$

Given a bound $r$ on the signals, we define the space of *graphon-signals* to be the set of pairs $\mathcal{WL}_r := \mathcal{W}_0 \times \mathcal{L}_r^\infty[0, 1]$. We define the *graphon-signal cut norm*, for measurable $W, V : [0, 1]^2 \to \mathbb{R}$ and $f, g : [0, 1] \to \mathbb{R}$, by

$$\|(W, f)\|_\square = \|W\|_\square + \|f\|_\square.$$

We define the *graphon-signal cut metric* by $d_\square\big((W, f), (V, g)\big) = \|(W, f) - (V, g)\|_\square$.

We next define a pseudo metric that makes the space of graphon-signals a compact space. Let $S'_{[0,1]}$ be the set of measurable measure preserving bijections between co-null sets of $[0, 1]$, namely,

$$S'_{[0,1]} = \big\{ \phi : A \to B \bigm| A, B \text{ co-null in } [0, 1], \ \text{and} \ \forall S \in A, \ \mu(S) = \mu(\phi(S)) \big\},$$

where $\phi$ is a measurable bijection and $A, B, S$ are measurable. For $\phi \in S'_{[0,1]}$, we define $W^\phi(x, y) := W(\phi(x), \phi(y))$, and $f^\phi(z) = f(\phi(z))$. Note that $W^\phi$ and $f^\phi$ are only defined up to a null-set, and we arbitrarily set $W, W^\phi, f$ and $f^\phi$ to 0 in their respective null-sets, which does not affect our analysis. Define the *cut distance* between two graphon-signals $(W, f), (V, g) \in \mathcal{WL}_r$ by

$$\delta_\square\big((W, f), (V, g)\big) = \inf_{\phi \in S'_{[0,1]}} d_\square\big((W, f), (V, g)^\phi\big). \tag{4}$$

Here, $(V, g)^\phi := (V^\phi, g^\phi)$. More details on this construction are given in Appendix A.3.

The graphon-signal cut distance $\delta_\square$ is a pseudo-metric, and can be made into a metric by introducing the equivalence relation: $(W, f) \sim (V, g)$ if $\delta_\square((W, f), (V, g)) = 0$. The quotient space $\widetilde{\mathcal{WL}}_r := \mathcal{WL}_r / \sim$ of equivalence classes $[(W, f)]$ of graphon-signals $(W, f)$ is a metric space with the metric $\delta_\square([(W, f)], [(V, g)]) = \delta_\square((W, f), (V, g))$. By abuse of terminology, we call elements of $\widetilde{\mathcal{WL}}_r$ also graphon-signals. A graphon-signal in $\widetilde{\mathcal{WL}}_r$ is defined irrespective of a specific "indexing" of the nodes in $[0, 1]$.

## 3.2 Induced graphon-signals

Any graph-signal can be identified with a corresponding graphon-signal as follows.

**Definition 3.2.** *Let $(G, \mathbf{f})$ be a graph-signal with node set $[n]$ and adjacency matrix $A = \{a_{i,j}\}_{i,j \in [n]}$. Let $\{I_k\}_{k=1}^n$ with $I_k = [(k-1)/n, k/n)$ be the equipartition of $[0, 1]$ into $n$ intervals. The graphon-signal $(W, f)_{(G, \mathbf{f})} = (W_G, f_\mathbf{f})$ induced by $(G, \mathbf{f})$ is defined by*

$$W_G(x, y) = \sum_{i,j=1}^n a_{ij} \mathbb{1}_{I_i}(x) \mathbb{1}_{I_j}(y), \quad \text{and} \quad f_\mathbf{f}(z) = \sum_i f_i \mathbb{1}_{I_i}(z).$$

We denote $(W, f)_{(G,\mathbf{f})} = (W_G, f_{\mathbf{f}})$. We identify any graph-signal with its induced graphon-signal. This way, we define the cut distance between a graph-signal and a graphon-signal. As before, the cut distance between a graph-signal $(G, \mathbf{f})$ and a graphon-signal $(W, g)$ can be interpreted as how much the deterministic graph-signal $(G, \mathbf{f})$ "looks like" it was randomly sampled from $(W, g)$.

## 3.3 Graphon-signal regularity lemma

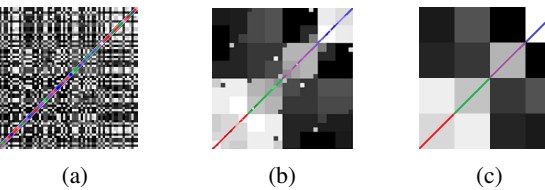

|       |       |       |
|:-----:|:-----:|:-----:|
|  (a)  |  (b)  |  (c)  |

Figure 2: Illustration of the graphon-signal regularity lemma. The values of the graphon are in gray scale over $[0,1]^2$, and the signal is plotted in color on the diagonal of $[0,1]^2$. (a) A graphon-signal. (b) Representation of the same graphon-signal under the "good" permutation/measure preserving bijection guaranteed by the regularity lemma. (c) The approximating step graphon-signal guaranteed by the regularity lemma.

To formulate our regularity lemma, we first define spaces of step functions.

**Definition 3.3.** *Given a partition $\mathcal{P}_k$, and $d \in \mathbb{N}$, we define the space $\mathcal{S}_{\mathcal{P}_k}^d$ of step functions of dimension $d$ over the partition $\mathcal{P}_k$ to be the space of functions $F : [0,1]^d \to \mathbb{R}$ of the form*

$$F(x_1, \ldots, x_d) = \sum_{j=(j_1,\ldots,j_d)\in[k]^d} c_j \prod_{l=1}^{d} \mathbb{1}_{P_{j_l}}(x_l), \tag{5}$$

*for any choice of $\{c_j \in \mathbb{R}\}_{j\in[k]^d}$.*

We call any element of $\mathcal{W}_0 \cap \mathcal{S}_{\mathcal{P}_k}^2$ a *step graphon* with respect to $\mathcal{P}_k$. A step graphon is also called a *stochastic block model (SBM)*. We call any element of $\mathcal{L}_r^\infty[0,1] \cap \mathcal{S}_{\mathcal{P}_k}^1$ a *step signal*. We also call $[\mathcal{WL}_r]_{\mathcal{P}_k} := (\mathcal{W}_0 \cap \mathcal{S}_{\mathcal{P}_k}^2) \times (\mathcal{L}_r^\infty[0,1] \cap \mathcal{S}_{\mathcal{P}_k}^1)$ the space of SBMs with respect to $\mathcal{P}_k$.

In Appendix B.2 we give a number of versions of the graphon-signal regularity lemma. Here, we show one version in which the partition is fixed regardless of the graphon-signal.

**Theorem 3.4** (Regularity lemma for graphon-signals – equipartition)**.** *For any $c > 1$, and any sufficiently small $\epsilon > 0$, for every $n \geq 2^{\lceil \frac{9c}{4\epsilon^2} \rceil}$ and every $(W, f) \in \mathcal{WL}_r$, there exists a step graphon-signal $(W_n, f_n) \in [\mathcal{WL}_r]_{\mathcal{I}_n}$ such that*

$$\delta_\square\big((W, f), (W_n, f_n)\big) \leq \epsilon, \tag{6}$$

*where $\mathcal{I}_n$ is the equipartition of $[0,1]$ into $n$ intervals.*

Figure 2 illustrates the graphon-signal regularity lemma. By identifying graph-signals with their induced graphon-signals, (6) shows that the space of graph-signals is dense in the space of graphon-signals with cut distance.

Similarly to the classical case, Theorem 3.4 is interpreted as follows. While deterministic graph-signals may seem intricate and complex, they are actually regular, and "look like" random graph-signals that were sampled from SBMs, where the number of blocks of the SBM only depends on the desired approximation error between the SBM and the graph-signal, and not on the graph-signal itself.

**Remark 3.5.** *The lower bound $n \geq 2^{\lceil \frac{9c}{4\epsilon^2} \rceil}$ on the number of steps in the graphon-signal regularity lemma is essentially tight in the following sense. There is a universal constant $C$ such that for every $\epsilon > 0$ there exists a graphon-signal $(W, f)$ such that no step graphon-signal $(W', f')$ with less than $2^{\lceil \frac{C}{\epsilon^2} \rceil}$ steps satisfies $\delta_\square\big((W, f), (W', f')\big) \leq \epsilon$. To see this, [8, Theorem 1.4, Theorem 7.1] shows that the bound in the standard weak regularity lemma (for graphs/graphons) is essentially tight in the above sense. For the graphon-signal case, we can take the graphon $W'$ from [8, Theorem 7.1] which does not allow a regularity partition with less than $2^{\lceil \frac{C}{\epsilon^2} \rceil}$ steps, and consider the graphon-signal $(W', 1)$, which then also does not allow such a regularity partition.*

## 3.4 Compactness of the graphon-signal space and its covering number

We prove that $\widetilde{\mathcal{WL}_r}$ is compact using Theorem 3.4, similarly to [25, Lemma 8]. Moreover, we can bound the number of balls of radius $\epsilon$ required to cover $\widetilde{\mathcal{WL}_r}$.

**Theorem 3.6.** *The metric space* $(\widetilde{\mathcal{WL}_r}, \delta_\square)$ *is compact. Moreover, given* $r > 0$ *and* $c > 1$, *for every sufficiently small* $\epsilon > 0$, *the space* $\widetilde{\mathcal{WL}_r}$ *can be covered by*

$$\kappa(\epsilon) = 2^{k^2} \tag{7}$$

*balls of radius* $\epsilon$, *where* $k = \lceil 2^{\frac{9c}{4\epsilon^2}} \rceil$.

The Proof of Theorem 3.6 is given in Appendix C. This is a powerful result – the space of arbitrarily large graph-signals is dense in the "small" space $\widetilde{\mathcal{WL}_r}$. We will use this property in Section 4.3 to prove a generalization bound for MPNNs.

## 3.5 Graphon-signal sampling lemmas

In this section we prove that randomly sampling a graphon signal produces a graph-signal that is close in cut distance to the graphon signal. Let us first describe the sampling setting. More details on the construction are given in Appendix D.1. Let $\Lambda = (\lambda_1, \dots \lambda_k) \in [0,1]^k$ be $k$ independent uniform random samples from $[0,1]$, and $(W, f) \in \mathcal{WL}_r$. We define the *random weighted graph* $W(\Lambda)$ as the weighted graph with $k$ nodes and edge weight $w_{i,j} = W(\lambda_i, \lambda_j)$ between node $i$ and node $j$. We similarly define the *random sampled signal* $f(\Lambda)$ with value $f_i = f(\lambda_i)$ at each node $i$. Note that $W(\Lambda)$ and $f(\Lambda)$ share the sample points $\Lambda$. We then define a random simple graph as follows. We treat each $w_{i,j} = W(\lambda_i, \lambda_j)$ as the parameter of a Bernoulli variable $e_{i,j}$, where $\mathbb{P}(e_{i,j} = 1) = w_{i,j}$ and $\mathbb{P}(e_{i,j} = 0) = 1 - w_{i,j}$. We define the *random simple graph* $\mathbb{G}(W, \Lambda)$ as the simple graph with an edge between each node $i$ and node $j$ if and only if $e_{i,j} = 1$.

We note that, given a graph signal $(G, \mathbf{f})$, sampling a graph-signal from $(W, f)_{(G,\mathbf{f})}$ is equivalent to subsampling the nodes of $G$ independently and uniformly (with repetitions), and considering the resulting subgraph and subsignal. Hence, we can study the more general case of sampling a graphon-signal, where graph-signal sub-sampling is a special case. We now extend [24, Lemma 10.16], which bounds the cut distance between a graphon and its sampled graph, to the case of a sampled graphon-signal.

**Theorem 3.7** (Sampling lemma for graphon-signals)**.** *Let* $r > 1$. *There exists a constant* $K_0 > 0$ *that depends on* $r$, *such that for every* $k \geq K_0$, *every* $(W, f) \in \mathcal{WL}_r$, *and for* $\Lambda = (\lambda_1, \dots \lambda_k) \in [0,1]^k$ *independent uniform random samples from* $[0,1]$, *we have*

$$\mathbb{E}\left(\delta_\square\Big((W, f), \big(W(\Lambda), f(\Lambda)\big)\Big)\right) < \frac{15}{\sqrt{\log(k)}},$$

*and*

$$\mathbb{E}\left(\delta_\square\Big((W, f), \big(\mathbb{G}(W, \Lambda), f(\Lambda)\big)\Big)\right) < \frac{15}{\sqrt{\log(k)}}.$$

The proof of Theorem 3.7 is given in Appendix D.2

# 4 Graphon-signal analysis of MPNNs

In this section, we propose utilizing the compactness of the graphon-signal space under cut distance, and the sampling lemma, to prove regularity results for MPNNs, uniform generalization bounds, and stability to subsampling theorems.

## 4.1 MPNNs on graphon signals

Next, we define MPNNs on graphon-signals, in such a way that the application of a MPNN on an induced graphon-signal is equivalent to applying the MPNN on the graph-signal and then inducing it. A similar construction was presented in [26], for average aggregation, but we use normalized sum aggregation.

At each layer, we define the message function $\Phi(x,y)$ as a linear combination of simple tensors as follows. Let $K \in \mathbb{N}$. For every $k \in [K]$, let $\xi_r^k, \xi_t^k : \mathbb{R}^d \to \mathbb{R}^p$ be Lipschitz continuous functions that we call the *receiver* and *transmitter message functions* respectively. Define the *message function* $\Phi : \mathbb{R}^{2d} \to \mathbb{R}^p$ by

$$\Phi(a,b) = \sum_{k=1}^K \xi_r^k(a)\xi_t^k(b),$$

where the multiplication is elementwise along the feature dimension. Given a signal $f$, define the *message kernel* $\Phi_f : [0,1]^2 \to \mathbb{R}^p$ by

$$\Phi_f(x,y) = \Phi(f(x), f(y)) = \sum_{k=1}^K \xi_r^k(f(x))\xi_t^k(f(y)).$$

We see the $x$ variable of $\Phi_f(x,y)$ as the receiver of the message, and $y$ as the transmitter. Define the aggregation of a message kernel $Q : [0,1]^2 \to \mathbb{R}^p$, with respect to the graphon $W \in \mathcal{W}_0$, to be the signal $\mathrm{Agg}(W,Q) \in \mathcal{L}_r^\infty[0,1]$, defined by

$$\mathrm{Agg}(W,Q)(x) = \int_0^1 W(x,y)Q(x,y)dy,$$

for an appropriate $r > 0$. A *message passing layer (MPL)* takes the form $f^{(t)} \mapsto \mathrm{Agg}(W, \Phi_{f^{(t)}}^{(t+1)})$, where $f^{(t)}$ is the signal at layer $t$. Each MPL is optionally followed by an *update layer*, which updates the signal pointwise via $f^{(t+1)} = \mu^{(t+1)}\big(f^{(t)}(x), \mathrm{Agg}(W, \Phi_{f^{(t)}}^{(t+1)})(x)\big)$, where $\mu^{(t+1)}$ is a learnable mapping called the *update function*. A MPNN is defined by choosing the number of layers $T$, and defining message and update functions $\{\mu^t, (^t\xi_r^k), (^t\xi_t^k)\}_{k \in [K], t \in [T]}$. A MPNN only modifies the signal, and keeps the graph/graphon intact. We denote by $\Theta_t(W,f)$ the output of the MPNN applied on $(W,f) \in \mathcal{WL}_r$ at layer $t \in [T]$. More details on the construction are given in Appendix E.1.

The above construction is rather general. Indeed, it is well known that many classes of functions $F : \mathbb{R}^d \times \mathbb{R}^d \to \mathbb{R}^C$ (e.g., $L^2$ functions) can be approximated by (finite) linear combinations of simple tensors $F(a,b) \approx \sum_{k=1}^K \xi_1^k(a)\xi_2^k(b)$. Hence, message passing based on general message functions $\Phi : \mathbb{R}^{2d} \to \mathbb{R}^p$ can be approximated by our construction. Moreover, many well-known MPNNs can be written using our formulation with a small $K$, e.g., [29, 36] and spectral convolutional networks [9, 20, 21], if we replace the aggregation in these method with normalized sum aggregation.

In Appendix E.1 we show that for any graph-signal $(G, \mathbf{f})$, we have $\Theta_t(W,f)_{(G,\mathbf{f})} = (W,f)_{\Theta_t(G,\mathbf{f})}$, where the MPNN on a graph-signal is defined with normalized sum aggregation

$$\big(\mathrm{Agg}(G, \Phi_{\mathbf{f}})\big)_i = \frac{1}{n} \sum_{j \in [n]} a_{i,j}(\Phi_{\mathbf{f}})_{i,j}.$$

Here, $n$ is the number of nodes, and $\{a_{i,j}\}_{i,j \in [n]}$ is the adjacency matrix of $G$. Hence, we may identify graph-signals with their induced graphon-signals when analyzing MPNNs.

## 4.2 Lipschitz continuity of MPNNs

We now show that, under the above construction, MPNNs are Lipschitz continuous with respect to cut distance.

**Theorem 4.1.** *Let $\Theta$ be a MPNN with $T$ layers. Suppose that there exist constants $L, B > 0$ such that for every layer $t \in [T]$, every $y \in \{t, r\}$ and every $k \in [K]$,*

$$\big|\mu^t(0)\big|, \big|^t\xi_y^k(0)\big| \le B, \quad and \quad L_{\mu^t}, L_{^t\xi_y^k} < L,$$

*where $L_{\mu^t}$ and $L_{^t\xi_y^k}$ are the Lipschitz constants of $\mu^t$ and $^t\xi_y^k$. Then, there exists a constant $L_\Theta$ (that depends on $T, K, B$ and $L$) such that for every $(W,f), (V,g) \in \mathcal{WL}_r$,*

$$\|\Theta(W,f) - \Theta(V,g)\|_\square \le L_\Theta\big(\|f - g\|_\square + \|W - V\|_\square\big).$$

The constant $L_\Theta$ depends exponentially on $T$, and polynomially on $K, B$ and $L$. For formulas of $L_\Theta$, under different assumptions on the hypothesis class of the MPNN, we refer to Appendix F.

## 4.3 A generalization theorem for MPNN

In this section we prove a uniform generalization bound for MPNNs. For background on generalization analysis, we refer the reader to Appendix G.1. While uniform generalization bounds are considered a classical approach in standard neural networks, the approach is less developed in the case of MPNNs. For some works on generalization theorems of MPNNs, see [14, 23, 26, 30, 33].

When a MPNN is used for classification or regression, $\Theta_T$ is followed by global pooling. Namely, for the output signal $g : [0, 1] \to \mathbb{R}^p$, we return $\int g(x)dx \in \mathbb{R}^p$. This is then typically followed by a learnable mapping $\mathbb{R}^p \to \mathbb{R}^C$. In our analysis, we see this mapping as part of the loss, which can hence be learnable. The combined loss is assumed to be Lipschitz continuous[2].

We model the ground truth classifier into $C$ classes as a piecewise constant function $\mathcal{C} : \widetilde{\mathcal{WL}_r} \to \{0, 1\}^C$, where the sets of different steps in $\widetilde{\mathcal{WL}_r}$ are Borel measurable sets, correspond to different classes. We consider an arbitrary probability Borel measure $\nu$ on $\widetilde{\mathcal{WL}_r}$ as the data distribution. More details on the construction are given in Appendix G.2.

Let $\mathrm{Lip}(\widetilde{\mathcal{WL}_r}, L_1)$ be the space of Lipschitz continuous mappings $\Upsilon : \widetilde{\mathcal{WL}_r} \to \mathbb{R}^C$ with Lipschitz constant $L_1$. By Theorem 4.1, we may assume that our hypothesis class of MPNNs is a subset of $\mathrm{Lip}(\widetilde{\mathcal{WL}_r}, L_1)$ for some given $L_1$. Let $\mathbf{X} = (X_1, \ldots, X_N)$ be independent random samples from the data distribution $(\widetilde{\mathcal{WL}_r}, \nu)$. Let $\Upsilon_\mathbf{X}$ be a model that may depend on the sampled data, e.g., via training. Let $\mathcal{E}$ be a Lipschitz continuous loss function[3] with Lipschitz constant $L_2$. For every function $\Upsilon$ in the hypothesis class $\mathrm{Lip}(\widetilde{\mathcal{WL}_r}, L_1)$ (i.e. $\Upsilon_\mathbf{X}$), define the *statistical risk*

$$\mathcal{R}(\Upsilon) = \mathbb{E}\big(\mathcal{E}(\Upsilon, \mathcal{C})\big) = \int \mathcal{E}(\Upsilon(x), \mathcal{C}(x))d\nu(x).$$

We define the empirical risk $\hat{\mathcal{R}}(\Upsilon_\mathbf{X}, \mathbf{X}) = \frac{1}{N} \sum_{i=1}^N \mathbb{E}\big(\Upsilon_\mathbf{X}(X_i), \mathcal{C}(X_i)\big)$.

**Theorem 4.2** (MPNN generalization theorem). *Consider the above classification setting, and let $L = L_1 L_2$. Let $X_1, \ldots, X_N$ be independent random samples from the data distribution $(\widetilde{\mathcal{WL}_r}, \nu)$. Then, for every $p > 0$, there exists an event $\mathcal{U}^p \subset \widetilde{\mathcal{WL}_r}^N$, with probability*

$$\nu^N(\mathcal{U}^p) \geq 1 - Cp - 2\frac{C^2}{N},$$

*in which*

$$\left| \mathcal{R}(\Upsilon_\mathbf{X}) - \hat{\mathcal{R}}(\Upsilon_\mathbf{X}, \mathbf{X}) \right| \leq \xi^{-1}(N/2C)\Big(2L + \frac{1}{\sqrt{2}}\big(L + \mathcal{E}(0, 0)\big)\big(1 + \sqrt{\log(2/p)}\big)\Big), \quad (8)$$

*where $\xi(\epsilon) = \frac{\kappa(\epsilon)^2 \log(\kappa(\epsilon))}{\epsilon^2}$, $\kappa$ is the covering number of $\widetilde{\mathcal{WL}_r}$ given in (7), and $\xi^{-1}$ is the inverse function of $\xi$.*

The theorem is proved in Appendix G.4. Note that the term $\xi^{-1}(N/2C)$ in (8) decreases to zero as the size of the training set $N$ goes to infinity.

In Table 1 we compare the assumptions and dependency on the number of data points of different generalization theorems. All past works consider special assumptions. Our work provides upper bounds under no assumptions on the data distribution, and only mild assumptions on the MPNN (Lipschitz continuity of the message passing and update functions). In Table 2 in Appendix G.5 we present experiments that illustrate the generalization capabilities of MPNNs with normalized sum aggregation.

## 4.4 Stability of MPNNs to graph-signal subsampling

When working with very large graphs, it is often the practice to subsample the large graph, and apply a MPNN to the smaller subsampled graph [5, 7, 16]. Here, we show that such an approach is justified

---

[2] We note that loss functions like cross-entropy are not Lipschitz continuous. However, the composition of cross-entropy on softmax is Lipschitz, which is the standard way of using cross-entropy.

[3] The loss $\mathcal{E}$ may have a learnable component (that depends on the dataset $\mathbf{X}$), as long as the total Lipschitz bound of $\mathcal{E}$ is $L_2$.

Table 1: Comparison of the assumptions made by different GNN generalization analysis papers.

| Generalization analysis paper | Assumption on the graphs | No weight sharing | General MPL | Dependency on $N$ |
|---|---|---|---|---|
| Generalization Limits of GNNs [14] | bounded degree | ✗ | ✗ | $N^{-1/2}$ |
| PAC-bayesian MPNN [23] | bounded degree | ✗ | ✗ | $N^{-1/2}$ |
| PAC-bayesian GCN [23] | bounded degree | ✓ | ✗ | $N^{-1/2}$ |
| VC meets 1WL [30] | bounded color complexity | ✓ | ✗ | $N^{-1/2}$ |
| Generalization Analysis of MPNNs [26] | sampled from a small set of graphons | ✓ | ✓ | $N^{-1/2}$ |
| **Our graphon-signal theory** | **non** | ✓ | ✓ | $\xi^{-1}(N)$ |

theoretically. Namely, any (Lipschitz continuous) MPNN has approximately the same outcome on the large graph and its subsampled version.

Transferability and stability analysis [18, 22, 27, 31, 32] often studies a related setting. Namely, it is shown that a MPNN applied on a randomly sampled graph $G$ approximates the MPNN on the graphon $W$ from which the graph is sampled. However, previous analyses assumed that the generating graphon $W$ has metric properties. Namely, it is assumed that there is some probability metric space $\mathcal{M}$ which is the graphon domain, and the graphon $W : \mathcal{M} \times \mathcal{M} \to [0, 1]$ is Lipschitz continuous with respect to $\mathcal{M}$, where the dimension of $\mathcal{M}$ affects the asymptotics. This is an unnatural setting, as general graphons are only assumed to be measurable, not continuous. Constraining the construction to Lipschitz continuous graphons with a uniformly bounded Lipschitz constant only accounts for a small subset of $\mathcal{WL}_r$, and, hence, limits the analysis significantly. In comparison, our analysis applies to any graphon-signal in $\mathcal{WL}_r$. When we only assume that the graphon is measurable, $[0, 1]$ is only treated as a standard (atomless) probability space, which is very general, and equivalent for example to $[0, 1]^d$ for any $d \in \mathbb{N}$, and to any Polish space. Note that graphon theory allows restricting the graphon domain to $[0, 1]$ since $[0, 1]$, as a measure space, is very generic.

**Theorem 4.3.** *Consider the setting of Theorem 4.2, and let $\Theta$ be a MPNN with Lipschitz constant $L$. Denote*

$$\Sigma = \big( W, \Theta(W, f) \big), \quad and \quad \Sigma(\Lambda) = \Big( \mathbb{G}(W, \Lambda), \Theta\big(\mathbb{G}(W, \Lambda), f(\Lambda)\big) \Big).$$

*Then*

$$\mathbb{E}\Big( \delta_\square\big( \Sigma, \Sigma(\Lambda) \big) \Big) < \frac{15}{\sqrt{\log(k)}} L.$$

## 5 Discussion

We presented an extension of graphon theory to a graphon-signal theory. Especially, we extended well-known regularity, compactness, and sampling lemmas from graphons to graphon-signals. We then showed that the normalized sum aggregation of MPNNs is in some sense compatible with the graphon-signal cut distance, which leads to the Lipschitz continuity of MPNNs with respect to cut distance. This then allowed us to derive generalization and sampling theorems for MPNNs. The strength of our analysis is in its generality and simplicity– it is based on a natural notion of graph similarity, that allows studying the space of *all* graph-signals, it applies to any graph-signal data distribution, and does not impose any restriction on the number of parameters of the MPNNs, only to their regularity through the Lipschitzness of the message functions.

The main limitation of the theory is the very slow asymptotics of the generalization and subsampling theorems. This follows the fact that the upper bound on the covering number of the compact space $\widetilde{\mathcal{WL}_r}$ grows faster than the covering number of any finite-dimensional compact space. Yet, we believe that our work can serve as a point of departure for future works, that 1) will model subspaces of $\widetilde{\mathcal{WL}_r}$ of lower complexity, which approximate the support of the data-distribution in real-life settings of graph machine learning, and, 2) will lead to improved asymptotics. Another open problem is to find an essentially tight estimate of the covering number of $\widetilde{\mathcal{WL}_r}$, which may be lower than the estimate presented in this paper.

## Acknowledgments

We thank Ningyuan (Teresa) Huang for providing the experiments of Table 2.

Ron Levie is partially funded by ISF (Israel Science Foundation) grant #1937/23: Analysis of graph deep learning using graphon theory.

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

# A Basic definitions and properties of graphon-signals

In this appendix, we give basic properties of graphon-signals, cut norm, and cut distance.

## A.1 Lebesgue spaces and signal spaces

For $1 \leq p < \infty$, the space $\mathcal{L}^p[0,1]$ is the space of (equivalence classes up to null-set) of measurable functions $f : [0,1] \to \mathbb{R}$, with finite $L_1$ norm

$$\|f\|_p = \left( \int_0^1 |f(x)|^p dx \right)^{1/p} < \infty.$$

The space $\mathcal{L}^\infty[0,1]$ is the space of (equivalence classes) of measurable functions with finite $L_\infty$ norm

$$\|f\|_\infty = \text{ess} \sup_{x \in [0,1]} |f(x)| = \inf\{a \geq 0 \mid |f(x)| \leq a \text{ for almost every } x \in [0,1]\}.$$

## A.2 Properties of cut norm

Every $f \in \mathcal{L}_r^\infty[0,1]$ can be written as $f = f_+ - f_-$, where

$$f_+(x) = \left\{ \begin{array}{cc} f(x) & f(x) > 0 \\ 0 & f(x) \leq 0. \end{array} \right.$$

and $f_-$ is defined similarly. It is easy to see that the supremum in (3) is attained for $S$ which is either the support of $f_+$ or $f_-$, and

$$\|f\|_\square = \max\{\|f_+\|_1, \|f_-\|_1\}.$$

As a result, the signal cut norm is equivalent to the $L_1$ norm

$$\frac{1}{2}\|f\|_1 \leq \|f\|_\square \leq \|f\|_1. \tag{9}$$

Moreover, for every $r > 0$ and measurable function $W : [0,1]^2 \to [-r,r]$,

$$0 \leq \|W\|_\square \leq \|W\|_1 \leq \|W\|_2 \leq \|W\|_\infty \leq r.$$

The following lemma is from [24, Lemma 8.10].

**Lemma A.1.** *For every measurable $W : [0,1]^2 \to \mathbb{R}$, the supremum*

$$\sup_{S,T \subset [0,1]} \left| \int_S \int_T W(x,y) dx dy \right|$$

*is attained for some $S, T$.*

## A.3 Properties of cut distance and measure preserving bijections

Recall that we denote the standard Lebesgue measure of $[0,1]$ by $\mu$. Let $S_{[0,1]}$ be the space of measurable bijections $[0,1] \to [0,1]$ with measurable inverse, that are measure preserving, namely, for every measurable $A \subset [0,1]$, $\mu(A) = \mu(\phi(A))$. Recall that $S'_{[0,1]}$ is the space of measurable bijections between co-null sets of $[0,1]$.

For $\phi \in S_{[0,1]}$ or $\phi \in S'_{[0,1]}$, we define $W^\phi(x,y) := W(\phi(x), \phi(y))$. In case $\phi \in S'_{[0,1]}$, $W^\phi$ is only define up to a null-set, and we arbitrarily set $W$ to 0 in this null-set. This does not affect our analysis, as the cut norm is not affected by changes to the values of functions on a null sets. The *cut-metric* between graphons is then defined to be

$$\delta_\square(W, W^\phi) = \inf_{\phi \in S_{[0,1]}} \|W - W^\phi\|_\square$$

$$= \inf_{\phi \in S_{[0,1]}} \sup_{S,T \subseteq [0,1]} \left| \int_{S \times T} \left( W(x,y) - W(\phi(x), \phi(y)) \right) dx dy \right|.$$

**Remark A.2.** *Note that $\delta_\square$ can be defined equivalently with respect to $\phi \in S'_{[0,1]}$. Indeed, By [24, Equation (8.17) and Theorem 8.13], $\delta_\square$ can be defined equivalently with respect to the measure preserving maps that are not necessarily invertible. These include the extensions of mappings from $S'_{[0,1]}$ by defining $\phi(x) = 0$ for every $x$ in the co-null set underlying $\phi$.*

Similarly to the graphon case, the graphon-signal distance $\delta_\square$ is a pseudo-metric. By introducing an equivalence relation $(W, f) \sim (V, g)$ if $\delta_\square((W, f), (V, g)) = 0$, and the quotient space $\widetilde{\mathcal{WL}}_r := \mathcal{WL}_r/\sim$, $\widetilde{\mathcal{WL}}_r$ is a metric space with a metric $\delta_\square$ defined by $\delta_\square([(W, f)], [V, g)]) = d_\square(W, V)$ where $[(W, f)], [(V, g)]$, are the equivalence classes of $(W, f)$ and $(V, g)$ respectively. By abuse of terminology, we call elements of $\widetilde{\mathcal{WL}}_r$ also graphon-signals.

**Remark A.3.** *We note that $\widetilde{\mathcal{WL}}_r \neq \widetilde{\mathcal{W}_0} \times \widetilde{\mathcal{L}_r^\infty[0,1]}$ (for the natural definition of $\widetilde{\mathcal{L}_r^\infty[0,1]}$), since in $\widetilde{\mathcal{WL}}_r$ we require that the measure preserving bijection is shared between the graphon $W$ and the signal $f$. Sharing the measure preserving bijetion between $W$ and $f$ is an important modelling requirement, as $\phi$ is seen as a "re-indexing" of the node set $[0,1]$. When re-indexing a node $x$, both the neighborhood $W(x, \cdot)$ of $x$ and the signal value $f(x)$ at $x$ should change together, otherwise, the graphon and the signal would fall out of alignment.*

We identify graphs with their induced graphons and signal with their induced signals

## B  Graphon-signal regularity lemmas

In this appendix, we prove a number of versions of the graphon-signal regularity lemma, where Theorem 3.4 is one version.

### B.1  Properties of partitions and step functions

Given a partition $\mathcal{P}_k$ and $d \in \mathbb{N}$, the next lemma shows that there is an equipartition $\mathcal{E}_n$ such that the space $\mathcal{S}_{\mathcal{E}_n}^d$ uniformly approximates the space $\mathcal{S}_{\mathcal{P}_k}^d$ in $\mathcal{L}^1[0,1]^d$ norm (see Definition 3.3).

**Lemma B.1** (Equitizing partitions). *Let $\mathcal{P}_k$ be a partition of $[0,1]$ into $k$ sets (generally not of the same measure). Then, for any $n > k$ there exists an equipartition $\mathcal{E}_n$ of $[0,1]$ into $n$ sets such that any function $F \in \mathcal{S}_{\mathcal{P}_k}^d$ can be approximated in $L_1[0,1]^d$ by a function from $F \in \mathcal{S}_{\mathcal{E}_n}^d$ up to small error. Namely, for every $F \in \mathcal{S}_{\mathcal{P}_k}^d$ there exists $F' \in \mathcal{S}_{\mathcal{E}_n}^d$ such that*

$$\|F - F'\|_1 \le d\|F\|_\infty \frac{k}{n}.$$

*Proof.* Let $\mathcal{P}_k = \{P_1, \ldots, P_k\}$ be a partition of $[0,1]$. For each $i$, we divide $P_i$ into subsets $\mathbf{P}_i = \{P_{i,1}, \ldots, P_{i,m_i}\}$ of measure $1/n$ (up to the last set) with a residual, as follows. If $\mu(P_i) < 1/n$, we choose $\mathbf{P}_i = \{P_{i,1} = P_i\}$. Otherwise, we take $P_{i,1}, \ldots, P_{i,m_i-1}$ of measure $1/n$, and $\mu(P_{i,m_i}) \le 1/n$. We call $P_{i,m_i}$ the remainder.

We now define the sequence of sets of measure $1/n$

$$\mathcal{Q} := \{P_{1,1}, \ldots, P_{1,m_1-1}, P_{2,1}, \ldots, P_{2,m_2-1}, \ldots, P_{k,1}, \ldots, P_{k,m_k-1}\}, \tag{10}$$

where, by abuse of notation, for any $i$ such that $m_i = 1$, we set $\{P_{i,1}, \ldots, P_{i,m_i-1}\} = \emptyset$ in the above formula. Note that in general $\cup \mathcal{Q} \neq [0,1]$. We moreover define the union of residuals $\Pi := P_{1,m_1} \cup P_{2,m_2} \cup \cdots \cup P_{k,m_k}$. Note that $\mu(\Pi) = 1 - \mu(\cup \mathcal{Q}) = 1 - k\frac{1}{n} = h/n$, where $k$ is the number of elements in $\mathcal{Q}$, and $h = n - k$. Hence, we can partition $\Pi$ into $h$ parts $\{\Pi_1, \ldots \Pi_h\}$ of measure $1/n$ with no residual. Thus we have obtain the equipartition of $[0,1]$ to $n$ sets of measure $1/n$

$$\mathcal{E}_n := \{P_{1,1}, \ldots, P_{1,m_1-1}, P_{2,1}, \ldots, P_{2,m_2-1}, \ldots, S_{k,1}, \ldots, S_{k,m_k-1}, \Pi_1, \Pi_2, \ldots, \Pi_h\}. \tag{11}$$

For convenience, we also denote $\mathcal{E}_n = \{Z_1, \ldots, Z_n\}$.

Let

$$F(x) = \sum_{j=(j_1,\ldots,j_d)\in[k]^d} c_j \prod_{l=1}^d \mathbb{1}_{P_{j_l}}(x_l) \in \mathcal{S}_{\mathcal{P}_k}^d.$$

We can write $F$ with respect to the equipartition $\mathcal{E}_n$ as

$$F(x) = \sum_{j=(j_1,\ldots,j_d)\in[n]^d;\ \forall l=1,\ldots,d,\ Z_{j_l}\not\subset\Pi} \tilde{c}_j \prod_{l=1}^d \mathbb{1}_{Z_{j_l}}(x_l) \ + \ E(x),$$

for some $\{\tilde{c}_j\}$ with the same values as the values of $\{c_j\}$. Here, $E$ is supported in the set $\Pi^{(d)} \subset [0,1]^d$, defied by

$$\Pi^{(d)} = \left(\Pi \times [0,1]^{d-1}\right) \cup \left([0,1] \times \Pi \times [0,1]^{d-2}\right) \cup \ldots \cup \left([0,1]^{d-1} \times \Pi\right).$$

Consider the step function

$$F'(x) = \sum_{j=(j_1,\ldots,j_d)\in[n]^d;\ \forall l=1,\ldots,d,\ Z_{j_l}\not\subset\Pi} \tilde{c}_j \prod_{l=1}^d \mathbb{1}_{Z_{j_l}}(x_l) \in \mathcal{S}_{\mathcal{E}_n}^d.$$

Since $\mu(\Pi) = k/n$, we have $\mu(\Pi^{(d)}) = dk/n$, and so

$$\|F - F'\|_1 \leq d\|F\|_\infty \frac{k}{n}.$$

$\blacksquare$

**Lemma B.2.** *Let* $\{Q_1, Q_2, \ldots, Q_m\}$ *partition of* $[0,1]$. *Let* $\{I_1, I_2, \ldots, I_m\}$ *be a partition of* $[0,1]$ *into intervals, such that for every* $j \in [m]$, $\mu(Q_j) = \mu(I_j)$. *Then, there exists a measure preserving bijection* $\phi : [0,1] \to [0,1] \in S'_{[0,1]}$ *such that*[4]

$$\phi(Q_j) = I_j$$

*Proof.* By the definition of a standard probability space, the measure space induced by $[0,1]$ on a non-null subset $Q_j \subseteq [0,1]$ is a standard probability space. Moreover, each $Q_j$ is atomless, since $[0,1]$ is atomless. Since there is a measure-preserving bijection (up to null-set) between any two atomless standard probability spaces, we obtain the result. $\blacksquare$

**Lemma B.3.** *Let* $\mathcal{S} = \{S_j \subset [0,1]\}_{j=0}^{m-1}$ *be a collection of measurable sets (that are not disjoint in general), and* $d \in \mathbb{N}$. *Let* $\mathcal{C}_{\mathcal{S}}^d$ *be the space of functions* $F : [0,1]^d \to \mathbb{R}$ *of the form*

$$F(x) = \sum_{j=(j_1,\ldots,j_d)\in[m]^d} c_j \prod_{l=1}^d \mathbb{1}_{S_{j_l}}(x_l),$$

*for some choice of* $\{c_j \in \mathbb{R}\}_{j\in[m]^d}$. *Then, there exists a partition* $\mathcal{P}_k = \{P_1, \ldots, P_k\}$ *into* $k = 2^m$ *sets, that depends only on* $\mathcal{S}$, *such that*

$$\mathcal{C}_{\mathcal{S}}^d \subset \mathcal{S}_{\mathcal{P}_k}^d.$$

*Proof.* The partition $\mathcal{P}_k = \{P_1, \ldots, P_k\}$ is defined as follows. Let

$$\tilde{\mathcal{P}} = \left\{P \subset [0,1] \mid \exists\, x \in [0,1],\ P = \cap\{S_j \in \mathcal{S} \mid x \in S_j\}\right\}.$$

We must have $|\tilde{\mathcal{P}}| \leq 2^m$. Indeed, there are at most $2^m$ different subsets of $\mathcal{S}$ for the intersections. We endow an arbitrarily order to $\tilde{\mathcal{P}}$ and turn it into a sequence. If the size of $\tilde{\mathcal{P}}$ is strictly smaller than $2^m$, we add enough copies of $\{\emptyset\}$ to $\tilde{\mathcal{P}}$ to make the size of the sequence $2^m$, that we denote by $\mathcal{P}_k$, where $k = 2^m$. $\blacksquare$

The following simple lemma is proved similarly to Lemma B.3. We give it without proof.

**Lemma B.4.** *Let* $\mathcal{P}_k = \{P_1, \ldots, P_k\}$, $\mathcal{Q}_m = \{Q_1, \ldots, Q_k\}$ *be two partitions. Then, there exists a partition* $\mathcal{Z}_{km}$ *into* $km$ *sets such that for every* $d$,

$$\mathcal{S}_{\mathcal{P}_k}^d \subset \mathcal{S}_{\mathcal{Z}_{mk}}^d, \quad and \quad \mathcal{S}_{\mathcal{Q}_m}^d \subset \mathcal{S}_{\mathcal{Z}_{mk}}^d.$$

---

[4]Namely, there is a measure preserving bijection $\phi$ between two co-null sets $C_1$ and $C_2$ of $[0,1]$, such that $\phi(Q_j \cap C_1) = I_j \cap C_2$.

## B.2 List of graphon-signal regularity lemmas

The following lemma from [25, Lemma 4.1] is a tool in the proof of the weak regularity lemma.

**Lemma B.5.** *Let $\mathcal{K}_1, \mathcal{K}_2, \ldots$ be arbitrary nonempty subsets (not necessarily subspaces) of a Hilbert space $\mathcal{H}$. Then, for every $\epsilon > 0$ and $v \in \mathcal{H}$ there is $m \leq \lceil 1/\epsilon^2 \rceil$ and $v_i \in \mathcal{K}_i$ and $\gamma_i \in \mathbb{R}$, $i \in [m]$, such that for every $w \in \mathcal{K}_{m+1}$*

$$\left| \left\langle w, v - \left( \sum_{i=1}^m \gamma_i v_i \right) \right\rangle \right| \leq \epsilon \|w\| \|v\|. \tag{12}$$

The following theorem is an extension of the graphon regularity lemma from [25] to the case of graphon-signals. Much of the proof follows the steps of [25].

**Theorem B.6** (Weak regularity lemma for graphon-signals). *Let $\epsilon, \rho > 0$. For every $(W, f) \in \mathcal{WL}_r$ there exists a partition $\mathcal{P}_k$ of $[0, 1]$ into $k = \lceil r/\rho \rceil \left( 2^{2\lceil 1/\epsilon^2 \rceil} \right)$ sets, a step function graphon $W_k \in \mathcal{S}_{\mathcal{P}_k}^2 \cap \mathcal{W}_0$ and a step function signal $f_k \in \mathcal{S}_{\mathcal{P}_k}^1 \cap \mathcal{L}_r^\infty[0, 1]$, such that*

$$\|W - W_k\|_\square \leq \epsilon \quad \text{and} \quad \|f - f_k\|_\square \leq \rho. \tag{13}$$

*Proof.* We first analyze the graphon part. In Lemma B.5, set $\mathcal{H} = \mathcal{L}^2([0, 1]^2)$ and for all $i \in \mathbb{N}$, set

$$\mathcal{K}_i = \mathcal{K} = \left\{ \mathbb{1}_{S \times T} \mid S, T \subset [0, 1] \text{ measurable} \right\}.$$

Then, by Lemma B.5, there exists $m \leq \lceil 1/\epsilon^2 \rceil$ two sequences of sets $\mathcal{S}_m = \{S_i\}_{i=1}^m$, $\mathcal{T}_m = \{T_i\}_{i=1}^m$, a sequence of coefficients $\{\gamma_i \in \mathbb{R}\}_{i=1}^m$, and

$$W_\epsilon' = \sum_{i=1}^m \gamma_i \mathbb{1}_{S_i \times T_i},$$

such that for any $V \in \mathcal{K}$, given by $V(x, y) = \mathbb{1}_S(x) \mathbb{1}_T(y)$, we have

$$\left| \int V(x, y) \big( W(x, y) - W_\epsilon'(x, y) \big) dx dy \right| = \left| \int_S \int_T \big( W(x, y) - W_\epsilon'(x, y) \big) dx dy \right| \tag{14}$$

$$\leq \epsilon \|\mathbb{1}_{S \times T}\| \|W\| \leq \epsilon. \tag{15}$$

We may choose exactly $m = \lceil 1/\epsilon^2 \rceil$ by adding copies of the empty set to $\mathcal{S}_m$ and $\mathcal{T}_m$, if the constant $m$ guaranteed by Lemma B.5 is strictly less than $\lceil 1/\epsilon^2 \rceil$. Let $W_\epsilon(x, y) = (W_\epsilon'(x, y) + W_\epsilon'(y, x))/2$. By the symmetry of $W$, it is easy to see that (15) is also true when replacing $W_\epsilon'$ by $W_\epsilon$. Indeed,

$$\left| \int V(x, y) \big( W(x, y) - W_\epsilon(x, y) \big) dx dy \right|$$

$$\leq 1/2 \left| \int V(x, y) \big( W(x, y) - W_\epsilon'(x, y) \big) dx dy \right| + 1/2 \left| \int V(y, x) \big( W(x, y) - W_\epsilon'(x, y) \big) dx dy \right|$$

$$\leq \epsilon.$$

Consider the concatenation of the two sequences $\mathcal{T}_m, \mathcal{S}_m$ given by $\mathcal{Y}_{2m} = \mathcal{T}_m \cup \mathcal{S}_m$. Note that in the notation of Lemma B.3, $W_\epsilon \in \mathcal{C}_{\mathcal{Y}_{2m}}^2$. Hence, by Lemma B.3, there exists a partition $\mathcal{Q}_n$ into $n = 2^{2m} = 2^{2\lceil \frac{1}{\epsilon^2} \rceil}$ sets, such that $W_\epsilon$ is a step graphon with respect to $\mathcal{Q}_n$.

To analyze the signal part, we partition the range of the signal $[-r, r]$ into $j = \lceil r/\rho \rceil$ intervals $\{J_i\}_{i=1}^j$ of length less or equal to $2\rho$, where the left edge point of each $J_i$ is $-r + (i-1) \frac{\rho}{r}$. Consider the partition of $[0, 1]$ based on the preimages $\mathcal{Y}_j = \{Y_i = f^{-1}(J_i)\}_{i=1}^j$. It is easy to see that for the step signal

$$f_\rho(x) = \sum_{i=1}^j a_i \mathbb{1}_{Y_i}(x),$$

where $a_i$ the midpoint of the interval $Y_i$, we have

$$\|f - f_\rho\|_\square \leq \|f - f_\rho\|_1 \leq \rho.$$

Lastly, by Lemma B.4, there is a partition $\mathcal{P}_k$ of $[0,1]$ into $k = \lceil r/\rho \rceil \left( 2^{2\lceil 1/\epsilon^2 \rceil} \right)$ sets such that $W_\epsilon \in \mathcal{S}_{\mathcal{P}_k}^2$ and $f_\rho \in \mathcal{S}_{\mathcal{P}_k}^1$.

∎

**Corollary B.7** (Weak regularity lemma for graphon-signals – version 2)**.** *Let $r > 0$ and $c > 1$. For every sufficiently small $\epsilon > 0$ (namely, $\epsilon$ that satisfies (17)), and for every $(W, f) \in \mathcal{WL}_r$ there exists a partition $\mathcal{P}_k$ of $[0,1]$ into $k = \left( 2^{\lceil 2c/\epsilon^2 \rceil} \right)$ sets, a step graphon $W_k \in \mathcal{S}_{\mathcal{P}_k}^2 \cap \mathcal{W}_0$ and a step signal $f_k \in \mathcal{S}_{\mathcal{P}_k}^1 \cap \mathcal{L}_r^\infty[0,1]$, such that*

$$d_\square\big((W, f), (W_k, f_k)\big) \le \epsilon.$$

*Proof.* First, evoke Theorem B.6, with errors $\|W - W_k\|_\square \le \nu$ and $\|f - f_k\|_\square \le \rho = \epsilon - \nu$. We now show that there is some $\epsilon_0 > 0$ such that for every $\epsilon < \epsilon_0$, there is a choice of $\nu$ such that the number of sets in the partition, guaranteed by Theorem B.6, satisfies

$$k(\nu) := \lceil r/(\epsilon - \nu) \rceil \left( 2^{2\lceil 1/\nu^2 \rceil} \right) \le 2^{\lceil 2c/\epsilon^2 \rceil}.$$

Denote $c = 1 + t$. In case

$$\nu \ge \sqrt{\frac{2}{2(1 + 0.5t)/\epsilon^2 - 1}}, \tag{16}$$

we have

$$2^{2\lceil 1/\nu^2 \rceil} \le 2^{2(1 + 0.5t)/\epsilon^2}.$$

On the other hand, for

$$\nu \le \epsilon - \frac{r}{2^{t/\epsilon^2} - 1},$$

we have

$$\lceil r/(\epsilon - \nu) \rceil \le 2^{2(0.5t)/\epsilon^2}.$$

The reconcile these two conditions, we restrict to $\epsilon$ such that

$$\epsilon - \frac{r}{2^{t/\epsilon^2} - 1} \ge \sqrt{\frac{2}{2(1 + 0.5t)/\epsilon^2 - 1}}. \tag{17}$$

There exists $\epsilon_0$ that depends on $c$ and $r$ (and hence also on $t$) such that for every $\epsilon < \epsilon_0$ (17) is satisfied. Indeed, for small enough $\epsilon$,

$$\frac{1}{2^{t/\epsilon^2} - 1} = \frac{2^{-t/\epsilon^2}}{1 - 2^{-t/\epsilon^2}} < 2^{-t/\epsilon^2} < \frac{\epsilon}{r}\left(1 - \frac{1}{1 + 0.1t}\right),$$

so

$$\epsilon - \frac{r}{2^{t/\epsilon^2} - 1} > \epsilon(1 + 0.1t).$$

Moreover, for small enough $\epsilon$,

$$\sqrt{\frac{2}{2(1 + 0.5t)/\epsilon^2 - 1}} = \epsilon \sqrt{\frac{1}{(1 + 0.5t) - \epsilon^2}} < \epsilon/(1 + 0.4t).$$

Hence, for every $\epsilon < \epsilon_0$, there is a choice of $\nu$ such that

$$k(\nu) = \lceil r/(\epsilon - \nu) \rceil \left( 2^{2\lceil 1/\nu^2 \rceil} \right) \le 2^{2(0.5t)/\epsilon^2} 2^{2(1 + 0.5t)/\epsilon^2} \le 2^{\lceil 2c/\epsilon^2 \rceil}.$$

Lastly, we add as many copies of $\emptyset$ to $\mathcal{P}_{k(\nu)}$ as needed so that we get a sequence of $k = 2^{\lceil 2c/\epsilon^2 \rceil}$ sets.

∎

**Theorem B.8** (Regularity lemma for graphon-signals – equipartition version). *Let $c > 1$ and $r > 0$. For any sufficiently small $\epsilon > 0$, and every $(W, f) \in \mathcal{WL}_r$ there exists $\phi \in S'_{[0,1]}$, a step function graphon $[W^\phi]_n \in \mathcal{S}^2_{\mathcal{I}_n} \cap \mathcal{W}_0$ and a step signal $[f^\phi]_n \in \mathcal{S}^1_{\mathcal{I}_n} \cap \mathcal{L}^\infty_r[0,1]$, such that*

$$d_\square\Big( (W^\phi, f^\phi), \, \big([W^\phi]_n, [f^\phi]_n\big) \Big) \leq \epsilon, \tag{18}$$

*where $\mathcal{I}_n$ is the equipartition of $[0,1]$ into $n = 2^{\lceil 2c/\epsilon^2 \rceil}$ intervals.*

*Proof.* Let $c = 1 + t > 1$, $\epsilon > 0$ and $0 < \alpha, \beta < 1$. In Corollary B.7, consider the approximation error

$$d_\square\big((W, f), (W_k, f_k)\big) \leq \alpha\epsilon.$$

with a partition $\mathcal{P}_k$ into $k = 2^{\lceil \frac{2(1+t/2)}{(\epsilon\alpha)^2} \rceil}$ sets. We next equatize the partition $\mathcal{P}_k$ up to error $\epsilon\beta$. More accurately, in Lemma B.1, we choose

$$n = \lceil 2^{\frac{2(1+0.5t)}{(\epsilon\alpha)^2}+1}/(\epsilon\beta) \rceil,$$

and note that

$$n \geq 2^{\lceil \frac{2(1+0.5t)}{(\epsilon\alpha)^2} \rceil}\lceil 1/\epsilon\beta \rceil = k\lceil 1/\epsilon\beta \rceil.$$

By Lemma B.1 and by the fact that the cut norm is bounded by $L_1$ norm, there exists an equipartition $\mathcal{E}_n$ into $n$ sets, and step functions $W_n$ and $f_n$ with respect to $\mathcal{E}_n$ such that

$$\|W_k - W_n\|_\square \leq 2\epsilon\beta \quad \text{and} \quad \|f_k - f_n\|_1 \leq r\epsilon\beta.$$

Hence, by the triangle inequality,

$$d_\square\big((W, f), (W_n, f_n)\big) \leq d_\square\big((W, f), (W_k, f_k)\big) + d_\square\big((W_k, f_k), (W_n, f_n)\big) \leq \epsilon(\alpha + (2+r)\beta).$$

In the following, we restrict to choices of $\alpha$ and $\beta$ which satisfy $\alpha + (2 + r)\beta = 1$. Consider the function $n : (0, 1) \to \mathbb{N}$ defined by

$$n(\alpha) := \lceil 2^{\frac{4(1+0.5t)}{(\epsilon\alpha)^2}+1}/(\epsilon\beta) \rceil = \lceil (2+r) \cdot 2^{\frac{9(1+0.5t)}{4(\epsilon\alpha)^2}+1}/(\epsilon(1-\alpha)) \rceil.$$

Using a similar technique as in the proof of Corollary B.7, there is $\epsilon_0 > 0$ that depends on $c$ and $r$ (and hence also on $t$) such that for every $\epsilon < \epsilon_0$, we may choose $\alpha_0$ (that depends on $\epsilon$) which satisfies

$$n(\alpha_0) = \lceil (2+r) \cdot 2^{\frac{2(1+0.5t)}{(\epsilon\alpha_0)^2}+1}/(\epsilon(1-\alpha_0)) \rceil < 2^{\lceil \frac{2c}{\epsilon^2} \rceil}. \tag{19}$$

Moreover, there is a choice $\alpha_1$ which satisfies

$$n(\alpha_1) = \lceil (2+r) \cdot 2^{\frac{2(1+0.5t)}{(\epsilon\alpha_1)^2}+1}/(\epsilon(1-\alpha_1)) \rceil > 2^{\lceil \frac{2c}{\epsilon^2} \rceil}. \tag{20}$$

We note that the function $n : (0,1) \to \mathbb{N}$ satisfies the following intermediate value property. For every $0 < \alpha_1 < \alpha_2 < 1$ and every $m \in \mathbb{N}$ between $n(\alpha_1)$ and $n(\alpha_2)$, there is a point $\alpha \in [\alpha_1, \alpha_2]$ such that $n(\alpha) = m$. This follows the fact that $\alpha \mapsto (2+r) \cdot 2^{\frac{2(1+0.5t)}{(\epsilon\alpha)^2}+1}/(\epsilon(1-\alpha))$ is a continuous function. Hence, by (19) and (20), there is a point $\alpha$ (and $\beta$ such that $\alpha + (2+r)\beta = 1$) such that

$$n(\alpha) = n = \lceil 2^{\frac{2(1+0.5t)}{(\epsilon\alpha)^2}+1}/(\epsilon\beta) \rceil = 2^{\lceil 2c/\epsilon^2 \rceil}.$$

$\blacksquare$

By a slight modification of the above proof, we can replace $n$ with the constant $n = \lceil 2^{\frac{2c}{\epsilon^2}} \rceil$. As a result, we can easily prove that for any $n' \geq 2^{\lceil \frac{2c}{\epsilon^2} \rceil}$ we have the approximation property (18) with $n'$ instead of $n$. This is done by choosing an appropriate $c' > c$ and using Theorem B.8 on $c'$, giving a constant $n' = \lceil 2^{\frac{2c'}{\epsilon^2}} \rceil \geq 2^{\lceil \frac{2c}{\epsilon^2} \rceil} = n$. This leads to the following corollary.

**Corollary B.9** (Regularity lemma for graphon-signals – equipartition version 2). *Let $c > 1$ and $r > 0$. For any sufficiently small $\epsilon > 0$, for every $n \geq 2^{\lceil \frac{2c}{\epsilon^2} \rceil}$ and every $(W, f) \in \mathcal{WL}_r$, there exists $\phi \in S'_{[0,1]}$, a step function graphon $[W^\phi]_n \in \mathcal{S}^2_{\mathcal{I}_n} \cap \mathcal{W}_0$ and a step function signal $[f^\phi]_n \in \mathcal{S}^1_{\mathcal{I}_n} \cap \mathcal{L}^\infty_r[0,1]$, such that*

$$d_\square\Big( (W^\phi, f^\phi), \, \big([W^\phi]_n, [f^\phi]_n\big) \Big) \leq \epsilon,$$

*where $\mathcal{I}_n$ is the equipartition of $[0,1]$ into $n$ intervals.*

Next, we prove that we can use the average of the graphon and the signal in each part for the approximating graphon-signal. For that we define the projection of a graphon signal upon a partition.

**Definition B.10.** *Let $\mathcal{P}_n = \{P_1, \ldots, P_n\}$ be a partition of $[0,1]$, and $(W, f) \in \mathcal{WL}_r$. We define the projection of $(W, f)$ upon $(\mathcal{S}_{\mathcal{P}}^2 \times \mathcal{S}_{\mathcal{P}}^1) \cap \mathcal{WL}_r$ to be the step graphon-signal $(W, f)_{\mathcal{P}_n} = (W_{\mathcal{P}_n}, f_{\mathcal{P}_n})$ that attains the value*

$$W_{\mathcal{P}_n}(x, y) = \int_{P_i \times P_j} W(x, y) dx dy , \quad f_{\mathcal{P}_n}(x) = \int_{P_i} f(x) dx$$

*for every $(x, y) \in P_i \times P_j$.*

At the cost of replacing the error $\epsilon$ by $2\epsilon$, we can replace $W'$ with its projection. This was shown in [1]. Since this paper does not use the exact same setting as us, for completeness, we write a proof of the claim below.

**Corollary B.11** (Regularity lemma for graphon-signals – projection version). *For any $c > 1$, and any sufficiently small $\epsilon > 0$, for every $n \geq 2^{\lceil \frac{8c}{\epsilon^2} \rceil}$ and every $(W, f) \in \mathcal{WL}_r$, there exists $\phi \in S'_{[0,1]}$, such that such that*

$$d_\square\Big( \left(W^\phi, f^\phi\right), \left([W^\phi]_{\mathcal{I}_n}, [f^\phi]_{\mathcal{I}_n}\right) \Big) \leq \epsilon.$$

*where $\mathcal{I}_n$ is the equipartition of $[0, 1]$ into $n$ intervals.*

We first prove a simple lemma.

**Lemma B.12.** *Let $\mathcal{P}_n = \{P_1, \ldots, P_n\}$ be a partition of $[0,1]$, and Let $V, R \in \mathcal{S}_{\mathcal{P}_n}^2 \cap \mathcal{W}_0$. Then, the supremum of*

$$\sup_{S, T \subset [0,1]} \left| \int_S \int_T \big(V(x, y) - R(x, y)\big) dx dy \right| \tag{21}$$

*is attained for $S, T$ of the form*

$$S = \bigcup_{i \in s} P_i , \quad T = \bigcup_{j \in t} P_j,$$

*where $t, s \subset [n]$. Similarly for any two signals $f, g \in \mathcal{S}_{\mathcal{P}_n}^1 \cap \mathcal{L}_r^\infty[0, 1]$, the supremum of*

$$\sup_{S \subset [0,1]} \left| \int_S \big(f(x) - g(x)\big) dx \right| \tag{22}$$

*is attained for $S$ of the form*

$$S = \bigcup_{i \in s} P_i,$$

*where $s \subset [n]$.*

*Proof.* First, by Lemma A.1, the supremum of (21) is attained for some $S, T \subset [0, 1]$. Given the maximizers $S, T$, without loss of generality, suppose that

$$\int_S \int_T \big(V(x, y) - R(x, y)\big) dx dy > 0.$$

we can improve $T$ as follows. Consider the set $t \subset [n]$ such that for every $j \in t$

$$\int_S \int_{T \cap P_j} \big(V(x, y) - R(x, y)\big) dx dy > 0.$$

By increasing the set $T \cap P_j$ to $P_j$, we can only increase the size of the above integral. Indeed,

$$\int_S \int_{P_j} \big(V(x, y) - R(x, y)\big) dx dy = \frac{\mu(P_j)}{\mu(T \cap P_j)} \int_S \int_{T \cap P_j} \big(V(x, y) - R(x, y)\big) dx dy$$

$$\geq \int_S \int_{T \cap P_j} \big(V(x, y) - R(x, y)\big) dx dy.$$

Hence, by increasing $T$ to

$$T' = \bigcup_{\{j \mid T \cap P_j \neq \emptyset\}} P_j,$$

we get

$$\int_S \int_{T'} \big( V(x,y) - R(x,y) \big) dx dy \geq \int_S \int_T \big( V(x,y) - R(x,y) \big) dx dy.$$

We similarly replace each $T \cap P_j$ such that

$$\int_S \int_{T \cap P_j} \big( V(x,y) - R(x,y) \big) dx dy \leq 0$$

by the empty set. We now repeat this process for $S$, which concludes the proof for the graphon part.

For the signal case, let $f = f_+ - f_-$, and suppose without loss of generality that $\|f\|_\square = \|f\|_1$. It is easy to see that the supremum of (22) is attained for the support of $f_+$, which has the required form. ∎

*Proof.* Proof of Corollary B.11 Let $W_n \in \mathcal{S}_{\mathcal{P}_n} \cap \mathcal{W}_0$ be the step graphon guaranteed by Corollary B.9, with error $\epsilon/2$ and measure preserving bijection $\phi \in S'_{[0,1]}$. Without loss of generality, we suppose that $W^\phi = W$. Otherwise, we just denote $W' = W^\phi$ and replace the notation $W$ with $W'$ in the following. By Lemma B.12, the infimum underlying $\|W_{\mathcal{P}_n} - W_n\|_\square$ is attained for for some

$$S = \bigcup_{i \in s} P_i, \quad T = \bigcup_{j \in t} P_j.$$

We now have, by definition of the projected graphon,

$$\|W_n - W_{\mathcal{P}_n}\|_\square = \left| \sum_{i \in s, j \in t} \int_{P_i} \int_{P_j} (W_{\mathcal{P}_n}(x,y) - W_n(x,y)) dx dy \right|$$

$$= \left| \sum_{i \in s, j \in t} \int_{P_i} \int_{P_j} (W(x,y) - W_n(x,y)) dx dy \right|$$

$$= \left| \int_S \int_T (W(x,y) - W_n(x,y)) dx dy \right| = \|W_n - W\|_\square.$$

Hence, by the triangle inequality,

$$\|W - W_{\mathcal{P}_n}\|_\square \leq \|W - W_n\|_\square + \|W_n - W_{\mathcal{P}_n}\|_\square < 2\|W_n - W\|_\square.$$

A similar argument shows

$$\|f - f_{\mathcal{P}_n}\|_\square < 2\|f_n - f\|_\square.$$

Hence,

$$d_\square\Big( (W^\phi, f^\phi), ([W^\phi]_{\mathcal{I}_n}, [f^\phi]_{\mathcal{I}_n}) \Big) \leq 2 d_\square\Big( (W^\phi, f^\phi), ([W^\phi]_n, [f^\phi]_n) \Big) \leq \epsilon.$$

∎

## C   Compactness and covering number of the graphon-signal space

In this appendix we prove Theorem 3.6.

Given a partition $\mathcal{P}_k$, recall that

$$[\mathcal{WL}_r]_{\mathcal{P}_k} := (\mathcal{W}_0 \cap \mathcal{S}^2_{\mathcal{P}_k}) \times (\mathcal{L}^\infty_r[0,1] \cap \mathcal{S}^1_{\mathcal{P}_k})$$

is called the space of SBMs or step graphon-signals with respect to $\mathcal{P}_k$. Recall that $\widetilde{\mathcal{WL}_r}$ is the space of equivalence classes of graphon-signals with zero $\delta_\square$ distance, with the $\delta_\square$ metric (defined on arbitrary representatives). By abuse of terminology, we call elements of $\widetilde{\mathcal{WL}_r}$ also graphon-signals.

**Theorem C.1.** *The metric space $(\widetilde{\mathcal{WL}_r}, \delta_\square)$ is compact.*

The proof is a simple extension of [25, Lemma 8] from the case of graphon to the case of graphon-signal. The proof relies on the notion of martingale. A martingale is a sequence of random variables for which, for each element in the sequence, the conditional expectation of the next value in the sequence is equal to the present value, regardless of all prior values. The Martingale convergence theorem states that for any bounded martingale $\{M_n\}_n$ over the probability pace $X$, the sequence $\{M_n(x)\}_n$ converges for almost every $x \in X$, and the limit function is bounded (see [12, 35]).

*Proof.* [Proof of Theorem C.1] Consider a sequence $\{[(W_n, f_n)]\}_{n \in \mathbb{N}} \subset \widetilde{\mathcal{WL}_r}$, with $(W_n, f_n) \in \mathcal{WL}_r$. For each $k$, consider the equipartition into $m_k$ intervals $\mathcal{I}_{m_k}$, where $m_k = 2^{30\lceil(r^2+1)\rceil k^2}$. By Corollary B.11, there is a measure preserving bijection $\phi_{n,k}$ (up to nullset) such that

$$\|(W_n, f_n)^{\phi_{n,k}} - (W_n, f_n)^{\phi_{n,k}}_{\mathcal{I}_{m_k}}\|_{\square;r} < 1/k,$$

where $(W_n, f_n)^{\phi_{n,k}}_{\mathcal{I}_{m_k}}$ is the projection of $(W_n, f_n)^{\phi_{n,k}}$ upon $\mathcal{I}_{m_k}$ (Definition B.10). For every fixed $k$, each pair of functions $(W_n, f_n)^{\phi_{n,k}}_{\mathcal{I}_{m_k}}$ is defined via $m_k^2 + m_k$ values in $[0, 1]$. Hence, since $[0, 1]^{m_k^2 + m_k}$ is compact, there is a subsequence $\{n_j^k\}_{j \in \mathbb{N}}$, such that all of these values converge. Namely, for each $k$, the sequence

$$\{(W_{n_j^k}, f_{n_j^k})^{\phi_{n_j^k, k}}_{\mathcal{I}_{m_k}}\}_{j=1}^{\infty}$$

converges pointwise to some step graphon-signal $(U_k, g_k)$ in $[\mathcal{WL}_r]_{\mathcal{P}_k}$ as $j \to \infty$. Note that $\mathcal{I}_{m_l}$ is a refinement of $\mathcal{I}_{m_k}$ for every $l > k$. As as a result, by the definition of projection of graphon-signals to partitions, for every $l > k$, the value of $(W_n^{\phi_{n,k}})_{\mathcal{I}_{m_k}}$ at each partition set $I_{m_k}^i \times I_{m_k}^j$ can be obtained by averaging the values of $(W_n^{\phi_{n,l}})_{\mathcal{I}_{m_l}}$ at all partition sets $I_{m_l}^{i'} \times I_{m_l}^{j'}$ that are subsets of $I_{m_k}^i \times I_{m_k}^j$. A similar property applies also to the signal. Moreover, by taking limits, it can be shown that the same property holds also for $(U_k, g_k)$ and $(U_l, g_l)$. We now see $\{(U_k, g_k)\}_{k=1}^{\infty}$ as a sequence of random variables over the standard probability space $[0, 1]^2$. The above discussion shows that $\{(U_k, g_k)\}_{k=1}^{\infty}$ is a bounded martingale. By the martingale convergence theorem, the sequence $\{(U_k, g_k)\}_{k=1}^{\infty}$ converges almost everywhere pointwise to a limit $(U, g)$, which must be in $\mathcal{WL}_r$.

Lastly, we show that there exist increasing sequences $\{k_z \in \mathbb{N}\}_{z=1}^{\infty}$ and $\{t_z = n_{j_z}^{k_z}\}_{z \in \mathbb{N}}$ such that $(W_{t_z}, f_{t_z})^{\phi_{t_z, k_z}}$ converges to $(U, g)$ in cut distance. By the dominant convergence theorem, for each $z \in \mathbb{N}$ there exists a $k_z$ such that

$$\|(U, g) - (U_{k_z}, g_{k_z})\|_1 < \frac{1}{3z}.$$

We choose such an increasing sequence $\{k_z\}_{z \in \mathbb{N}}$ with $k_z > 3z$. Similarly, for ever $z \in \mathbb{N}$, there is a $j_z$ such that, with the notation $t_z = n_{j_z}^{k_z}$,

$$\|(U_{k_z}, g_{k_z}) - (W_{t_z}, f_{t_z})^{\phi_{t_z, k_z}}_{\mathcal{I}_{m_{k_z}}}\|_1 < \frac{1}{3z},$$

and we may choose the sequence $\{t_z\}_{z \in \mathbb{N}}$ increasing. Therefore, by the triangle inequality and by the fact that the $L_1$ norm bounds the cut norm,

$$\begin{aligned}
\delta_{\square}\big((U, g), (W_{t_z}, f_{t_z})\big) &\leq \|(U, g) - (W_{t_z}, f_{t_z})^{\phi_{t_z, k_z}}\|_{\square} \\
&\leq \|(U, g) - (U_{k_z}, g_{k_z})\|_1 + \|(U_{k_z}, g_{k_z}) - (W_{t_z}, f_{t_z})^{\phi_{t_z, k_z}}_{\mathcal{I}_{m_{k_z}}}\|_1 \\
&\quad + \|(W_{t_z}, f_{t_z})^{\phi_{t_z, k_z}}_{\mathcal{I}_{m_{k_z}}} - (W_{t_z}, f_{t_z})^{\phi_{t_z, k_z}}\|_{\square} \\
&\leq \frac{1}{3z} + \frac{1}{3z} + \frac{1}{3z} \leq \frac{1}{z}.
\end{aligned}$$

∎

The next theorem bounds the covering number of $\widetilde{\mathcal{WL}_r}$.

**Theorem C.2.** *Let $r > 0$ and $c > 1$. For every sufficiently small $\epsilon > 0$, the space $\widetilde{\mathcal{WL}_r}$ can be covered by*

$$\kappa(\epsilon) = 2^{k^2} \tag{23}$$

*balls of radius $\epsilon$ in cut distance, where $k = \lceil 2^{2c/\epsilon^2} \rceil$.*

*Proof.* Let $1 < c < c'$ and $0 < \alpha < 1$. Given an error tolerance $\alpha\epsilon > 0$, using Theorem B.8, we take the equipartition $\mathcal{I}_n$ into $n = 2^{\lceil \frac{2c}{\alpha^2 \epsilon^2} \rceil}$ intervals, for which any graphon-signal $(W, f) \in \widetilde{\mathcal{WL}_r}$ can be approximated by some $(W, f)_n$ in $[\widetilde{\mathcal{WL}_r}]_{\mathcal{I}_n}$, up to error $\alpha\epsilon$. Consider the rectangle $\mathcal{R}_{n,r} = [0, 1]^{n^2} \times [-r, r]^n$. We identify each element of $[\widetilde{\mathcal{WL}_r}]_{\mathcal{I}_n}$ with an element of $\mathcal{R}_{n,r}$ using the coefficients of (5). More accurately, the coefficients $c_{i,j}$ of the step graphon are identifies with the first $n^2$ entries of a point in $\mathcal{R}_{n,r}$, and the the coefficients $b_i$ of the step signals are identifies with the last $n$ entries of a point in $\mathcal{R}_{n,r}$. Now, consider the quantized rectangle $\tilde{\mathcal{R}}_{n,r}$, defined as

$$\tilde{\mathcal{R}}_{n,r} = \left((1-\alpha)\epsilon\mathbb{Z}\right)^{n^2+2rn} \cap \mathcal{R}_{n,r}.$$

Note that $\tilde{\mathcal{R}}_n$ consists of

$$M \leq \lceil \frac{1}{(1-\alpha)\epsilon} \rceil^{n^2+2rn} \leq 2^{\left(-\log\left((1-\alpha)\epsilon\right)+1\right)(n^2+2rn)}$$

points. Now, every point $x \in \mathcal{R}_{n,r}$ can be approximated by a quantized version $x_Q \in \tilde{\mathcal{R}}_{n,r}$ up to error in normalized $\ell_1$ norm

$$\|x - x_Q\|_1 := \frac{1}{M} \sum_{j=1}^{M} \left| x^j - x_Q^j \right| \leq (1-\alpha)\epsilon,$$

where we re-index the entries of $x$ and $x_Q$ in a 1D sequence. Let us denote by $(W, f)_Q$ the quantized version of $(W_n, f_n)$, given by the above equivalence mapping between $(W, f)_n$ and $\mathcal{R}_{n,r}$. We hence have

$$\|(W, f) - (W, f)_Q\|_\square \leq \|(W, f) - (W_n, f_n)\|_\square + \|(W_n, f_n) - (W, f)_Q\|_\square \leq \epsilon.$$

We now choose the parameter $\alpha$. Note that for any $c' > c$, there exists $\epsilon_0 > 0$ that depends on $c' - c$, such that for any $\epsilon < \epsilon_0$ there is a choice of $\alpha$ (close to 1) such that

$$M \leq \lceil \frac{1}{(1-\alpha)\epsilon} \rceil^{n^2+2rn} \leq 2^{\left(-\log\left((1-\alpha)\epsilon\right)+1\right)(n^2+2rn)} \leq 2^{k^2}$$

where $k = \lceil 2^{2c'/\epsilon^2} \rceil$. This is shown similarly to the proof of Corollary B.7 and Theorem B.8. We now replace the notation $c' \to c$, which concludes the proof. $\blacksquare$

## D  Graphon-signal sampling lemmas

In this appendix, we prove Theorem 3.7. We denote by $\mathcal{W}_1$ the space of measurable functions $U : [0, 1] \to [-1, 1]$, and call each $U \in \mathcal{W}_1$ a kernel.

### D.1  Formal construction of sampled graph-signals

Let $W \in \mathcal{W}_0$ be a graphon, and $\Lambda' = (\lambda'_1, \ldots \lambda'_k) \in [0, 1]^k$. We denote by $W(\Lambda')$ the adjacency matrix

$$W(\Lambda') = \{W(\lambda'_i, \lambda'_j)\}_{i,j \in [k]}.$$

By abuse of notation, we also treat $W(\Lambda')$ as a weighted graph with $k$ nodes and the adjacency matrix $W(\Lambda')$. We denote by $\Lambda = (\lambda_1, \ldots, \lambda_k) : (\lambda'_1, \ldots \lambda'_k) \mapsto (\lambda'_1, \ldots \lambda'_k)$ the identity random variable in $[0, 1]^k$. We hence call $(\lambda_1, \ldots, \lambda_k)$ random independent samples from $[0, 1]$. We call the random variable $W(\Lambda)$ a *random sampled weighted graph*.

Given $f \in \mathcal{L}_r^\infty[0, 1]$ and $\Lambda' = (\Lambda'_1, \ldots, \Lambda'_k) \in [0, 1]^k$, we denote by $f(\Lambda')$ the discrete signal with $k$ nodes, and value $f(\lambda'_i)$ for each node $i = 1, \ldots, k$. We define the *sampled signal* as the random variable $f(\Lambda)$.

We then define the random sampled simple graph as follows. First, for a deterministic $\Lambda' \in [0, 1]^k$, we define a 2D array of Bernoulli random variables $\{e_{i,j}(\Lambda')\}_{i,j \in [k]}$ where $e_{i,j}(\Lambda') = 1$ in probability

$W(\lambda_i', \lambda_j')$, and zero otherwise, for $i, j \in [k]$. We define the probability space $\{0, 1\}^{k \times k}$ with normalized counting measure, defined for any $S \subset \{0, 1\}^{k \times k}$ by

$$P_{\Lambda'}(S) = \sum_{\mathbf{z} \in S} \prod_{i,j \in [k]} P_{\Lambda';i,j}(z_{i,j}),$$

where

$$P_{\Lambda';i,j}(z_{i,j}) = \left\{ \begin{array}{ll} W(\lambda_i', \lambda_j') & \text{if } z_{i,j} = 1 \\ 1 - W(\lambda_i', \lambda_j') & \text{if } z_{i,j} = 0. \end{array} \right.$$

We denote the identity random variable by $\mathbb{G}(W, \Lambda') : \mathbf{z} \mapsto \mathbf{z}$, and call it a *random simple graph sampled from* $W(\Lambda')$.

Next we also allow to "plug" the random variable $\Lambda$ into $\Lambda'$. For that, we define the joint probability space $\Omega = [0, 1]^k \times \{0, 1\}^{k \times k}$ with the product $\sigma$-algebra of the Lebesgue sets in $[0, 1]^k$ with the power set $\sigma$-algebra of $\{0, 1\}^{k \times k}$, with measure, for any measurable $S \subset \Omega$,

$$\mu(S) = \int_{[0,1]^k} P_{\Lambda'}\big(S(\Lambda')\big) d\Lambda',$$

where

$$S(\Lambda') \subset \{0, 1\}^{k \times k} := \{\mathbf{z} = \{z_{i,j}\}_{i,j \in [k]} \in \{0, 1\}^{k \times k} \mid (\Lambda', \mathbf{z}) \in S\},$$

We call the random variable $\mathbb{G}(W, \Lambda) : \Lambda' \times \mathbf{z} \mapsto \mathbf{z}$ the *random simple graph generated by* $W$. We extend the domains of the random variables $W(\Lambda)$, $f(\Lambda)$ and $\mathbb{G}(W, \Lambda')$ to $\Omega$ trivially (e.g., $f(\Lambda)(\Lambda', \mathbf{z}) = f(\Lambda)(\Lambda')$ and $\mathbb{G}(W, \Lambda')(\Lambda', \mathbf{z}) = \mathbb{G}(W, \Lambda')(\mathbf{z})$), so that all random variables are defined over the same space $\Omega$. Note that the random sampled graphs and the random signal share the same sample points.

Given a kernel $U \in \mathcal{W}_1$, we define the random sampled kernel $U(\Lambda)$ similarly.

Similarly to the above construction, given a weighted graph $H$ with $k$ nodes and edge weights $h_{i,j}$, we define the *simple graph sampled from* $H$ as the random variable simple graph $\mathbb{G}(H)$ with $k$ nodes and independent Bernoulli variables $e_{i,j} \in \{0, 1\}$, with $\mathbb{P}(e_{i,j} = 1) = h_{i,j}$, as the edge weights. The following lemma is taken from [24, Equation (10.9)].

**Lemma D.1.** *Let $H$ be a weighted graph of $k$ nodes. Then*

$$\mathbb{E}\big(d_\square(\mathbb{G}(H), H)\big) \leq \frac{11}{\sqrt{k}}.$$

The following is a simple corollary of Lemma D.1, using the law of total probability.

**Corollary D.2.** *Let $W \in \mathcal{W}_0$ and $k \in \mathbb{N}$. Then*

$$\mathbb{E}\big(d_\square(\mathbb{G}(W, \Lambda), W(\Lambda))\big) \leq \frac{11}{\sqrt{k}}.$$

### D.2 Sampling lemmas of graphon-signals

The following lemma, from [24, Lemma 10.6], shows that the cut norm of a kernel is approximated by the cut norm of its sample.

**Lemma D.3** (First Sampling Lemma for kernels)**.** *Let $U \in \mathcal{W}_1$, and $\Lambda \in [0, 1]^k$ be uniform independent samples from $[0, 1]$. Then, with probability at least $1 - 4e^{-\sqrt{k}/10}$,*

$$-\frac{3}{k} \leq \|U[\Lambda]\|_\square - \|U\|_\square \leq \frac{8}{k^{1/4}}.$$

We derive a version of Lemma D.3 with expected value using the following lemma.

**Lemma D.4.** *Let $z : \Omega \to [0, 1]$ be a random variable over the probability space $\Omega$. Suppose that in an event $\mathcal{E} \subset \Omega$ of probability $1 - \epsilon$ we have $z < \alpha$. Then*

$$\mathbb{E}(z) \leq (1 - \epsilon)\alpha + \epsilon.$$

*Proof.*

$$\mathbb{E}(z) = \int_\Omega z(x)dx = \int_\mathcal{E} z(x)dx + \int_{\Omega \setminus \mathcal{E}} z(x)dx \le (1-\epsilon)\alpha + \epsilon.$$

∎

As a result of this lemma, we have a simple corollary of Lemma D.3.

**Corollary D.5** (First sampling lemma - expected value version)**.** *Let $U \in \mathcal{W}_1$ and $\Lambda \in [0,1]^k$ be chosen uniformly at random, where $k \ge 1$. Then*

$$\mathbb{E}\left| \|U[\Lambda]\|_\square - \|U\|_\square \right| \le \frac{14}{k^{1/4}}.$$

*Proof.* By Lemma D.4, and since $6/k^{1/4} > 4e^{-\sqrt{k}/10}$,

$$\mathbb{E}\left| \|U[\Lambda]\|_\square - \|U\|_\square \right| \le \left(1 - 4e^{-\sqrt{k}/10}\right)\frac{8}{k^{1/4}} + 4e^{-\sqrt{k}/10} < \frac{14}{k^{1/4}}.$$

∎

We note that a version of the first sampling lemma, Lemma D.3, for signals instead of kernels, is just a classical Monte Carlo approximation, when working with the $L_1[0,1]$ norm, which is equivalent to the signal cut norm.

**Lemma D.6** (First sampling lemma for signals)**.** *Let $f \in \mathcal{L}_r^\infty[0,1]$. Then*

$$\mathbb{E}\left| \|f(\Lambda)\|_1 - \|f\|_1 \right| \le \frac{r}{k^{1/2}}.$$

*Proof.* By standard Monte Carlo theory, since $r^2$ bounds the variance of $f(\lambda)$, where $\lambda$ is a random uniform sample from $[0,1]$, we have

$$\mathbb{V}(\|f(\Lambda)\|_1) = \mathbb{E}\left( \left| \|f(\Lambda)\|_1 - \|f\|_1 \right|^2 \right) \le \frac{r^2}{k}.$$

Here, $\mathbb{V}$ denotes variance, and we note that $\mathbb{E}\|f(\Lambda)\|_1 = \frac{1}{k}\sum_{j=1}^k |f(\lambda_j)| = \|f\|_1$. Hence, by Cauchy Schwarz inequality,

$$\mathbb{E}\left| \|f(\Lambda)\|_1 - \|f\|_1 \right| \le \sqrt{\mathbb{E}\left( \left| \|f(\Lambda)\|_1 - \|f\|_1 \right|^2 \right)} \le \frac{r}{k^{1/2}}.$$

∎

We now extend [24, Lemma 10.16], which bounds the cut distance between a graphon and its sampled graph, to the case of a sampled graphon-signal.

**Theorem D.7** (Second sampling lemma for graphon signals)**.** *Let $r > 1$. Let $k \ge K_0$, where $K_0$ is a constant that depends on $r$, and let $(W, f) \in \mathcal{WL}_r$. Then,*

$$\mathbb{E}\left( \delta_\square\big((W, f), (W(\Lambda), f(\Lambda))\big) \right) < \frac{15}{\sqrt{\log(k)}},$$

*and*

$$\mathbb{E}\left( \delta_\square\big((W, f), (\mathbb{G}(W, \Lambda), f(\Lambda))\big) \right) < \frac{15}{\sqrt{\log(k)}}.$$

The proof follows the steps of [24, Lemma 10.16] and [4]. We note that the main difference in our proof is that we explicitly write the measure preserving bijection that optimizes the cut distance. While this is not necessary in the classical case, where only a graphon is sampled, in our case we need to show that there is a measure preserving bijection that is shared by the graphon and the signal. We hence write the proof for completion.

*Proof.*

Denote a generic error bound, given by the regularity lemma Theorem B.8 by $\epsilon$. If we take $n$ intervals in the Theorem B.8 , then the error in the regularity lemma will be, for $c$ such that $2c = 3$,

$$\lceil 3/\epsilon^2 \rceil = \log(n)$$

so
$$3/\epsilon^2 + 1 \geq \log(n).$$
For small enough $\epsilon$, we increase the error bound in the regularity lemma to satisfy
$$4/\epsilon^2 > 3/\epsilon^2 + 1 \geq \log(n).$$
More accurately, for the equipartition to intervals $\mathcal{I}_n$, there is $\phi' \in S'_{[0,1]}$ and a piecewise constant graphon signal $([W^\phi]_n, [f^\phi]_n)$ such that
$$\|W^{\phi'} - [W^{\phi'}]_n\|_\square \leq \alpha \frac{2}{\sqrt{\log(n)}}$$
and
$$\|f^{\phi'} - [f^{\phi'}]_n\|_\square \leq (1 - \alpha) \frac{2}{\sqrt{\log(n)}},$$
for some $0 \leq \alpha \leq 1$. If we choose $n$ such that
$$n = \lceil \frac{\sqrt{k}}{r \log(k)} \rceil,$$
then an error bound in the regularity lemma is
$$\|W^{\phi'} - [W^{\phi'}]_n\|_\square \leq \alpha \frac{2}{\sqrt{\frac{1}{2}\log(k) - \log\big(\log(k)\big) - \log(r)}}$$
and
$$\|f^{\phi'} - [f^{\phi'}]_n\|_\square \leq (1 - \alpha) \frac{2}{\sqrt{\frac{1}{2}\log(k) - \log\big(\log(k)\big) - \log(r)}},$$
for some $0 \leq \alpha \leq 1$. Without loss of generality, we suppose that $\phi'$ is the identity. This only means that we work with a different representative of $[(W, f)] \in \widehat{\mathcal{WL}}_r$ throughout the proof. We hence have
$$d_\square(W, W_n) \leq \alpha \frac{2\sqrt{2}}{\sqrt{\log(k) - 2\log\big(\log(k)\big) - 2\log(r)}}$$
and
$$\|f - f_n\|_1 \leq (1 - \alpha) \frac{4\sqrt{2}}{\sqrt{\log(k) - 2\log\big(\log(k)\big) - 2\log(r)}},$$
for some step graphon-signal $(W_n, f_n) \in [\mathcal{WL}_r]_{\mathcal{I}_n}$.

Now, by the first sampling lemma (Corollary D.5),
$$\mathbb{E}\big|d_\square\big(W(\Lambda), W_n(\Lambda)\big) - d_\square(W, W_n)\big| \leq \frac{14}{k^{1/4}}.$$
Moreover, by the fact that $f - f_n \in \mathcal{L}_{2r}^\infty[0, 1]$, Lemma D.6 implies that
$$\mathbb{E}\big|\|f(\Lambda) - f_n(\Lambda)\|_1 - \|f - f_n\|_1\big| \leq \frac{2r}{k^{1/2}}.$$
Therefore,
$$\mathbb{E}\Big(d_\square\big(W(\Lambda), W_n(\Lambda)\big)\Big) \leq \mathbb{E}\big|d_\square\big(W(\Lambda), W_n(\Lambda)\big) - d_\square(W, W_n)\big| + d_\square(W, W_n)$$
$$\leq \frac{14}{k^{1/4}} + \alpha \frac{2\sqrt{2}}{\sqrt{\log(k) - 2\log\big(\log(k)\big) - 2\log(r)}}.$$
Similarly, we have
$$\mathbb{E}\|f(\Lambda) - f_n(\Lambda)\|_1 \leq \mathbb{E}\big|\|f(\Lambda) - f_n(\Lambda)\|_1 - \|f - f_n\|_1\big| + \|f - f_n\|_1$$
$$\leq \frac{2r}{k^{1/2}} + (1 - \alpha) \frac{4\sqrt{2}}{\sqrt{\log(k) - 2\log\big(\log(k)\big) - 2\log(r)}}.$$

Now, let $\pi_\Lambda$ be a sorting permutation in $[k]$, such that
$$\pi_\Lambda(\Lambda) := \{\Lambda_{\pi_\Lambda^{-1}(i)}\}_{i=1}^k = (\lambda_1', \ldots, \lambda_k')$$
is a sequence in a non-decreasing order. Let $\{I_k^i = [i-1, i)/k\}_{i=1}^k$ be the intervals of the equipartition $\mathcal{I}_k$. The sorting permutation $\pi_\Lambda$ induces a measure preserving bijection $\phi$ that sorts the intervals $I_k^i$. Namely, we define, for every $x \in [0, 1]$,
$$\text{if } x \in I_k^i, \quad \phi(x) = J_{i, \pi_\Lambda(i)}(x), \tag{24}$$
where $J_{i,j} : I_k^i \to I_k^j$ are defined as $x \mapsto x - i/k + j/k$, for all $x \in I_k^i$.

By abuse of notation, we denote by $W_n(\Lambda)$ and $f_n(\Lambda)$ the induced graphon and signal from $W_n$ and $f_n(\Lambda)$ respectively. Hence, $W_n(\Lambda)^\phi$ and $f_n(\Lambda)^\phi$ are well defined. Note that the graphons $W_n$ and $W_n(\Lambda)^\phi$ are stepfunctions, where the set of values of $W_n(\Lambda)^\phi$ is a subset of the set of values of $W_n$. Intuitively, since $k \gg m$, we expect the partition $\{[\lambda_i', \lambda_{i+1}')\}_{i=1}^k$ to be "close to a refinement" of $\mathcal{I}_n$ in high probability. Also, we expect the two sets of values of $W_n(\Lambda)^\phi$ and $W_n$ to be identical in high probability. Moreover, since $\Lambda'$ is sorted, when inducing a graphon from the graph $W_n(\Lambda)$ and "sorting" it to $W_n(\Lambda)^\phi$, we get a graphon that is roughly "aligned" with $W_n$. The same philosophy also applied to $f_n$ and $f_n(\Lambda)^\phi$. We next formalize these observations.

For each $i \in [n]$, let $\lambda_{j_i}'$ be the smaller point of $\Lambda'$ that is in $I_n^i$, set $j_i = j_{i+1}$ if $\Lambda' \cap I_n^i = \emptyset$, and set $j_{n+1} = k + 1$. For every $i = 1, \ldots, n$, we call
$$J_i := [j_i - 1, j_{i+1} - 1)/k$$
the $i$-th step of $W_n(\Lambda)^\phi$ (which can be the empty set). Let $a_i = \frac{j_i - 1}{k}$ be the left edge point of $J_i$. Note that $a_i = |\Lambda \cap [0, i/n)|/k$ is distributed binomially (up to the normalization $k$) with $k$ trials and success in probability $i/n$.

$$\|W_n - W_n(\Lambda)^\phi\|_\square \le \|W_n - W_n(\Lambda)^\phi\|_1$$
$$= \sum_i \sum_k \int_{I_n^i \cap J_i} \int_{I_n^k \cap J_k} |W_n(x, y) - W_n(\Lambda)^\phi(x, y)| \, dx dy$$
$$+ \sum_i \sum_{j \ne i} \sum_k \sum_{l \ne k} \int_{I_n^i \cap J_j} \int_{I_n^k \cap J_l} |W_n(x, y) - W_n(\Lambda)^\phi(x, y)| \, dx dy$$
$$= \sum_i \sum_{j \ne i} \sum_k \sum_{l \ne k} \int_{I_n^i \cap J_j} \int_{I_n^k \cap J_l} |W_n(x, y) - W_n(\Lambda)^\phi(x, y)| \, dx dy$$
$$= \sum_i \sum_k \int_{I_n^i \setminus J_i} \int_{I_n^k \setminus J_k} |W_n(x, y) - W_n(\Lambda)^\phi(x, y)| \, dx dy$$
$$\le \sum_i \sum_k \int_{I_n^i \setminus J_i} \int_{I_n^k \setminus J_k} 1 \, dx dy \le 2 \sum_i \int_{I_n^i \setminus J_i} 1 \, dx dy$$
$$\le 2 \sum_i (|i/n - a_i| + |(i+1)/n - a_{i+1}|).$$

Hence,
$$\mathbb{E}\|W_n - W_n(\Lambda)^\phi\|_\square \le 2 \sum_i (\mathbb{E}|i/n - a_i| + \mathbb{E}|(i+1)/n - a_{i+1}|)$$
$$\le 2 \sum_i \left( \sqrt{\mathbb{E}(i/n - a_i)^2} + \sqrt{\mathbb{E}((i+1)/n - a_{i+1})^2} \right)$$

By properties of the binomial distribution, we have $\mathbb{E}(k a_i) = ik/n$, so
$$\mathbb{E}(ik/n - k a_i)^2 = \mathbb{V}(k a_i) = k(i/n)(1 - i/n).$$

As a result
$$\mathbb{E}\|W_n - W_n(\Lambda)^\phi\|_\square \le 5 \sum_{i=1}^n \sqrt{\frac{(i/n)(1 - i/n)}{k}}$$
$$\le 2 \int_1^n \sqrt{\frac{(i/n)(1 - i/n)}{k}} \, di,$$

and for $n > 10$,

$$\leq 5\frac{n}{\sqrt{k}}\int_0^{1.1}\sqrt{z-z^2}dz \leq 5\frac{n}{\sqrt{k}}\int_0^{1.1}\sqrt{z}dz \leq 10/3(1.1)^{3/2}\frac{n}{\sqrt{k}} < 4\frac{n}{\sqrt{k}}.$$

Now, by $n = \lceil\frac{\sqrt{k}}{r\log(k)}\rceil \leq \frac{\sqrt{k}}{r\log(k)} + 1$, for large enough $k$,

$$\mathbb{E}\|W_n - W_n(\Lambda)^\phi\|_\square \leq 4\frac{1}{r\log(k)} + 4\frac{1}{\sqrt{k}} \leq \frac{5}{r\log(k)}.$$

Similarly,

$$\mathbb{E}\|f_n - f_n(\Lambda)^\phi\|_1 \leq \frac{5}{\log(k)}.$$

Note that in the proof of Corollary B.7, in (16), $\alpha$ is chosen close to 1, and especially, for small enough $\epsilon$, $\alpha > 1/2$. Hence, for large enough $k$,

$$\mathbb{E}(d_\square(W, W(\Lambda)^\phi)) \leq d_\square(W, W_n) + \mathbb{E}\big(d_\square(W_n, W_n(\Lambda)^\phi)\big) + \mathbb{E}(d_\square(W_n(\Lambda), W(\Lambda)))$$

$$\leq \alpha\frac{2\sqrt{2}}{\sqrt{\log(k) - 2\log\big(\log(k)\big) - 2\log(r)}} + \frac{5}{r\log(k)} + \frac{14}{k^{1/4}}$$

$$+ \alpha\frac{2\sqrt{2}}{\sqrt{\log(k) - 2\log\big(\log(k)\big) - 2\log(r)}}$$

$$\leq \alpha\frac{6}{\sqrt{\log(k)}},$$

Similarly, for each $k$, if $1 - \alpha < \frac{1}{\sqrt{\log(k)}}$, then

$$\mathbb{E}(d_\square(f, f(\Lambda)^\phi)) \leq (1 - \alpha)\frac{2\sqrt{2}}{\sqrt{\log(k) - 2\log\big(\log(k)\big) - 2\log(r)}} + \frac{5}{\log(k)}$$

$$+ \frac{2r}{k^{1/2}} + (1 - \alpha)\frac{4\sqrt{2}}{\sqrt{\log(k) - 2\log\big(\log(k)\big) - 2\log(r)}} \leq \frac{14}{\log(k)}.$$

Moreover, for each $k$ such that $1 - \alpha > \frac{1}{\sqrt{\log(k)}}$, if $k$ is large enough (where the lower bound of $k$ depends on $r$), we have

$$\frac{5}{\log(k)} + \frac{2r}{k^{1/2}} < \frac{5.5}{\log(k)} < \frac{1}{\sqrt{\log(k)}}\frac{6}{\sqrt{\log(k)}} < (1 - \alpha)\frac{6}{\sqrt{\log(k)}}$$

so, by $6\sqrt{2} < 9$,

$$\mathbb{E}(d_\square(f, f(\Lambda)^\phi)) \leq (1 - \alpha)\frac{2\sqrt{2}}{\sqrt{\log(k) - 2\log\big(\log(k)\big) - 2\log(r)}} + \frac{2}{\log(k)}$$

$$+ \frac{2r}{k^{1/2}} + (1 - \alpha)\frac{4\sqrt{2}}{\sqrt{\log(k) - 2\log\big(\log(k)\big) - 2\log(r)}}$$

$$\leq (1 - \alpha)\frac{15}{\sqrt{\log(k)}}.$$

Lastly, by Corollary D.2,

$$\mathbb{E}\Big(d_\square\big(W, \mathbb{G}(W, \Lambda)^\phi\big)\Big) \leq \mathbb{E}\Big(d_\square\big(W, W(\Lambda)^\phi\big)\Big) + \mathbb{E}\Big(d_\square\big(W(\Lambda)^\phi, \mathbb{G}(W, \Lambda)^\phi\big)\Big)$$

$$\leq \alpha\frac{6}{\sqrt{\log(k)}} + \frac{11}{\sqrt{k}} \leq \alpha\frac{7}{\sqrt{\log(k)}},$$

As a result, for large enough $k$,

$$\mathbb{E}\Big(\delta_\square\big((W,f),(W(\Lambda),f(\Lambda))\big)\Big) < \frac{15}{\sqrt{\log(k)}},$$

and

$$\mathbb{E}\Big(\delta_\square\big((W,f),(\mathbb{G}(W,\Lambda),f(\Lambda))\big)\Big) < \frac{15}{\sqrt{\log(k)}}.$$

∎

# E  Graphon-signal MPNNs

In this appendix we give properties and examples of MPNNs.

## E.1  Properties of graphon-signal MPNNs

Consider the construction of MPNN from Section 4.1. We first explain how a MPNN on a grpah is equivalent to a MPNN on the induced graphon.

Let $G$ be a graph of $n$ nodes, with adjacency matrix $A = \{a_{i,j}\}_{i,j\in[n]}$ and signal $\mathbf{f} \in \mathbb{R}^{n\times d}$. Consider a MPL $\theta$, with receiver and transmitter message functions $\xi_r^k, \xi_t^k : \mathbb{R}^d \to \mathbb{R}^p$, for $k \in [K]$, where $K \in \mathbb{N}$, and update function $\mu : \mathbb{R}^{d+p} \to \mathbb{R}^s$. The application of the MPL on $(G, \mathbf{f})$ is defined as follows. We first define the message kernel $\Phi_{\mathbf{f}} : [n]^2 \to \mathbb{R}^p$, with entries

$$\Phi_{\mathbf{f}}(i,j) = \Phi(\mathbf{f}_i, \mathbf{f}_j) = \sum_{k=1}^K \xi_r^k(\mathbf{f}_i)\xi_t^k(\mathbf{f}_j).$$

We then aggregate the message kernel with normalized sum aggregation

$$\big(\mathrm{Agg}(G, \Phi_{\mathbf{f}})\big)_i = \frac{1}{n}\sum_{j\in[n]} a_{i,j}\Phi_{\mathbf{f}}(i,j).$$

Lastly, we apply the update function, to obtain the output $\theta(G, \mathbf{f})$ of the MPL with value at each node $i$

$$\theta(G, \mathbf{f})_i = \eta\Big(\mathbf{f}_i, \big(\mathrm{Agg}(G, \Phi_{\mathbf{f}})\big)_i\Big) \in \mathbb{R}^s.$$

**Lemma E.1.** *Consider a MPL $\theta$ as in the above setting. Then, for every graph signal $(G, A, \mathbf{f})$,*

$$\theta\Big((W,f)_{(G,\mathbf{f})}\Big) = (W,f)_{\theta(G,\mathbf{f})}.$$

*Proof.* Let $\{I_1, \ldots, I_n\}$ be the equipartition to intervals. For each $j \in [n]$, let $y_j \in I_j$ be an arbitrary point. Let $i \in [n]$ and $x \in I_i$. We have

$$\mathrm{Agg}(G, \Phi_{\mathbf{f}})_i = \frac{1}{n}\sum_{j\in[n]} a_{i,j}\Phi_{\mathbf{f}}(i,j) = \frac{1}{n}\sum_{j\in[n]} W_G(x, y_j)\Phi_{f_{\mathbf{f}}}(x, y_j)$$

$$= \int_0^1 W_G(x,y)\Phi_{f_{\mathbf{f}}}(x,y)dy = \mathrm{Agg}(W_G, \Phi_{f_{\mathbf{f}}})(x).$$

Therefore, for every $i \in [n]$ and every $x \in I_i$,

$$f_{\theta(G,\mathbf{f})}(x) = f_{\eta\big(\mathbf{f}, \mathrm{Agg}(G,\Phi_{\mathbf{f}})\big)}(x) = \eta\big(\mathbf{f}_i, \mathrm{Agg}(G, \Phi_{\mathbf{f}})_i\big)$$

$$= \eta\big(f_{\mathbf{f}}(x), \mathrm{Agg}(W_G, \Phi_{f_{\mathbf{f}}})(x)\big) = \theta(W_G, f_{\mathbf{f}})(x).$$

∎

### E.2 Examples of MPNNs

The GIN convolutional layer [36] is defined as follows. First, the message function is

$$\Phi(a,b) = b$$

and the update function is

$$\eta(x,y) = M\big((1+\epsilon)x + y\big).$$

where $M$ is a multi-layer perceptron (MLP) and $\epsilon$ a constant. Each layer may have a different MLP and different constant $\epsilon$. The standard GIN is defined with sum aggregation, but we use normalized sum aggregation.

Given a graph-signal $(G, \mathbf{f})$, with $\mathbf{f} \in \mathbb{R}^{n \times d}$ with adjacency matrix $A \in \mathbb{R}^{n \times n}$, a spectral convolutional layer based on a polynomial filter $p(\lambda) = \sum_{j=0}^{J} \lambda^j C_j$, where $C_j \in \mathbb{R}^{d \times p}$, is defined to be

$$p(A)\mathbf{f} = \sum_{j=0}^{J} \frac{1}{n^j} A^j \mathbf{f} C_j,$$

followed by a pointwise non-linearity like ReLU. Such a convolutional layer can be seen as $J+1$ MPLs. We first apply $J$ MPLs, where each MPL is of the form

$$\theta(\mathbf{f}) = \big(\mathbf{f}, \frac{1}{n} A\mathbf{f}\big).$$

We then apply an update layer

$$U(\mathbf{f}) = \mathbf{f}C$$

for some $C \in \mathbb{R}^{(J+1)d \times p}$, followed by the pointwise non-linearity. The message part of $\theta$ can be written in our formulation with $\Phi(a,b) = b$, and the update part of $\theta$ with $\eta(c,d) = (c,d)$. The last update layer $U$ is linear followed by the pointwise non-linearity.

## F  Lipschitz continuity of MPNNs

In this appendix we prove Theorem 4.1. For $v \in \mathbb{R}^d$, we often denote by $|v| = \|v\|_\infty$. We define the $L_1$ norm of a measurable function $h : [0,1] \to \mathbb{R}^d$ by

$$\|h\|_1 := \int_0^1 |h(x)|\, dx = \int_0^1 \|h(x)\|_\infty dx.$$

Similarly,

$$\|h\|_\infty := \sup_{x \in \mathbb{R}^d} |h(x)| = \sup_{x \in \mathbb{R}^d} \|h(x)\|_\infty.$$

We define Lipschitz continuity with respect to the infinity norm. Namely, $Z : \mathbb{R}^d \to \mathbb{R}^c$ is called Lipschitz continuous with Lipschitz constant $L$ if

$$|Z(x) - Z(y)| = \|Z(x) - Z(y)\|_\infty \le L\|x - z\|_\infty = L\,|x - z|\,.$$

We denote the minimal Lipschitz bound of the function $Z$ by $L_Z$.

We extend $\mathcal{L}_r^\infty[0,1]$ to the space of functions $f : [0,1] \to \mathbb{R}^d$ with the above $L_1$ norm.

Define the space $\mathcal{K}_q$ of *kernels* bounded by $q > 0$ to be the space of measurable functions

$$K : [0,1]^2 \to [-q, q].$$

The cut norm, cut metric, and cut distance are defined as usual for kernels in $\mathcal{K}_q$.

### F.1  Lipschitz continuity of message passing and update layers

In this subsection we prove that message passing layers and update layers are Lipschitz continuous with respect to he graphon-signal cut metric.

**Lemma F.1** (Product rule for message kernels). *Let $\Phi_f, \Phi_g$ be the message kernels corresponding to the signals $f, g$. Then*

$$\|\Phi_f - \Phi_g\|_{L^1[0,1]^2} \le \sum_{k=1}^K \left( L_{\xi_r^k} \|\xi_t^k\|_\infty + \|\xi_r^k\|_\infty L_{\xi_t^k} \right) \|f - g\|_1.$$

*Proof.* Suppose $p = 1$ For every $x, y \in [0,1]^2$

$$|\Phi_f(x,y) - \Phi_g(x,y)| = \left| \sum_{k=1}^K \xi_r^k(f(x))\xi_t^k(f(y)) - \sum_{k=1}^K \xi_r^k(g(x))\xi_t^k(g(y)) \right|$$

$$\le \sum_{k=1}^K \left| \xi_r^k(f(x))\xi_t^k(f(y)) - \xi_r^k(g(x))\xi_t^k(g(y)) \right|$$

$$\le \sum_{k=1}^K \left( \left| \xi_r^k(f(x))\xi_t^k(f(y)) - \xi_r^k(g(x))\xi_t^k(f(y)) \right| + \left| \xi_r^k(g(x))\xi_t^k(f(y)) - \xi_r^k(g(x))\xi_t^k(g(y)) \right| \right)$$

$$\le \sum_{k=1}^K \left( L_{\xi_r^k} |f(x) - g(x)| \left| \xi_t^k(f(y)) \right| + \left| \xi_r^k(g(x)) \right| L_{\xi_t^k} |f(y) - g(y)| \right).$$

Hence,

$$\|\Phi_f - \Phi_g\|_{L^1[0,1]^2}$$

$$\le \sum_{k=1}^K \int_0^1 \int_0^1 \left( L_{\xi_r^k} |f(x) - g(x)| \left| \xi_t^k(f(y)) \right| + \left| \xi_r^k(g(x)) \right| L_{\xi_t^k} |f(y) - g(y)| \right) dx dy$$

$$\le \sum_{k=1}^K \left( L_{\xi_r^k} \|f - g\|_1 \|\xi_t^k\|_\infty + \|\xi_r^k\|_\infty L_{\xi_t^k} \|f - g\|_1 \right)$$

$$= \sum_{k=1}^K \left( L_{\xi_r^k} \|\xi_t^k\|_\infty + \|\xi_r^k\|_\infty L_{\xi_t^k} \right) \|f - g\|_1.$$

∎

**Lemma F.2.** *Let $Q, V$ be two message kernels, and $W \in \mathcal{W}_0$. Then*

$$\|\mathrm{Agg}(W,Q) - \mathrm{Agg}(W,V)\|_1 \le \|Q - V\|_1.$$

*Proof.*

$$\mathrm{Agg}(W,Q)(x) - \mathrm{Agg}(W,V)(x) = \int_0^1 W(x,y)(Q(x,y) - V(x,y))dy$$

So

$$\|\mathrm{Agg}(W,Q) - \mathrm{Agg}(W,V)\|_1 = \int_0^1 \left| \int_0^1 W(x,y)(Q(x,y) - V(x,y))dy \right| dx$$

$$\le \int_0^1 \int_0^1 |W(x,y)(Q(x,y) - V(x,y))|\, dy dx$$

$$\le \int_0^1 \int_0^1 |(Q(x,y) - V(x,y))|\, dy dx = \|Q - V\|_1.$$

∎

As a result of [Lemma F.2](#) and the product rule [Lemma F.1](#), we have the following corollary, that computes the error in aggregating two message kernels with the same graphon.

**Corollary F.3.**

$$\|\mathrm{Agg}(W,\Phi_f) - \mathrm{Agg}(W,\Phi_g)\|_1 \le \sum_{k=1}^K \left( L_{\xi_r^k} \|\xi_t^k\|_\infty + \|\xi_r^k\|_\infty L_{\xi_t^k} \right) \|f - g\|_1.$$

Next we fix the message kernel, and bound the difference between the aggregation of the message kernel with respect to two different graphons. Let $L^+[0,1]$ be the space of measurable function $f : [0,1] \to [0,1]$. The following lemma is a trivial extension of [24, Lemma 8.10] from $\mathcal{K}_1$ to $\mathcal{K}_r$.

**Lemma F.4.** *For any kernel $Q \in \mathcal{K}_r$*

$$\|Q\|_\square = \sup_{f,g \in L^+[0,1]} \left| \int_{[0,1]^2} f(x)Q(x,y)g(y)dxdy \right|,$$

*where the supremum is attained for some $f, g \in L^+[0,1]$.*

The following Lemma is proven as part of the proof of [24, Lemma 8.11].

**Lemma F.5.** *For any kernel $Q \in \mathcal{K}_r$*

$$\sup_{f,g \in L_1^\infty[0,1]} \left| \int_{[0,1]^2} f(x)Q(x,y)g(y)dxdy \right| \le 4\|Q\|_\square.$$

For completeness, we give here a self-contained proof.

*Proof.* Any function $f \in L_1^\infty[0,1]$ can be written as $f = f_+ - f_-$, where $f_+, f_- \in L^+[0,1]$. Hence, by Lemma F.4,

$$\sup_{f,g \in L_1^\infty[0,1]} \left| \int_{[0,1]^2} f(x)Q(x,y)g(y)dxdy \right|$$

$$= \sup_{f_+,f_-,g_+,g_- \in L^+[0,1]} \left| \int_{[0,1]^2} (f_+(x) - f_-(x))Q(x,y)(g_+(y) - g_-(y))dxdy \right|$$

$$\le \sum_{s \in \{+,-\}} \sup_{f_s,g_s \in L^+[0,1]} \left| \int_{[0,1]^2} f_s(x)Q(x,y)g_s(y)dxdy \right| = 4\|Q\|_\square.$$

∎

Next we state a simple lemma.

**Lemma F.6.** *Let $f = f_+ - f_-$ be a signal, where $f_+, f_- : [0,1] \to (0,\infty)$ are measurable. Then the supremum in the cut norm $\|f\|_\square = \sup_{S \subset [0,1]} \left| \int_S f(x)dx \right|$ is attained as the support of either $f_+$ or $f_-$.*

**Lemma F.7.** *Let $f \in \mathcal{L}_r^\infty[0,1]$ , $W, V \in \mathcal{W}_0$, and suppose that $\left| \xi_r^k(f(x)) \right|, \left| \xi_t^k(f(x)) \right| \le \rho$ for every $x \in [0,1]$ and $k = 1, \dots, K$. Then*

$$\|\mathrm{Agg}(W, \Phi_f) - \mathrm{Agg}(V, \Phi_f)\|_\square \le 4K\rho^2 \|W - V\|_\square.$$

*Moreover, if $\xi_r^k$ and $\xi_t^k$ are non-negatively valued for every $k = 1, \dots, K$, then*

$$\|\mathrm{Agg}(W, \Phi_f) - \mathrm{Agg}(V, \Phi_f)\|_\square \le K\rho^2 \|W - V\|_\square.$$

*Proof.* Let $T = W - V$. Let $S$ be the maximizer of the supremum underlying the cut norm of $\mathrm{Agg}(T, \Phi_f)$. Suppose without loss of generality that $\int_S \mathrm{Agg}(T, \Phi_f)(x)dx > 0$. Denote $q_r^k(x) = \xi_r^k(f(x))$ and $q_t^k(x) = \xi_t^k(f(x))$. We have

$$\int_S \big(\mathrm{Agg}(W, \Phi_f)(x) - \mathrm{Agg}(V, \Phi_f)(x)\big)dx = \int_S \mathrm{Agg}(T, \Phi_f)(x)dx$$

$$= \sum_{k=1}^K \int_S \int_0^1 q_r^k(x)T(x,y)q_t^k(y)dydx.$$

Let

$$v_r^k(x) = \begin{cases} q_r^k(x)/\rho & x \in S \\ 0 & x \notin S. \end{cases} \tag{25}$$

Moreover, define $v_t^k = q_t^k/\rho$, and note that $v_r^k, v_t^k \in L_1^\infty[0,1]$. We hence have, by Lemma F.5,

$$\int_S \text{Agg}(T, \Phi_f)(x)dx = \sum_{k=1}^K \rho^2 \int_0^1 \int_0^1 v_r^k(x)T(x,y)v_t^k(y)dydx$$

$$\leq \sum_{k=1}^K \rho^2 \left| \int_0^1 \int_0^1 v_r^k(x)T(x,y)v_t^k(y)dydx \right|$$

$$\leq 4K\rho^2 \|T\|_\square.$$

Hence,

$$\|\text{Agg}(W, \Phi_f) - \text{Agg}(V, \Phi_f)\|_\square \leq 4K\rho^2 \|T\|_\square$$

Lastly, in case $\xi_r^k, \xi_t^k$ are nonnegatively valued, so are $q_r^k, q_t^k$, and hence by Lemma F.4,

$$\int_S \text{Agg}(T, \Phi_f)(x)dx \leq K\rho^2 \|T\|_\square.$$

∎

**Theorem F.8.** *Let $(W, f), (V, g) \in \mathcal{WL}_r$, and suppose that $\left|\xi_r^k(f(x))\right|, \left|\xi_t^k(f(x))\right| \leq \rho$ and $L_{\xi_t^k}, L_{\xi_t^k} < L$ for every $x \in [0,1]$ and $k = 1, \ldots, K$. Then,*

$$\|\text{Agg}(W, \Phi_f) - \text{Agg}(V, \Phi_g)\|_\square \leq 4KL\rho\|f - g\|_\square + 4K\rho^2 \|W - V\|_\square.$$

*Proof.* By Lemma F.1, Lemma F.2 and Lemma F.7,

$$\|\text{Agg}(W, \Phi_f) - \text{Agg}(V, \Phi_g)\|_\square$$
$$\leq \|\text{Agg}(W, \Phi_f) - \text{Agg}(W, \Phi_g)\|_\square + \|\text{Agg}(W, \Phi_g) - \text{Agg}(V, \Phi_g)\|_\square$$
$$\leq \sum_{k=1}^K \left( L_{\xi_r^k}\|\xi_t^k\|_\infty + \|\xi_r^k\|_\infty L_{\xi_t^k} \right)\|f - g\|_1 + 4K\rho^2 \|W - V\|_\square$$
$$\leq 4KL\rho\|f - g\|_\square + 4K\rho^2 \|W - V\|_\square.$$

∎

Lastly, we show that update layers are Lipschitz continuous. Since the update function takes two functions $f : [0,1] \to \mathbb{R}^{d_i}$ (for generally two different output dimensions $d_1, d_2$), we "concatenate" these two inputs and treat it as one input $f : [0,1] \to \mathbb{R}^{d_1+d_2}$.

**Lemma F.9.** *Let $\eta : \mathbb{R}^{d+p} \to \mathbb{R}^s$ be Lipschitz with Lipschitz constant $L_\eta$, and let $f, g \in \mathcal{L}_r^\infty[0,1]$ with values in $\mathbb{R}^{d+p}$ for some $d, p \in \mathbb{N}$.*

*Then*

$$\|\eta(f) - \eta(g)\|_1 \leq L_\eta\|f - g\|_1.$$

*Proof.*

$$\|\eta(f) - \eta(g)\|_1 = \int_0^1 \left|\eta\big(f(x)\big) - \eta\big(g(x)\big)\right| dx$$

$$\leq \int_0^1 L_\eta \left|f(x) - g(x)\right| dx = L_\eta\|f - g\|_1.$$

∎

### F.2 Bounds of signals and MPLs with Lipschitz message and update functions

We will consider three settings for the MPNN Lipschitz bounds. In all settings, the transmitter, receiver, and update functions are Lipschitz. In the first setting all message and update functions are assumed to be bounded. In the second setting, there is no additional assumption over Lipschtzness of the transmitter, receiver, and update functions. In the third setting, we assume that the message function $\Phi$ is also Lipschitz with Lipschitz bound $L_\Phi$, and that all receiver and transmitter functions

are non-negatively bounded (e.g., via an application of ReLU or sigmoid in their implementation). Note that in case $K = 1$ and all functions are differentiable, by the product rule, $\Phi$ can be Lipschitz only in two cases: if both $\xi_r$ and $\xi_t$ are bounded and Lipschitz, or if either $\xi_r$ or $\xi_t$ is constant, and the other function is Lipschitz. When $K > 1$, we can have combinations of these cases.

We next derive bounds for the different settings. A bound for setting 1 is given in Theorem F.8. Moreover, When the receiver and transmitter message functions and the update functions are bounded, so is the signal at each layer.

**Bounds for setting 2.**

Next we show boundedness when the receiver and transmitter message and update functions are only assumed to be Lipschitz.

Define the *formal bias* $B_\xi$ of a function $\xi : \mathbb{R}^{d_1} \to \mathbb{R}^{d_2}$ to be $\xi(0)$ [26]. We note that the formal bias of an affine-linear operator is its classical bias.

**Lemma F.10.** *Let* $(W, f) \in \mathcal{WL}_r$, *and suppose that for every* $y \in \{r, t\}$ *and* $k = 1, \ldots, K$

$$\left| \xi_y^k(0) \right| \leq B, \quad L_{\xi_y^k} < L.$$

*Then,*

$$\| \xi_y^k \circ f \|_\infty \leq Lr + B$$

*and*

$$\| \mathrm{Agg}(W, \Phi_f) \|_\infty \leq K(Lr + B)^2.$$

*Proof.* Let $y \in \{r, t\}$. We have

$$\left| \xi_y^k(f(x)) \right| \leq \left| \xi_y^k(f(x)) - \xi_y^k(0) \right| + B \leq L_{\xi_y^k} |f(x)| + B \leq Lr + B,$$

so,

$$|\mathrm{Agg}(W, \Phi_f)(x)| = \left| \sum_{k=1}^{K} \int_0^1 \xi_r^k(f(x)) W(x, y) \xi_t^k(f(y)) dy \right|$$
$$\leq K(Lr + B)^2.$$

$\blacksquare$

Next, we have a direct result of Theorem F.8.

**Corollary F.11.** *Suppose that for every* $y \in \{r, t\}$ *and* $k = 1, \ldots, K$

$$\left| \xi_y^k(0) \right| \leq B, \quad L_{\xi_y^k} < L.$$

*Then, for every* $(W, f), (V, g) \in \mathcal{WL}_r$,

$$\| \mathrm{Agg}(W, \Phi_f) - \mathrm{Agg}(V, \Phi_g) \|_\square \leq 4K(L^2 r + LB) \| f - g \|_\square + 4K(Lr + B)^2 \| W - V \|_\square.$$

**Bound for setting 3.**

**Lemma F.12.** *Let* $(W, f) \in \mathcal{WL}_r$, *and suppose that*

$$|\Phi(0, 0)| < B, \quad L_\Phi < L.$$

*Then,*

$$\| \Phi_f \|_\infty \leq Lr + B$$

*and*

$$\| \mathrm{Agg}(W, \Phi_f) \|_\infty \leq Lr + B.$$

*Proof.* We have

$$|\Phi(f(x), f(y))| \leq |\Phi(f(x), f(y)) - \Phi(0, 0)| + B \leq L_\Phi |(f(x), f(y))| + B \leq Lr + B,$$

so,

$$|\mathrm{Agg}(W, \Phi_f)(x)| = \left| \int_0^1 W(x, y) \Phi(f(x), f(y)) dy \right|$$
$$\leq Lr + B.$$

$\blacksquare$

**Additional bounds.**

**Lemma F.13.** *Let $f$ be a signal, $W, V \in \mathcal{W}_0$, and suppose that $\|\Phi_f\|_\infty \leq \rho$ for every $k = 1, \ldots, K$, and that $\xi_r^k$ and $\xi_t^k$ are non-negatively valued. Then*

$$\|\mathrm{Agg}(W, \Phi_f) - \mathrm{Agg}(V, \Phi_f)\|_\square \leq K\rho\|W - V\|_\square.$$

*Proof.* The proof follows the steps of Lemma F.7 until (25), from where we proceed differently. Since all of the functions $q_r^k$ and $q_t^k$, $k \in [K]$, and since $\|\Phi_f\|_\infty \leq \rho$, the product of each $q_r^k(x)q_t^k(y)$ must be also bounded by $\rho$ for every $x \in [0,1]$ and $k \in [K]$. Hence, we may replace the normalization in (25) with

$$v_r^k(x) = \begin{cases} q_r^k(x)/\rho_r^k & x \in S \\ 0 & x \notin S \end{cases}, \quad v_t^k(y) = \begin{cases} q_t^k(y)/\rho_t^k & y \in S \\ 0 & y \notin S, \end{cases}$$

where for every $k \in [K]$, $\rho_r^k \rho_t^k = \rho$. This guarantees that $v_r^k, v_t^k \in L_1^\infty[0,1]$. Hence,

$$\int_S \mathrm{Agg}(T, \Phi_f)(x)dx = \sum_{k=1}^K \int_0^1 \int_0^1 \rho_r^k v_r^k(x) T(x,y) \rho_t^k v_t^k(y) dy dx$$

$$\leq \sum_{k=1}^K \rho \left| \int_0^1 \int_0^1 v_r^k(x) T(x,y) v_t^k(y) dy dx \right| \leq K\rho\|T\|_\square.$$

∎

**Theorem F.14.** *Let $(W, f), (V, g) \in \mathcal{WL}_r$, and suppose that $\|\Phi\|_\infty, \|\xi_r^k\|_\infty, \|\xi_t^k\|_\infty \leq \rho$, all message functions $\xi$ are non-negative valued, and $L_{\xi_t^k}, L_{\xi_t^k} < L$, for every $k = 1, \ldots, K$. Then,*

$$\|\mathrm{Agg}(W, \Phi_f) - \mathrm{Agg}(V, \Phi_g)\|_\square \leq 4KL\rho\|f - g\|_\square + K\rho\|W - V\|_\square.$$

The proof follows the steps of Theorem F.8.

**Corollary F.15.** *Suppose that for every $\mathrm{y} \in \{\mathrm{r}, \mathrm{t}\}$ and $k = 1, \ldots, K$*

$$|\Phi(0,0)|, \left|\xi_\mathrm{y}^k(0)\right| \leq B, \quad L_\phi, L_{\xi_\mathrm{y}^k} < L,$$

*and $\xi, \Phi$ are all non-negatively valued. Then, for every $(W, f), (V, g) \in \mathcal{WL}_r$,*

$$\|\mathrm{Agg}(W, \Phi_f) - \mathrm{Agg}(V, \Phi_g)\|_\square \leq 4K(L^2 r + LB)\|f - g\|_\square + K(Lr + B)\|W - V\|_\square.$$

The proof follows the steps of Corollary F.11.

### F.3 Lipschitz continuity theorems for MPNNs

The following recurrence sequence will govern the propagation of the Lipschitz constant of the MPNN and the bound of signal along the layers.

**Lemma F.16.** *Let $\mathbf{a} = (a_1, a_2, \ldots)$ and $\mathbf{b} = (b_1, b_2, \ldots)$. The solution to $e_{t+1} = a_t e_t + b_t$, with initialization $e_0$, is*

$$e_t = Z_t(\mathbf{a}, \mathbf{b}, e_0) := \prod_{j=0}^{t-1} a_j e_0 + \sum_{j=1}^{t-1} \prod_{i=1}^{j-1} a_{t-i} b_{t-j}, \tag{26}$$

*where, by convention,*

$$\prod_{i=1}^0 a_{t-i} := 1.$$

*In case there exist $a, b \in \mathbb{R}$ such that $a_i = a$ and $b_i = b$ for every $i$,*

$$e_t = a^t e_0 + \sum_{j=0}^{t-1} a^j b.$$

**Setting 1.**

**Theorem F.17.** *Let $\Theta$ be a MPNN with $T$ layers. Suppose that for every layer and every $\mathrm{y}$ and $k$,*

$$\|{}^t\xi_{\mathrm{y}}^k\|_\infty, \ \|\eta^t\|_\infty \leq \rho, \quad L_{\eta^t}, L_{{}^t\xi_{\mathrm{y}}^k} < L.$$

*Let $(W, f), (V, g) \in \mathcal{WL}_r$. Then, for MPNN with no update function*

$$\|\Theta_t(W, f) - \Theta_t(V, g)\|_\square \leq (4KL\rho)^t \|f - g\|_\square + \sum_{j=0}^{t-1} (4KL\rho)^j 4K\rho^2 \|W - V\|_\square,$$

*and for MPNN with update function*

$$\|\Theta_t(W, f) - \Theta_t(V, g)\|_\square \leq (4KL^2\rho)^t \|f - g\|_\square + \sum_{j=0}^{t-1} (4KL^2\rho)^j 4K\rho^2 L \|W - V\|_\square.$$

*Proof.* We prove for MPNNs with update function, where the proof without update function is similar. We can write a recurrence sequence for a bound $\|\Theta_t(W, f) - \Theta_t(V, g)\|_\square \leq e_t$, by Theorem F.8 and Lemma F.9, as

$$e_{t+1} = 4KL^2\rho e_t + 4K\rho^2 L \|W - V\|_\square.$$

The proof now follows by applying Lemma F.16 with $a = 4KL^2\rho$ and $b = 4K\rho^2 L$. ∎

**Setting 2.**

**Lemma F.18.** *Let $\Theta$ be a MPNN with $T$ layers. Suppose that for every layer $t$ and every $\mathrm{y} \in \{\mathrm{r}, \mathrm{t}\}$ and $k \in [K]$,*

$$\left|\eta^t(0)\right|, \ \left|{}^t\xi_{\mathrm{y}}^k(0)\right| \leq B, \quad L_{\eta^t}, \ L_{{}^t\xi_{\mathrm{y}}^k} < L$$

*with $L, B > 1$. Let $(W, f) \in \mathcal{WL}_r$. Then, for MPNN without update function, for every layer $t$,*

$$\|\Theta_t(W, f)\|_\infty \leq (2KL^2 B^2)^{2^t} \|f\|_\infty^{2^t},$$

*and for MPNN with update function, for every layer $t$,*

$$\|\Theta_t(W, f)\|_\infty \leq (2KL^3 B^2)^{2^t} \|f\|_\infty^{2^t},$$

*Proof.* We first prove for MPNNs without update functions. Denote by $C_t$ a bound on $\|{}^t f\|_\infty$, and let $C_0$ be a bound on $\|f\|_\infty$. By Lemma F.10, we may choose bounds such that

$$C_{t+1} \leq K(LC_t + B)^2 = KL^2 C_t^2 + 2KLBC_t + KB^2.$$

We can always choose $C_t, K, L > 1$, and therefore,

$$C_{t+1} \leq KL^2 C_t^2 + 2KLBC_t + KB^2 \leq 2KL^2 B^2 C_t^2.$$

Denote $a = 2KL^2 B^2$. We have

$$\begin{aligned}
C_{t+1} &= a(C_t)^2 = a(aC_{t-1}^2)^2 = a^{1+2} C_{t-1}^4 = a^{1+2}(a(C_{t-2})^2)^4 \\
&= a^{1+2+4}(C_{t-2})^8 = a^{1+2+4+8}(C_{t-3})^{16} \leq a^{2^t} C_0^{2^t}.
\end{aligned}$$

Now, for MPNNs with update function, we have

$$\begin{aligned}
C_{t+1} &\leq LK(LC_t + B)^2 + B \\
&= KL^3 C_t^2 + 2KL^2 BC_t + KB^2 L + B \\
&\leq 2KL^3 B^2 C_t^2,
\end{aligned}$$

and we proceed similarly.

∎

**Theorem F.19.** *Let $\Theta$ be a MPNN with $T$ layers. Suppose that for every layer $t$ and every $\mathrm{y} \in \{\mathrm{r}, \mathrm{t}\}$ and $k \in [K]$,*

$$\left|\eta^t(0)\right|, \ \left|{}^t\xi_{\mathrm{y}}^k(0)\right| \leq B, \quad L_{\eta^t}, \ L_{{}^t\xi_{\mathrm{y}}^k} < L,$$

*with $L, B > 1$. Let $(W, g), (V, g) \in \mathcal{WL}_r$. Then, for MPNNs without update functions*

$$\|\Theta_t(W, f) - \Theta_t(V, g)\|_{\square} \leq \prod_{j=0}^{t-1} 4K(L^2 r_j + LB)\|f - g\|_{\square}$$

$$+ \sum_{j=1}^{t-1} \prod_{i=1}^{j-1} 4K(L^2 r_{t-i} + LB) 4K(Lr_{t-j} + B)^2 \|W - V\|_{\square},$$

*where*

$$r_i = (2KL^2 B^2)^{2^i} \|f\|_{\infty}^{2^i},$$

*and for MPNNs with update functions*

$$\|\Theta_t(W, f) - \Theta_t(V, g)\|_{\square} \leq \prod_{j=0}^{t-1} 4K(L^3 r_j + L^2 B)\|f - g\|_{\square}$$

$$+ \sum_{j=1}^{t-1} \prod_{i=1}^{j-1} 4K(L^3 r_{t-i} + L^2 B) 4KL(Lr_{t-j} + B)^2 \|W - V\|_{\square},$$

*where*

$$r_i = (2KL^3 B^2)^{2^i} \|f\|_{\infty}^{2^i}.$$

*Proof.* We prove for MPNNs without update functions. The proof for the other case is similar. By [Corollary F.11](#), since the signals at layer $t$ are bounded by

$$r_t = (2KL^2 B^2)^{2^t} \|f\|_{\infty}^{2^t},$$

we have

$$\|\Theta_{t+1}(W, f) - \Theta_{t+1}(V, g)\|_{\square}$$
$$\leq 4K(L^2 r_t + LB)\|\Theta_t(W, f) - \Theta_t(V, g)\|_{\square} + 4K(Lr_t + B)^2 \|W - V\|_{\square}.$$

We hence derive a recurrence sequence for a bound $\|\Theta_t(W, f) - \Theta_t(V, g)\|_{\square} \leq e_t$, as

$$e_{t+1} = 4K(L^2 r_t + LB)e_t + 4K(Lr_t + B)^2 \|W - V\|_{\square}.$$

We now apply [Lemma F.16](#). ∎

**Setting 3.**

**Lemma F.20.** *Suppose that for every layer $t$ and every $\mathrm{y} \in \{\mathrm{r}, \mathrm{t}\}$ and $k = 1, \ldots, K$,*

$$\left|\eta^t(0)\right|, \ \left|\Phi^t(0, 0)\right|, \ \left|{}^t\xi_{\mathrm{y}}^k(0)\right| \leq B, \quad L_{\eta^t}, \ L_{\Phi^t}, \ L_{{}^t\xi_{\mathrm{y}}^k} < L,$$

*and $\xi, \Phi$ are all non-negatively valued. Then, for MPNNs without update function*

$$\|\Theta^t(W, f)\|_{\infty} \leq L^t \|f\|_{\infty} + \sum_{j=1}^{t-1} L^j B,$$

*and for MPNNs with update function*

$$\|\Theta^t(W, f)\|_{\infty} \leq L^{2t} \|f\|_{\infty} + \sum_{j=1}^{t-1} L^{2j}(LB + B),$$

*Proof.* We first prove for MPNNs without update functions. By [Lemma F.10](#), there is a bound $e_t$ of $\|\Theta^t(W, f)\|_{\infty}$ that satisfies

$$e_t = Le_{t-1} + B.$$

Solving this recurrence sequence via [Lemma F.16](#) concludes the proof.

Lastly, for MPNN with update functions, we have a bound that satisfies

$$e_t = L^2 e_{t-1} + LB + B,$$

and we proceed as before. ∎

**Lemma F.21.** *Suppose that for every* $\mathrm{y} \in \{\mathrm{r}, \mathrm{t}\}$ *and* $k = 1, \ldots, K$

$$\left|\eta^t(0)\right|, \ \left|\Phi(0,0)\right|, \left|\xi_{\mathrm{y}}^k(0)\right| \leq B, \quad L_\Phi, L_{\xi_{\mathrm{y}}^k} < L,$$

*and* $\xi, \Phi$ *are all non-negatively valued. Let* $(W, g), (V, g) \in \mathcal{WL}_r$. *Then, for MPNNs without update functions*

$$\|\Theta^t(W, \Phi_f) - \Theta^t(V, \Phi_g)\|_\square = O(K^t L^{2t+t^2} r^t B^t)\Big(\|W - V\|_\square + \|f - g\|_\square\Big),$$

*and for MPNNs with update functions*

$$\|\Theta^t(W, \Phi_f) - \Theta^t(V, \Phi_g)\|_\square = O(K^t L^{3t+2t^2} r^t B^t)\Big(\|W - V\|_\square + \|f - g\|_\square\Big)$$

*Proof.* We start with MPNNs without update functions. By [Corollary F.15](#) and [Lemma F.20](#), there is a bound $e_t$ on the error $\|\Theta^t(W, \Phi_f) - \Theta^t(V, \Phi_g)\|_\square$ at step $t$ that satisfies

$$e_t = 4K(L^2 r_{t-1} + LB)e_{t-1} + K(Lr + B)\|W - V\|_\square$$

$$= 4K\Big(L^2\big(L^t\|f\|_\infty + \sum_{j=1}^{t-1} L^j B\big) + LB\Big)e_{t-1} + K\Big(L\big(L^t\|f\|_\infty + \sum_{j=1}^{t-1} L^j B\big) + B\Big)\|W - V\|_\square.$$

Hence, by [Lemma F.16](#), and $Z$ defined by [(26)](#),

$$e_t = Z_t(\mathbf{a}, \mathbf{b}, \|f - g\|_\square) = O(K^t L^{2t+t^2} r^t B^t)\big(\|f - g\|_\square + \|W - V\|_\square\big),$$

where in the notations of [Lemma F.16](#),

$$a_t = 4K\Big(L^2(L^t\|f\|_\infty + \sum_{j=1}^{t-1} L^j B) + LB\Big)$$

and

$$b_t = K\Big(L(L^t\|f\|_\infty + \sum_{j=1}^{t-1} L^j B) + B\Big)\|W - V\|_\square.$$

Next, for MPNNs with update functions, there is a bound that satisfies

$$e_t = 4K(L^3 r_{t-1} + L^2 B)e_{t-1} + K(L^2 r + LB)\|W - V\|_\square$$

$$= 4K\Big(L^3\big(L^{2t}\|f\|_\infty + \sum_{j=1}^{t-1} L^{2j}(LB + B)\big) + L^2 B\Big)e_{t-1}$$

$$+ K\Big(L^2\big(L^{2t}\|f\|_\infty + \sum_{j=1}^{t-1} L^{2j}(LB + B)\big) + LB\Big)\|W - V\|_\square.$$

Hence, by [Lemma F.16](#), and $Z$ defined by [(26)](#),

$$e_t = O(K^t L^{3t+2t^2} r^t B^t)\big(\|f - g\|_\square + \|W - V\|_\square\big).$$

∎

# G   Generalization bound for MPNNs

In this appendix we prove [Theorem 4.2](#).

## G.1   Statistical learning and generalization analysis

In the statistical setting of learning, we suppose that the dataset comprises independent random samples from a probability space that describes all possible data $\mathcal{P}$. We suppose that for each $x \in \mathcal{P}$ there is a ground truth value $y_x \in \mathcal{Y}$, e.g., the ground truth class or value of $x$, where $\mathcal{Y}$ is, in general, some measure space. The *loss* is a measurable function $\mathcal{L} : \mathcal{Y}^2 \to \mathbb{R}_+$ that defines

similarity in $\mathcal{Y}$. Given a measurable function $\Theta : \mathcal{P} \to \mathcal{Y}$, that we call the *model* or *network*, its accuracy on all potential inputs is defined as the *statistical risk* $R_{\text{stat}}(\Theta) = \mathbb{E}_{x \sim \mathcal{P}}\Big(\mathcal{L}(\Theta(x), y_x)\Big)$. The goal in learning is to find a network $\Theta$, from some *hypothesis space* $\mathcal{T}$, that has a low statistical risk. In practice, the statistical risk cannot be computed analytically. Instead, we suppose that a dataset $\mathcal{X} = \{x_m\}_{m=1}^M \subset \mathcal{P}$ of $M \in \mathbb{N}$ random independent samples with corresponding values $\{y_m\}_{m=1}^M \subset \mathcal{Y}$ is given. We estimate the statistical risk via a "Monte Carlo approximation," called the *empirical risk* $R_{\text{emp}}(\Theta) = \frac{1}{M} \sum_{m=1}^M \mathcal{L}(\Theta(x_m), y_m)$. The network $\Theta$ is chosen in practice by optimizing the empirical risk. The goal in generalization analysis is to show that if a learned $\Theta$ attains a low empirical risk, then it is also guaranteed to have a low statistical risk.

One technique for bounding the statistical risk in terms of the empirical risk is to use the bound $R_{\text{stat}}(\Theta) \leq R_{\text{emp}}(\Theta) + E$, where $E$ is the *generalization error* $E = \sup_{\Theta \in \mathcal{T}} |R_{\text{stat}}(\Theta) - R_{\text{emp}}(\Theta)|$, and to find a bound for $E$. Since the trained network $\Theta = \Theta_{\mathcal{X}}$ depends on the data $\mathcal{X}$, the network is not a constant when varying the dataset, and hence the empirical risk is not really a Monte Carlo approximation of the statistical risk in the learning setting. If the network $\Theta$ was fixed, then Monte Carlo theory would have given us a bound of $E^2$ of order $O\big(\kappa(p)/M\big)$ in an event of probability $1 - p$, where, for example, in Hoeffding's inequality Theorem G.2, $\kappa(p) = \log(2/p)$. Let us call such an event a *good sampling event*. Since the good sampling event depends on $\Theta$, computing a naive bound to the generalization error would require intersecting all good sampling events for all $\Theta \in \mathcal{T}$. Uniform convergence bounds are approaches for intersecting adequate sampling events that allow bounding the generalization error more efficiently. This intersection of events leads to a term in the generalization bound, called the *complexity/capacity*, that describes the richness of the hypothesis space $\mathcal{T}$. This is the philosophy behind approaches such as VC-dimension, Rademacher dimension, fat-shattering dimension, pseudo-dimension, and uniform covering number (see, e.g., [34]).

## G.2 Classification setting

We define a ground truth classifier into $C$ classes as follows. Let $\mathcal{C} : \widetilde{\mathcal{WL}_r} \to \mathbb{R}^C$ be a measurable piecewise constant function of the following form. There is a partition of $\mathcal{WL}_r$ into disjoint measurable sets $B_1, \ldots, B_C \subset \widetilde{\mathcal{WL}_r}$ such that $\bigcup_{i=1}^C B_i = \widetilde{\mathcal{WL}_r}$, and for every $i \in [C]$ and every $x \in B_i$,

$$\mathcal{C}(x) = e_i,$$

where $e_i \in \mathbb{R}^C$ is the standard basis element with entries $(e_i)_j = \delta_{i,j}$, where $\delta_{i,j}$ is the Kronecker delta.

We define an arbitrary data distribution as follows. Let $\mathcal{B}$ be the Borel $\sigma$-algebra of $\widetilde{\mathcal{WL}_r}$, and $\nu$ be any probability measure on the measurable space $(\widetilde{\mathcal{WL}_r}, \mathcal{B})$. We may assume that we complete $\mathcal{B}$ with respect to $\nu$, obtaining the $\sigma$-algebra $\Sigma$. If we do not complete the measure, we just denote $\Sigma = \mathcal{B}$. Defining $(\widetilde{\mathcal{WL}_r}, \Sigma, \nu)$ as a complete measure space or not will not affect our construction.

Let $\mathcal{S}$ be a metric space. Let $\text{Lip}(\mathcal{S}, L)$ be the space of Lipschitz continuous mappings $\Upsilon : \mathcal{S} \to \mathbb{R}^C$ with Lipschitz constant $L$. Note that by Theorem 4.1, for every $i \in [C]$, the space of MPNN with Lipschitz continuous input and output message functions and Lipschitz update functions, restricted to $B_i$, is a subset of $\text{Lip}(B_i, L_1)$ which is the restriction of $\text{Lip}(\widetilde{\mathcal{WL}_r}, L_1)$ to $B_i \subset \widetilde{\mathcal{WL}_r}$, for some $L_1 > 0$. Moreover, $B_i$ has finite covering $\kappa(\epsilon)$ given in (23). Let $\mathcal{E}$ be a Lipschitz continuous loss function with Lipschitz constant $L_2$. Therefore, since $\mathcal{C}|_{B_i}$ is in $\text{Lip}(B_i, 0)$, for any $\Upsilon \in \text{Lip}(\widetilde{\mathcal{WL}_r}, L_1)$, the function $\mathcal{E}(\Upsilon|_{B_i}, \mathcal{C}|_{B_i})$ is in $\text{Lip}(B_i, L)$ with $L = L_1 L_2$.

## G.3 Uniform Monte Carlo approximation of Lipschitz continuous functions

The proof of Theorem 4.2 is based on the following Theorem G.3, which studies uniform Monte Carlo approximations of Lipschitz continuous functions over metric spaces with finite covering.

**Definition G.1.** *A metric space $\mathcal{M}$ is said to have* covering number $\kappa : (0, \infty) \to \mathbb{N}$, *if for every $\epsilon > 0$, the space $\mathcal{M}$ can be covered by $\kappa(\epsilon)$ ball of radius $\epsilon$.*

**Theorem G.2** (Hoeffding's Inequality). *Let $Y_1, \ldots, Y_N$ be independent random variables such that $a \leq Y_i \leq b$ almost surely. Then, for every $k > 0$,*

$$\mathbb{P}\Big(\Big|\frac{1}{N}\sum_{i=1}^{N}(Y_i - \mathbb{E}[Y_i])\Big| \geq k\Big) \leq 2\exp\Big(-\frac{2k^2 N}{(b-a)^2}\Big).$$

The following theorem is an extended version of [26, Lemma B.3], where the difference is that we use a general covering number $\kappa(\epsilon)$, where in [26, Lemma B.3] the covering number is exponential in $\epsilon$. For completion, we repeat here the proof, with the required modification.

**Theorem G.3** (Uniform Monte Carlo approximation for Lipschitz continuous functions). *Let $\mathcal{X}$ be a probability metric space[5], with probability measure $\mu$, and covering number $\kappa(\epsilon)$. Let $X_1, \ldots, X_N$ be drawn i.i.d. from $\mathcal{X}$. Then, for every $p > 0$, there exists an event $\mathcal{E}_{\text{Lip}}^p \subset \mathcal{X}^N$ (regarding the choice of $(X_1, \ldots, X_N)$), with probability*

$$\mu^N(\mathcal{E}_{\text{Lip}}^p) \geq 1 - p,$$

*such that for every $(X_1, \ldots, X_N) \in \mathcal{E}_{\text{Lip}}^p$, for every bounded Lipschitz continuous function $F : \mathcal{X} \to \mathbb{R}^d$ with Lipschitz constant $L_F$, we have*

$$\Big\|\int F(x)d\mu(x) - \frac{1}{N}\sum_{i=1}^{N}F(X_i)\Big\|_\infty \leq 2\xi^{-1}(N)L_f + \frac{1}{\sqrt{2}}\xi^{-1}(N)\|F\|_\infty(1 + \sqrt{\log(2/p)}), \quad (27)$$

*where $\xi(r) = \frac{\kappa(r)^2 \log(\kappa(r))}{r^2}$ and $\xi^{-1}$ is the inverse function of $\xi$.*

*Proof.* Let $r > 0$. There exists a covering of $\mathcal{X}$ by a set of balls $\{B_j\}_{j\in[J]}$ of radius $r$, where $J = \kappa(r)$. For $j = 2, \ldots, J$, we define $I_j := B_j \setminus \cup_{i<j}B_i$, and define $I_1 = B_1$. Hence, $\{I_j\}_{j\in[J]}$ is a family of measurable sets such that $I_j \cap I_i = \emptyset$ for all $i \neq j \in [J]$, $\bigcup_{j\in[J]} I_j = \chi$, and $\text{diam}(I_j) \leq 2r$ for all $j \in [J]$, where by convention $\text{diam}(\emptyset) = 0$. For each $j \in [J]$, let $z_j$ be the center of the ball $B_j$.

Next, we compute a concentration of error bound on the difference between the measure of $I_j$ and its Monte Carlo approximation, which is uniform in $j \in [J]$. Let $j \in [J]$ and $q \in (0, 1)$. By Hoeffding's inequality Theorem G.2, there is an event $\mathcal{E}_j^q$ with probability $\mu(\mathcal{E}_j^q) \geq 1 - q$, in which

$$\Big\|\frac{1}{N}\sum_{i=1}^{N}\mathbb{1}_{I_j}(X_i) - \mu(I_k)\Big\|_\infty \leq \frac{1}{\sqrt{2}}\frac{\sqrt{\log(2/q)}}{\sqrt{N}}. \quad (28)$$

Consider the event

$$\mathcal{E}_{\text{Lip}}^{Jq} = \bigcap_{j=1}^{J}\mathcal{E}_j^q,$$

with probability $\mu^N(\mathcal{E}_{\text{Lip}}^{Jq}) \geq 1 - Jq$. In this event, (28) holds for all $j \in \mathcal{J}$. We change the failure probability variable $p = Jq$, and denote $\mathcal{E}_{\text{Lip}}^p = \mathcal{E}_{\text{Lip}}^{Jq}$.

Next we bound uniformly the Monte Carlo approximation error of the integral of bounded Lipschitz continuous functions $F : \chi \to \mathbb{R}^F$. Let $F : \chi \to \mathbb{R}^F$ be a bounded Lipschitz continuous function with Lipschitz constant $L_F$. We define the step function

$$F^r(y) = \sum_{j\in[J]}F(z_j)\mathbb{1}_{I_j}(y).$$

---

[5]A metric space with a probability Borel measure, where we either take the completion of the measure space with respect to $\mu$ (adding all subsets of null-sets to the $\sigma$-algebra) or not.

Then,

$$\left\| \frac{1}{N} \sum_{i=1}^{N} F(X_i) - \int_{\mathcal{X}} F(y) d\mu(y) \right\|_\infty \leq \left\| \frac{1}{N} \sum_{i=1}^{N} F(X_i) - \frac{1}{N} \sum_{i=1}^{N} F^r(X_i) \right\|_\infty$$

$$+ \left\| \frac{1}{N} \sum_{i=1}^{N} F^r(X_i) - \int_{\mathcal{X}} F^r(y) d\mu(y) \right\|_\infty \qquad (29)$$

$$+ \left\| \int_{\mathcal{X}} F^r(y) d\mu(y) - \int_{\mathcal{X}} F(y) d\mu(y) \right\|_\infty$$

$$=: (1) + (2) + (3).$$

To bound (1), we define for each $X_i$ the unique index $j_i \in [J]$ s.t. $X_i \in I_{j_i}$. We calculate,

$$\left\| \frac{1}{N} \sum_{i=1}^{N} F(X_i) - \frac{1}{N} \sum_{i=1}^{N} F^r(X_i) \right\|_\infty \leq \frac{1}{N} \sum_{i=1}^{N} \left\| F(X_i) - \sum_{j \in \mathcal{J}} F(z_j) \mathbb{1}_{I_j}(X_i) \right\|_\infty$$

$$= \frac{1}{N} \sum_{i=1}^{N} \| F(X_i) - F(z_{j_i}) \|_\infty$$

$$\leq r L_F.$$

We proceed by bounding (2). In the event of $\mathcal{E}_{\text{Lip}}^p$, which holds with probability at least $1 - p$, equation (28) holds for all $j \in \mathcal{J}$. In this event, we get

$$\left\| \frac{1}{N} \sum_{i=1}^{N} F^r(X_i) - \int_{\mathcal{X}} F^r(y) d\mu(y) \right\|_\infty = \left\| \sum_{j \in [J]} \left( \frac{1}{N} \sum_{i=1}^{N} F(z_j) \mathbb{1}_{I_j}(X_i) - \int_{I_j} F(z_j) dy \right) \right\|_\infty$$

$$\leq \sum_{j \in [J]} \| F \|_\infty \left| \frac{1}{N} \sum_{i=1}^{N} \mathbb{1}_{I_j}(X_i) - \mu(I_j) \right|$$

$$\leq J \| F \|_\infty \frac{1}{\sqrt{2}} \frac{\sqrt{\log(2J/p)}}{\sqrt{N}}.$$

Recall that $J = \kappa(r)$. Then, with probability at least $1 - p$

$$\left\| \frac{1}{N} \sum_{i=1}^{N} F^r(X_i) - \int_{\mathcal{X}} F^r(y) d\mu(y) \right\|_\infty$$

$$\leq \kappa(r) \| F \|_\infty \frac{1}{\sqrt{2}} \frac{\sqrt{\log(\kappa(r)) + \log(2/p)}}{\sqrt{N}}.$$

To bound (3), we calculate

$$\left\| \int_{\mathcal{X}} F^r(y) d\mu(y) - \int_{\mathcal{X}} F(y) d\mu(y) \right\|_\infty = \left\| \int_{\mathcal{X}} \sum_{j \in [J]} F(z_j) \mathbb{1}_{I_j} d\mu(y) - \int_{\mathcal{X}} F(y) d\mu(y) \right\|_\infty$$

$$\leq \sum_{j \in [J]} \int_{I_j} \| F(z_j) - F(y) \|_\infty d\mu(y)$$

$$\leq r L_F.$$

By plugging the bounds of $(1), (2)$ and $(3)$ into $(29)$, we get

$$\left\| \frac{1}{N} \sum_{i=1}^{N} F(X_i) - \int_{\chi} F(y) d\mu(y) \right\|_{\infty} \leq 2rL_F + \kappa(r)\|F\|_{\infty} \frac{1}{\sqrt{2}} \frac{\sqrt{\log(\kappa(r)) + \log(2/p)}}{\sqrt{N}}$$

$$\leq 2rL_F + \frac{1}{\sqrt{2}}\kappa(r)\|F\|_{\infty} \frac{\sqrt{\log(\kappa(r))} + \sqrt{\log(2/p)}}{\sqrt{N}}$$

$$\leq 2rL_F + \frac{1}{\sqrt{2}}\kappa(r)\|F\|_{\infty} \frac{\sqrt{\log(\kappa(r))}}{\sqrt{N}}(1 + \sqrt{\log(2/p)}).$$

Lastly, choosing $r = \xi^{-1}(N)$ for $\xi(r) = \frac{\kappa(r)^2 \log(\kappa(r))}{r^2}$, gives $\frac{\kappa(r)\sqrt{\log(\kappa(r))}}{\sqrt{N}} = r$, so

$$\left\| \frac{1}{N} \sum_{i=1}^{N} F(X_i) - \int_{\chi} F(y) d\mu(y) \right\|_{\infty}$$

$$\leq 2\xi^{-1}(N)L_f + \frac{1}{\sqrt{2}}\xi^{-1}(N)\|F\|_{\infty}(1 + \sqrt{\log(2/p)}).$$

Since the event $\mathcal{E}_{\mathrm{Lip}}^p$ is independent of the choice of $F : \chi \to \mathbb{R}^F$, the proof is finished. ∎

## G.4 A generalization theorem for MPNNs

The following generalization theorem of MPNN is now a direct result of Theorem G.3.

Let $\mathrm{Lip}(\widetilde{\mathcal{WL}_r}, L_1)$ denote the space of Lipschitz continuous functions $\Theta : \mathcal{WL}_r \to \mathbb{R}^C$ with Lipschitz bound bounded by $L_1$ and $\|\Theta\|_{\infty} \leq L_1$. We note that the theorems of Appendix F.2 prove that MPNN with Lipschitz continuous message and update functions, and bounded formal biases, are in $\mathrm{Lip}(\widetilde{\mathcal{WL}_r}, L_1)$.

**Theorem G.4** (MPNN generalization theorem). *Consider the classification setting of Appendix G.2. Let $X_1, \ldots, X_N$ be independent random samples from the data distribution $(\widetilde{\mathcal{WL}_r}, \Sigma, \nu)$. Then, for every $p > 0$, there exists an event $\mathcal{E}^p \subset \widetilde{\mathcal{WL}_r}^N$ regarding the choice of $(X_1, \ldots, X_N)$, with probability*

$$\nu^N(\mathcal{E}^p) \geq 1 - Cp - 2\frac{C^2}{N},$$

*in which for every function $\Upsilon$ in the hypothesis class $\mathrm{Lip}(\widetilde{\mathcal{WL}_r}, L_1)$, with we have*

$$\left| \mathcal{R}(\Upsilon_{\mathbf{X}}) - \hat{\mathcal{R}}(\Upsilon_{\mathbf{X}}, \mathbf{X}) \right| \leq \xi^{-1}(N/2C)\left( 2L + \frac{1}{\sqrt{2}}\big(L + \mathcal{E}(0,0)\big)\big(1 + \sqrt{\log(2/p)}\big) \right), \qquad (30)$$

*where $\xi(r) = \frac{\kappa(r)^2 \log(\kappa(r))}{r^2}$, $\kappa$ is the covering number of $\widetilde{\mathcal{WL}_r}$ given in $(23)$, and $\xi^{-1}$ is the inverse function of $\xi$.*

*Proof.* For each $i \in [C]$, let $S_i$ be the number of samples of $\mathbf{X}$ that falls within $B_i$. The random variable $(S_1, \ldots, S_C)$ is multinomial, with expected value $(N/C, \ldots, N/C)$ and variance $(\frac{N(C-1)}{C^2}, \ldots, \frac{N(C-1)}{C^2}) \leq (\frac{N}{C}, \ldots, \frac{N}{C})$. We now use Chebyshev's inequality, which states that for any $a > 0$,

$$P\left( |S_i - N/C| > a\sqrt{\frac{N}{C}} \right) < a^{-2}.$$

We choose $a\sqrt{\frac{N}{C}} = \frac{N}{2C}$, so $a = \frac{N^{1/2}}{2C^{1/2}}$, and

$$P(|S_i - N/C| > \frac{N}{2C}) < \frac{2C}{N}.$$

Therefore,

$$P(S_i > \frac{N}{2C}) > 1 - \frac{2C}{N}.$$

We intersect these events of $i \in [C]$, and get an event $\mathcal{E}_{\text{mult}}$ of probability more than $1 - 2\frac{C^2}{N}$ in which $S_i > \frac{N}{2C}$ for every $i \in [C]$. In the following, given a set $B_i$ we consider a realization $M = S_i$, and then use the law of total probability.

From Theorem G.3 we get the following. For every $p > 0$, there exists an event $\mathcal{E}_i^p \subset B_i^M$ regarding the choice of $(X_1, \ldots, X_M) \subset B_i$, with probability

$$\nu^M(\mathcal{E}_{\text{Lip}}^p) \geq 1 - p,$$

such that for every function $\Upsilon'$ in the hypothesis class $\text{Lip}(\widetilde{\mathcal{WL}}_r, L_1)$, we have

$$\left| \int \mathcal{E}\big(\Upsilon'(x), \mathcal{C}(x)\big) d\nu(x) - \frac{1}{M} \sum_{i=1}^{M} \mathbb{E}\big(\Upsilon'(X_i), \mathcal{C}(X_i)\big) \right| \tag{31}$$

$$\leq 2\xi^{-1}(M)L + \frac{1}{\sqrt{2}}\xi^{-1}(M)\|\mathcal{E}\big(\Upsilon'(\cdot), \mathcal{C}(\cdot)\big)\|_\infty (1 + \sqrt{\log(2/p)}) \tag{32}$$

$$\leq 2\xi^{-1}(N/2C)L + \frac{1}{\sqrt{2}}\xi^{-1}(N/2C)(L + \mathcal{E}(0,0))(1 + \sqrt{\log(2/p)}), \tag{33}$$

where $\xi(r) = \frac{\kappa(r)^2 \log(\kappa(r))}{r^2}$, $\kappa$ is the covering number of $\widetilde{\mathcal{WL}}_r$ given in (23), and $\xi^{-1}$ is the inverse function of $\xi$. In the last inequality, we use the bound, for every $x \in \widetilde{\mathcal{WL}}_r$,

$$\left| \mathcal{E}\big(\Upsilon'(x), \mathcal{C}(x)\big) \right| \leq \left| \mathcal{E}\big(\Upsilon'(x), \mathcal{C}(x)\big) - \mathcal{E}(0,0) \right| + |\mathcal{E}(0,0)| \leq L_2 |L_1 - 0| + |\mathcal{E}(0,0)|.$$

Since (31) is true for any $\Upsilon' \in \text{Lip}(\widetilde{\mathcal{WL}}_r, L_1)$, it is also true for $\Upsilon_{\mathbf{X}}$ for any realization of $\mathbf{X}$, so we also have

$$\left| \mathcal{R}(\Upsilon_{\mathbf{X}}) - \hat{\mathcal{R}}(\Upsilon_{\mathbf{X}}, \mathbf{X}) \right| \leq 2\xi^{-1}(N/2C)L + \frac{1}{\sqrt{2}}\xi^{-1}(N/2C)(L + \mathcal{E}(0,0))(1 + \sqrt{\log(2/p)}).$$

Lastly, we denote

$$\mathcal{E}^p = \mathcal{E}_{\text{mult}} \cap \Big( \bigcup_{i=1}^{C} \mathcal{E}_i^p \Big).$$

■

### G.5 Experiments

The nontrivial part in our construction of the MPNN architecture is the choice of normalized sum aggregation as the aggregation method of the MPNNs. We hence show the accuracy and generalization gap of this aggregation scheme in practice in Table 1.

Most MPNNs typically use sum, mean or max aggregation. Intuitively, normalized sum aggregation is close to average aggregation, due its "normalized nature." For example, normalized sum and mean aggregations are well behaved for dense graphs with number of nodes going to infinity, while sum aggregation diverges for such graphs. Moreover, sum aggregation cannot be extended to graphons, while normalized sum and mean aggregations can. In Table 2, we first show that MPNNs with normalized sum aggregation perform well and generalize well. We then compare the normalized sum aggregation MPNNs (in rows 1 and 3 of Table 2) to baseline MPNNs with mean aggregation (rows 2 and 4 in Table 2), and show that normalized sum aggregation is not worse than mean aggregation.

The source code, courtesy of Ningyuan (Teresa) Huang, is available as part of `https://github.com/nhuang37/finegrain_expressivity_GNN` .

## H  Stability of MPNNs to graph subsampling

Lastly, we prove Theorem 4.3.

**Theorem H.1.** *Consider the setting of Theorem 4.2, and let $\Theta$ be a MPNN with Lipschitz constant $L$. Denote*

$$\Sigma = \big(W, \Theta(W, f)\big), \quad and \quad \Sigma(\Lambda) = \Big(\mathbb{G}(W, \Lambda), \Theta\big(\mathbb{G}(W, \Lambda), f(\Lambda)\big)\Big).$$

*Then*

$$\mathbb{E}\Big(\delta_\square\big(\Sigma, \Sigma(\Lambda)\big)\Big) < \frac{15}{\sqrt{\log(k)}}L.$$

Table 2: Standard MPNN architectures with normalized sum aggregation (nsa) and mean aggregation (ma), 3-layers with $512$-hidden-dimension, and global mean pooling, denoted by "MPNN-nsa" and "MPNN-ma." We use the MPNNs GIN [34] and GraphConv [28], and report the mean accuracy $\pm$ std over ten data splits. Nsa has good generalization and better performance than ma.

| Accuracy ↑ | MUTAG | IMDB-BINARY | IMDB-MULTI | NCI1 | PROTEINS | REDDIT-BINARY |
|---|---|---|---|---|---|---|
| GIN-nsa (train) | $83.94 \pm 3.25$ | $70.54 \pm 0.79$ | $47.01 \pm 0.8$ | $83.12 \pm 0.59$ | $74.06 \pm 0.44$ | $90.43 \pm 0.53$ |
| GIN-nsa (test) | $79.36 \pm 2.93$ | $69.83 \pm 0.93$ | $46.01 \pm 1.01$ | $78.55 \pm 0.3$ | $73.11 \pm 0.81$ | $89.38 \pm 0.57$ |
| GIN-ma (trained) | $74.63 \pm 2.93$ | $49.48 \pm 1.56$ | $33.70 \pm 1.35$ | $73.74 \pm 0.45$ | $71.53 \pm 0.93$ | $50.04 \pm 0.70$ |
| GIN-ma (untrained) | $72.46 \pm 2.56$ | $49.18 \pm 1.83$ | $33.03 \pm 1.12$ | $77.16 \pm 0.39$ | $70.33 \pm 0.95$ | $49.90 \pm 0.83$ |
| GraphConv-nsa (train) | $82.48 \pm 0.99$ | $59.34 \pm 2.34$ | $40.53 \pm 1.85$ | $63.14 \pm 0.55$ | $71.07 \pm 0.5$ | $82.4 \pm 0.19$ |
| GraphConv-nsa (test) | $82.04 \pm 1.05$ | $59.03 \pm 2.77$ | $40.25 \pm 1.59$ | $63.16 \pm 0.32$ | $70.92 \pm 0.7$ | $82.38 \pm 0.26$ |
| GraphConv-ma (trained) | $65.87 \pm 3.24$ | $49.32 \pm 1.35$ | $33.15 \pm 1.19$ | $54.39 \pm 1.25$ | $66.76 \pm 0.96$ | $49.68 \pm 0.82$ |
| GraphConv-ma (untrained) | $63.30 \pm 3.55$ | $48.80 \pm 1.91$ | $32.51 \pm 0.90$ | $55.84 \pm 0.53$ | $70.73 \pm 0.69$ | $49.39 \pm 0.48$ |

*Proof.* By Lipschitz continuity of $\Theta$,

$$\delta_\square\big(\Sigma, \Sigma(\Lambda)\big) \leq L\delta_\square\Big(\big(W, f\big), \big(\mathbb{G}(W, \Lambda), f(\Lambda)\big)\Big).$$

Hence,

$$\mathbb{E}\Big(\delta_\square\big(\Sigma, \Sigma(\Lambda)\big)\Big) \leq L\mathbb{E}\Big(\delta_\square\Big(\big(W, f\big), \big(\mathbb{G}(W, \Lambda), f(\Lambda)\big)\Big)\Big),$$

and the claim of the theorem follows from Theorem 3.7. ∎

As explained in Section 3.5, the above theorem of stability of MPNNs to graphon-signal sampling also applies to subsampling graph-signals.

# I   Notations

$[n] = \{1, \ldots, n\}$.

$\mathcal{L}^p(\mathcal{X})$ or $\mathcal{L}^p$: Lebesgue $p$ space over the measure space $\mathcal{X}$.

$\mu$: standard Lebesgue measure on $[0, 1]$.

$\mathcal{P}_k = \{P_1, \ldots, P_k\}$: partition (page 3)

$G = \{V, E\}$: simple graph with nodes $V$ and edges $E$.

$A = \{a_{i,j}\}_{i,j=1}^m$: graph adjacency matrix (page 4).

$e_G(U, S)$: the number of edges with one end point at $U$ and the other at $S$, where $U, S \subset V$ (page 3).

$e_{\mathcal{P}(U,S)}$: density of of edges between $U$ and $S$ (page 3).

$\mathrm{irreg}_G(\mathcal{P})$: irregularity (1).

$\mathcal{W}_0$: space of graphons (page 4).

$W$: graphon (page 4).

$W_G$: induced graphon from the graph $G$ (page 4).

$\|W\|_\square$: cut norm (page 4).

$d_\square(W, V)$: cut metric (page 4).

$\delta_\square(W, V)$: cut distance (page 4).

$S_{[0,1]}$: the space of measure preserving bijections $[0, 1] \to [0, 1]$ (page 4).

$S'_{[0,1]}$: the set of measurable measure preserving bijections between co-null sets of $[0, 1]$ (page 5).

$V^\phi(x, y) = V(\phi(x), \phi(y))$ (page 4).

$\widetilde{\mathcal{W}}_0$: space of graphons modulo zero cut distance (page 4).

$\mathcal{L}_r^\infty[0, 1]$: signal space (2).

$\|f\|_\square$: cut norm of a signal Definition 3.1.

$\mathcal{WL}_r$: graphon-signal space (page 5).

$\|(W, f)\|_\square$: graphon-signal cut distance (page 5).

$\delta_\square\big((W, f), (V, g)\big)$: graphon-signal cut distance (4).

$\widetilde{\mathcal{WL}_r}$: graphon-signal space modulo zero cut distance (page 5).

$(W, f)_{(G,\mathbf{f})} = (W_G, f_\mathbf{f})$: induced graphon-signal Definition 3.2.

$\mathcal{S}^d_{\mathcal{P}_k}$: the space of step functions of dimension $d$ over the partition $\mathcal{P}_k$ Definition 3.3.

$\mathcal{W}_0 \cap \mathcal{S}^2_{\mathcal{P}_k}$: the space of step graphons/ stochastic block models (page 6).

$\mathcal{L}^\infty_r[0, 1] \cap \mathcal{S}^1_{\mathcal{P}_k}$: the space of step signals (page 6).

$[\mathcal{WL}_r]_{\mathcal{P}_k}$: the space of graphon-signal stochastic block models with respect to the partition $\mathcal{P}_k$ (page 6).

$W(\Lambda)$: random weighted graph (page 7).

$f(\Lambda)$: random sampled signal (page 7).

$\mathbb{G}(W, \Lambda)$: random simple graph (page 7).

$\Phi(x, y)$: message function (page 8).

$\xi^k_\mathrm{r}, \xi^k_\mathrm{t} : \mathbb{R}^d \to \mathbb{R}^p$: receiver and transmitter message functions (page 8).

$\Phi_f : [0, 1]^2 \to \mathbb{R}^p$: message kernel (page 8).

$\mathrm{Agg}(W, Q)$: aggregation of message $Q$ with respect to graphon $W$ (page 8).

$f^{(t)}$: signal at layer $t$ (page 8).

$\mu^{(t+1)}$: update function (page 8).

$\Theta_t(W, f)$: the output of the MPNN applied on $(W, f) \in \mathcal{WL}_r$ at layer $t \in [T]$ (page 8).

$\mathrm{Lip}(\widetilde{\mathcal{WL}_r}, L_1)$: the space of Lipschitz continuous mappings $\Upsilon : \widetilde{\mathcal{WL}_r} \to \mathbb{R}^C$ with Lipschitz constant $L_1$ (page 9).

