# A graphon-signal analysis of graph neural networks
## Supplementary material

**Note to reviewers on modified constants:** when finalizing the writing of the proofs in the supplementary material, we realized that we can improve the constant in the regularity lemma from $9/4$ to $2$. Hence, there is a difference in this constant between the appendix and the main paper. We also corrected the constant in the sampling lemmas. We will make the minor modification of changing the constants in the main paper in the revised paper.

# A

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

Denote $a = 2KL^2B^2$. We have

$$\begin{aligned}
C_{t+1} &= a(C_t)^2 = a(aC_{t-1}^2)^2 = a^{1+2}C_{t-1}^4 = a^{1+2}(a(C_{t-2})^2)^4 \\
&= a^{1+2+4}(C_{t-2})^8 = a^{1+2+4+8}(C_{t-3})^{16} \le a^{2^t}C_0^{2^t}.
\end{aligned}$$

Now, for MPNNs with update function, we have

$$\begin{aligned}
C_{t+1} &\le LK(LC_t + B)^2 + B \\
&= KL^3C_t^2 + 2KL^2BC_t + KB^2L + B \\