# OpenReview forum: "A graphon-signal analysis of graph neural networks"
_NeurIPS.cc/2023/Conference — NeurIPS 2023 poster_

### Official Review · Reviewer_KJfc · 2023-07-03

**Soundness:** 3 good
**Presentation:** 2 fair
**Contribution:** 3 good
**Rating:** 7
**Confidence:** 3

**Summary:**

The authors introduce an extension of the graphon analysis to a graphon-signal analysis, in order to measure generalization bounds of and stability of MPNNs. Specifically, they propose a variant of cut distance as a new graph-signal similarity measure that transforms the space of graph-signals to a dense subset of a compact metric space.

Given the definition of the cut distance, the authors are able to show Lipschitz continuity of MPNNs, that can yield results on generalization and stability properties of MPNNs.

The authors do not show any empirical result on the connection between the graphon-signal analysis and MPNNs.

**Strengths:**

1. The methodology of analysing MPNNs through the extension of graphon theory is quite interesting and fairly novel.
2. The proposed approach can derive generalization and stability arguments for the MPNNs.
3. In contrast to related work, the authors provide a rigorous analysis that is justified for modeling the graph space, even when the Lipschitz continuity of graphons is required.

**Weaknesses:**

1. There is no empirical evidence that can highlight the connection of the graphon-signal analysis with the MPNNs.

2. There is no discussion about a related work [1], that also related stability arguments of MPNNs with respect to the graphon analysis.


[1] Luana Ruiz, Zhiyang Wang and Alejandro Ribeiro. GRAPHON AND GRAPH NEURAL NETWORK STABILITY. 2021.

**Questions:**

1. Can the authors provide any empirical result that can show stability or generalization capabilities of MPNNs with respect to the graph-signals?

2. Can the authors provide a further comparison of their analysis with the analysis in [1]?

Typos:
Inconsistencies in title case format
- '2.3 graphon analysis'
- '3.3 graphon-signal regularity lemma'

[1] Luana Ruiz, Zhiyang Wang and Alejandro Ribeiro. GRAPHON AND GRAPH NEURAL NETWORK STABILITY. 2021.

**Limitations:**

- The authors discuss the limitations of their theory with respect to the slow asymptotics and the subsampling theorems.
- The authors do not discuss any potential negative impact (although I do not see any negative impact coming out of this theory).

---

> ### Author Rebuttal · Authors · 2023-08-06
>
> We thank the reviewer for their constructive and supportive comments.
>
> >
>
> ***Experiments:*** We ran new experiments that we will include in the camera ready version if accepted. The only non trivial assumption in our model is that it uses normalized sum aggregation. To verify that such a model generalizes well in practice, we ran experiments on MPNN with normalized sum aggregation. The table shows that the model generalizes well. See the rebuttal PDF for the result. In our theory, there are no assumptions on the other aspects of the learning setting (number of parameters, data distribution, message function architecture). This is the point in the paper - to find a bound (conjectured to be essentially tight – see common response) on generalization under no assumptions. Hence, there are no other aspects to check, and standard results from other papers, that did assume something on the data and the model, also apply in our case.
>
> >
>
> ***Comparison to additional sampling results and [1]:*** We will extend the survey of other works that considered stability/transferability/sampling, including the suggested ref [1]. We note that, as explained in the text, such works give smaller lower bounds than our work by considering additional assumptions on the structure of the sampled graphon. Namely there are assumptions on the gap between eigenvalues of the graphon as a kernel operator, and assumptions that the graphon is Lipschitz continuous over a metric space. In our work the graphon is only assumed to be measurable, which leads to more general results with slower asymptotics. We note that mathematical modeling assumptions like Lipschitz continuity of the graphon need to be verified in practice by running experiments. On the other hand, we have no such assumptions, and any graph-signal is a special case of a measurable graphon-signal.
>
> In general, we may have not explained well enough the motivation of our construction. We will fix this in the camera ready version if accepted. All other GNN generalization (and sampling) papers have strong assumptions on the data distribution, (on the sampled graphons), and often on the model, which are often difficult to guarantee in real-life. In this work, we give bounds that do not depend on any assumption on the data distribution, (on the sampled graphon), and only mild assumptions on the model, assuming that the message functions are Lipschitz. Our work derives the bounds in case all assumptions of other papers fail. It is in a sense the ``bound on all other bounds.’’ We moreover conjecture that our bounds are optimal, as explained in the common response. **In this sense, our work complements rather than competes with past works: our work gives an “insurance card” to special constructions, by giving guarantees for convergence even in case the special assumptions are not met in practice.**
>
> >
>
> ***Typos:*** Thank you for spotting the typos. We will correct them.

---

> > ### Comment · Reviewer_KJfc · 2023-08-18
> >
> > Thank you very much for your response!
> > Your reactions to my comments are satisfying and I look forward to the new results in the camera-ready version if accepted. I retain my score to 7 (Accept)

---

> ### Comment · Area_Chair_51xX · 2023-08-18
>
> Hi,
>
> Could you please acknowledge (at least) the authors' rebuttal, and engage in the discussion if you have additional point that you wish to raise ?
>
> Thanks in advance for helping NeurIPS reviewing process,
>
> Best,
> AC

---

### Official Review · Reviewer_Leoh · 2023-07-08

**Soundness:** 3 good
**Presentation:** 3 good
**Contribution:** 3 good
**Rating:** 7
**Confidence:** 2

**Summary:**

This work focuses on analyzing message passing neural networks (MPNNs) by extending graphon analysis. It casts MPNNs as a function taking a graph-signal input (graph + signal on the graph) and proposing a similarity measure over graph signals, viewed as a dense subset of a compact metric space. This allows the authors to compare graph signals of different sizes and topological structures. More specifically, the authors formalize "regularity statements" for MPNNs by extending Szemerédi's weak regularity lemma to graphon signals. Furthermore, the authors demonstrate the power of their framework by using it to prove generalization bounds for MPNNs and establish stability of MPNNs under subsampling.

The work is a theoretical work that establishes a new framework for analyzing MPNNs and does not include any empirical results.

**Strengths:**

The work is original and introduces a new framework for analyzing MPNNs via graphon signals. The approach is original and differs from prior graphon-based MPNN works in that it does not assume a generative model of the data.

The paper is generally well-written and clear.



**Weaknesses:**

It is unclear what the practical implications of this result or analysis are. The paper could be strengthened by some experiments that show, for instance, that theoretically established generalization guarantees transfer over to empirical performance.

The exponential dependence in covering number, etc. render some of the theorems a bit impractical (though still of theoretical value).

**Questions:**

The exponential growth of the covering number seems to be a limitation for practicality of the Generalization Theorem in Thm 4.2. Are there important settings for which the covering number is much smaller? Does the author believe there is any hope for improving the dependence on covering number (e.g., improving ξ(ε)) in Them 4.2, or is the bound somewhat tight in the general setting?

**Limitations:**

Limitations of this work pertaining to slow asymptotics are referred to in the Discussion section.

---

> ### Author Rebuttal · Authors · 2023-08-06
>
>
> We thank the reviewer for their important and insightful comments.
>
> >
>
> ***Slow asymptotics and implications:*** The motivation may have been not clear enough in the first submission. We will extend the discussion in the introduction and generalization sections and stress the following point. All other GNN generalization papers have strong assumptions on the data distribution, and often on the model, which are often simplistic and not guaranteed in real-life. In this work we give generalization bounds that do not depend on any assumption on the data distribution, and only weak assumptions on the model, assuming that the message functions are Lipschitz. This work derives the generalization bounds in case all assumptions from other papers fail. It is in a sense the ``bound on all possible bounds.’’ We moreover conjecture that our bound is optimal, as explained next. In this sense, our work complements past works, by giving guarantees in case their assumptions are not met in practice.
>
> >
>
> ***Tightness of the bound:*** **We believe that our generalization bound is essentially tight.** Indeed, [1, Theorem 1.4] shows that the bound in the weak regularity lemma is tight up to the choice of the constant $c$ in $2^{c/\epsilon^2}$. Since in our graphon-signal theory, the bound is also of the form $2^{c/\epsilon^2}$, and a graphon can be seen as special graphon-signal, where the signal is constant 1, our bound in the regularity lemma is also tight up to the constant $c$. Indeed, the graphon which does not allow a regularity partition with less than $2^{c’/\epsilon^2}$ intervals in the graphon regularity lemma can also give a lower bound for the graphon-signal regularity lemma.
>
> Moreover, we also conjecture that the error bound in Theorem H.3 is optimal up to dividing $\epsilon$ by $\log(1/\epsilon)$. Let us explain the motivation. Kolmogorov and Tihomirov gave an exponential upper and lower bounds on the covering number of the L-Lipschitz continuous space over a metric space with finite covering [2, Theorem 17]. The lower bound only differs by a log factor from the upper bound. The construction in [2] is similar in nature to our construction of Theorem H.3. We hence believe that we can also prove in future work that Theorem H.3 is essentially optimal, and as a result so is our generalization bound. **We will write this conjecture in the camera ready version, add the required references, and leave the proof to future work.**
>
> >
>
> ***Experiments:*** We ran new experiments that we will include in the camera ready version if accepted. The only non trivial assumption in our model is that it uses normalized sum aggregation. To verify that such a model generalizes well in practice, we ran experiments on MPNNs with normalized sum aggregation. The table shows that the model generalizes well. See the rebuttal PDF for the result. In our theory, there are no assumptions on the other aspects of the learning setting (number of parameters, data distribution, message function architecture). This is the point in the paper - to find a bound (conjectured to be essentially tight) on generalization under no assumptions. Hence, there are no other aspects to directly check in experiments, and standard results from other papers, that did assume something about the data and the model, also apply in our case.
>
> >
>
> ***Settings in which the covering number is smaller:*** As explained above, other papers consider special assumptions on the data that lead to lower generalization bounds. The goal of our paper is to show what the bounds look like under no assumptions. We do plan to study in future work special data distributions defined via their covering number in the graphon-signal space, or their so-called metric entropy. We see the current paper as laying the groundwork on which future works can formulate criteria on data distributions that allow small generalization bounds.
>
> >
>
> [1] *D. Conlon, J. Fox. Bounds for graph regularity and removal lemmas. 2012.*
>
> [2] *A.N. Kolmogorov, V.M. Tihomirov, ε-Entropy and ε-capacity of sets in functional space. 1961.*

---

> > ### Comment · Reviewer_Leoh · 2023-08-11
> > **Acknowledgement of rebuttal**
> >
> > Thank you for the response to my questions. I appreciate the responses to my questions about the theory (covering number bounds, etc).
> >
> > Regarding the experiments, I am a bit unsure how to interpret Table 2. Could you clarify which results are baselines and which results use ideas from the submission?

---

> > > ### Author Response · Authors · 2023-08-12
> > > **Experiments**
> > >
> > > The main nontrivial part in our construction is the choice of normalized sum aggregation as the aggregation method of the MPNNs. We hence show the accuracy and generalization gap of this aggregation scheme in practice (in rows 1 and 3 of Table 2).
> > >
> > > Standard MPNNs typically use sum, mean or max aggregation. Intuitively, normalized sum aggregation is close to average aggregation, due its ``normalized nature.'' For example, normalize sum and mean aggregations are well behaved for dense graphs with number of nodes going to infinity, while sum aggregation diverges for such graphs. Moreover, sum aggregation cannot be extended to graphons, while normalized sum and mean aggregations can. We hence show two things in our experiments. First, we show that MPNNs with normalized sum aggregation perform well and generalize well. We then compare the normalized sum aggregation MPNNs (in rows 1 and 3 of Table 2) to baseline MPNNs with mean aggregation (rows 2 and 4 in Table 2), and show that normalized sum aggregation is not worse than mean aggregation (it is actually better).

---

> > > > ### Comment · Reviewer_Leoh · 2023-08-16
> > > > **Thank you for the response**
> > > >
> > > > Thank you for the clarification on the experiments. You have addressed my questions, and I would like to increase my score by 1.

---

### Official Review · Reviewer_9vxX · 2023-07-22

**Soundness:** 3 good
**Presentation:** 2 fair
**Contribution:** 3 good
**Rating:** 5
**Confidence:** 3

**Summary:**

This work proposes a new metric to measure the distance between graphon signals (graphon parameters and signals defined over graphon jointly). The metric generalizes the previous metric for graphon only. The work also uses the metric to characterize the approximation error of graphon signals by performing equipartition of the space, and by sampling. The work also uses the metric to characterize the generalization bound of GNNs and stability of GNNs when they are employed to encode graphon signals.

**Strengths:**

1. The work reads solid and investigates a very fundamental problem. The defined metric may be useful for the future study.
2. The study of the proposed metric is comprehensive, including approximation, generalization and stability.

**Weaknesses:**

The extension of the metric from graphon to graphon signal seems incremental to me. The following results on equipartition and sampling approximation, GNN generalization, and stability results seem to be also just extensions. A better explanation of what the technical difficulty behind such an extension should be discussed. Even if there is no substantial technical challenge, what new insights can be collected based on the new extension should be provided and compared with previous observations in a better way. Currently, it is hard for readers to collect insights into these results. What I learned is that the approximation error converges to 0 very slowly ($O(\sqrt{log k})$). But why do we have such a slow rate? Is it the best we can have? For the stability results, the issue is similar: does the result mean GNNs if used here are super unstable? In previous literature, the rate is about $O(1/k)$ or $O(1/\sqrt{k})$. Also, the implication behind the generalization results is almost not discussed at all.

Some minors:

1. the notations in this work are complicated and are used in a kind of arbitrary way. It is hard for readers to track the used notations. I suggest to provide a table list of the key notations and think about how to unify them.
2. Why the output of $\Phi$ is in $R^p$ in Sec 4.1 if the output of $\xi^k$ is $R^p$.


**Questions:**

Please check the above weaknesses.

I think the biggest issue behind this work is that discussion and comparison between the new results and previous ones are missing, which loses the chance to collect insights from the paper. Notations should be also improved.

**Limitations:**

Pure theoretical work, not see negative societal impact

---

> ### Author Rebuttal · Authors · 2023-08-06
>
> We thank the reviewer for their constructive and in-depth comments.
>
> >
>
> ***New insights from the theory and comparison to past results:*** Thank you for this comment. We see your point that the motivation is not explained clearly enough in the first submission. There is actually a clear reason to consider our proposed theory, that we will clarify in the camera ready paper if accepted. **All past works consider special assumptions on the data distribution, and often on the model. Our work provides upper bounds under no assumptions on the data distribution, and only mild Lipschitz continuity assumptions on the message passing functions. Hence, our theory shows how the bounds behave when all special assumptions from other papers are not met. We show that when all assumptions fail, MPNNs still have generalization and sampling guarantees, albeit much slower ones.** We see this as an important contribution, since the special assumptions from other papers, that are often simplistic, could fail to be satisfied in real data. We also note that our bounds do not directly depend on the number of parameters of the mode, in contrast to past works.
>
> This was written very shortly in the abstract and introduction in the submitted version. We will add paragraphs that explain this in detail in the introduction and in the generalization and sampling sections. **We will also add a section to the supplementary material that discusses what special assumptions each past work considers.** To summarize this comparison, see Table 1 in the rebuttal PDF.
>
> Regarding sampling, we already compared our results to past works in Section 4.4, in the paragraph of line 320. Past works assume that the sampled graphon is Lipschitz continuous over some metric space, which is an unnatural assumption in graphon theory – a theory of measurable functions, not continuous functions. Working with Lipschitz continuous graphon-signals with bounded Lipschitz constant does not allow approximating any graph-signal. Observe that when you talk about the Lipschitz continuity of the graphon, you actually treat [0,1] as the 1D metric line, and not the general atomless standard probability space. Hence, any theory that imposes Lipschitz continuity on the graphon actually models only 1D geometric graphs, which is very restrictive. On the other hand, when we only assume that the graphon is measurable, [0,1] is only treated as a standard (atomless) probability space, which is very general, and equivalent for example to [0,1]^d for any d, and to any Polish space. Basically, the reason graphon theory allows restricting the domain to [0,1] is because [0,1] as a measure space is very generic. **We will extend this comparison in the camera ready version if accepted.**
>
> >
>
> ***Optimality of results:*** Thank you for this important comment. We will clarify in the camera ready version the reason and meaning of the slow convergence rates. **We conjecture that our generalization bound is essentially tight.** Indeed, [1, Theorem 1.4] shows that the bound in the weak regularity lemma is tight up to the choice of the constant $c$ in $2^{c/\epsilon^2}$. Since in our graphon-signal theory the bound is also of the form $2^{c/\epsilon^2}$, and a graphon can be seen as special graphon-signal, where the signal is constant 1, our bound in the regularity lemma is also tight up to the constant $c$. Indeed, the graphon which does not allow a regularity partition with less than $2^{c’/\epsilon^2}$ intervals in the graphon regularity lemma can also give a lower bound for the graphon-signal regularity lemma.
>
> Moreover, we also conjecture that the error bound in Theorem H.3 is optimal up to dividing $\epsilon$ by $\log(1/\epsilon)$. Let us explain the motivation. Kolmogorov and Tihomirov gave an exponential upper and lower bounds on the covering number of the L-Lipschitz continuous space over a metric space with finite covering [2, Theorem 17]. The lower bound only differs by a log factor from the upper bound. The construction in [2] is similar in nature to our construction of Theorem H.3. We hence believe that we can also prove in future work that Theorem H.3 is essentially optimal, and as a result so is our generalization bound. **We will write this conjecture in the camera ready version, add the required references, and leave the proof to future work.**
>
> To conclude, our bounds apply to any data distribution and any model, and hence are very pessimistic, but still conjectured to be essentially optimal (when no assumptions on the data are made). Real-life data is distributed in special unknown ways that make MPNNs perform better than our pessimistic bound. While other works try to model special distributions and restrict to special MPNNs to obtain “tighter” bounds, our goal was to give the “bound of all bounds,” that is guaranteed always, and conjectured to be essentially tight when no assumption on the data and only mild assumption on the model are made. Our work hence complements past works, rather than compete with them.
>
> >
>
> ***Triviality of results:*** To use graphon analysis for GNNs you must extend it to graphon-signal analysis, as the input to GNNs are graph-signal. Even if this extension is trivial (which is a matter of opinion), the extension was not done before. Since the results are useful and lead to new insights, and were not done before, they are worthy of publication in our opinion, even if easy to prove. Moreover, we do not see the Lipschitz continuity of MPNNs as a trivial result.
>
> >
>
> ***Notations:*** We will provide a notation table in supplementary material in the camera ready paper, and simplify notations when possible.
>
> >
>
> ***Multiplication in Sec 4.1:*** The multiplication is elementwise, not a dot product. We will clarify this in the camera ready version.
>
> >
>
> [1] *D. Conlon, J. Fox. Bounds for graph regularity and removal lemmas. 2012.*
>
> [2] *A.N. Kolmogorov, V.M. Tihomirov, ε-Entropy and ε-capacity of sets in functional space. 1961.*

---

> > ### Comment · Reviewer_9vxX · 2023-08-11
> > **Thanks for the response**
> >
> > I greatly appreciate the detailed response. The table for the comparison across different works regarding the assumptions, data models, and results, is useful.
> >
> > Regarding the optimality of the results, I have one follow-up question: Since the derived results do not depend on assuming graphon as the data model, it is unclear to me why the result suffers from the same low bound for graphons as indicated by the reviewer. I still think the achieved much slower order is due to some underlying limit of the proof strategy, say via the weak regularity lemma.
> >
> > I am okay if the proof technique is not new, but the results are new and interesting, given the scope of NeurIPS.
> >
> > Overall, I would like to increase my evaluation to borderline accept.

---

> > > ### Author Response · Authors · 2023-08-12
> > > **Regarding the results being a proof artifact**
> > >
> > > Thank you for acknowledging our response and reconsidering your rating!
> > >
> > > **Regarding the result being a proof artifact:** We mostly agree with your comment. However, we can still conjecture essential tightness of our bounds for the problem of learning piecewise Lipschitz functions with Lipschitz functions over the graphon-signal space. In this setting, one could say that the slow convergence rates are artifacts of a setting with very weak assumptions, not of the specific proof technique. Let us explain this in more detail.
> > >
> > > The generalization problem is not a well defined mathematical question without defining the measurable space of graph-signals (so you can define data probability distributions). If you accept that the graphon-signal cut distance is a natural metric on graph-signals, then it defines a reasonable $\sigma$-algebra (of the Borel sets), and hence a measurable space. Note that we can restrict this $\sigma$-algebra to the subspace of graph-signals (which are just special cases of graphon-signals). Now, we conjecture that in this measurable space, the generalization bound is essentially optimal if you work with general Lipschitz continuous functions as the ``neural networks,'' and you don’t take any special assumptions on the probability measure.
> > >
> > > Under these assumptions, you are trying to bound the probability of a sample set of graph-signals being “good Monte-Carlo samples.” Namely, you hope that in high probability, the random graph-signals form a set on which the numerical integration (empirical risk) approximates well the integral of every (piecewise) Lipschitz continuous function (statistical risk). In our previous comment, we explained why we conjecture that our bound is essentially tight for this problem.
> > >
> > > The only missing link in this approach, is that not every Lipschitz continuous function from graphon-signals to $\mathbb{R}$ can be approximated by a MPNN. Due to the 1-WL limitation of MPNNs, there are graphon-signals that cannot be separated by MPNNs. It could be that the covering number of equivalence classes of graphon-signals that can be separated by MPNNs is lower. **We will hence clarify further, that we conjecture that the generalization bound is essentially tight for classification with general Lipschitz continuous functions over the space of graphon-signals**.

---

### Official Review · Reviewer_3Bfr · 2023-07-25

**Soundness:** 3 good
**Presentation:** 3 good
**Contribution:** 3 good
**Rating:** 6
**Confidence:** 2

**Summary:**

This paper extends the graphon theory to the graphon-signal theory. Especially, the authors introduce a distance metric "cut distance" on graphon-signal and analyze their regularity, and compactness. Furthermore, the authors use Lipschitez properties of MPNN to bound the model generalization.

**Strengths:**

I am not an expert in this field. But I carefully check the sketch of the points of the theoretical analysis is systemic and comprehensive.

**Weaknesses:**

I am not an expert in this field, but I think there are at least some points to improve the paper.

1. There are a lot of notations in this paper, it is better to summarize all the notations of this work and make a table in the appendix. It will help the review's comments

2. Some references are still missing for the training since adaptation. For example, the generalization gap in non-adversarial attacks should also be briefly discussed such as ( https://www.pnas.org/doi/10.1073/pnas.1907378117)

3. It would be better to have a figure to demonstrate the motivation of the distance metric.

**Questions:**

The questions are demonstrated in the weakness part.

**Limitations:**

The authors have adequately addressed the limitation of the public.

---

> ### Author Rebuttal · Authors · 2023-08-06
>
> We thank the reviewer for their constructive comments.
>
> >
>
> ***Notations:*** We will add a notation table in the supplementary material of the camera ready version if accepted.
>
> >
>
> ***References and discussion:*** We will add a discussion and references on benign overfitting and double descent to the introduction in the camera ready version, if accepted. We will make clear where our theory sits:  our results are in the classical setting of uniform convergence bounds, and we do not try to explain benign interpolation or double descent. We note that uniform bounds (e.g., using VC dimension or Rademacher complexity), while being classical for Euclidean data, are not well understood for graph data. Hence, one goal of this paper is to derive such bounds for GNNs.
>
> We will extend the discussion in the introduction and generalization sections regarding the motivation of our work, and stress the following point better. All other GNN generalization papers have strong assumptions on the data distribution, and often on the model, which are often simplistic and not guaranteed in real-life. In this work we give generalization bounds that do not depend on any assumption on the data distribution, and only weak assumptions on the model, assuming that the message functions are Lipschitz. This work derives the generalization bounds in case all assumptions from other papers fail. It is in a sense the ``bound on all possible bounds.’’ We moreover conjecture that our bound is optimal (when no assumptions on the data are made), as explained in the common response. In this sense, our work complements past works, by giving guarantees in case their assumptions are not met in practice.
>
> >
>
> ***Figures:*** Thank you for this important comment. We composed two new figures (see the rebuttal PDF), that will be added to the camera ready version if accepted. Figure 1 will be in the introduction. The figure illustrates two graph-signals that are close in cut-distance. Figure 2 will be put in the graphon-signal regularity lemma section (Section 3.3). The figure illustrates the measure-preserving bijection and step graphon-signal that approximates a graph-signal, guaranteed by the graphon-signal regularity lemma.

---

> ### Comment · Area_Chair_51xX · 2023-08-18
>
> Hi,
>
> Could you please acknowledge (at least) the authors' rebuttal, and engage in the discussion if you still have concerns?
>
> Thanks in advance for helping NeurIPS reviewing process,
>
> Best,
> AC

---

### Author Rebuttal · Authors · 2023-08-06

We thank the reviewers for their constructive and in-depth comments. Following the comments, we will edit the paper to clarify the motivation for our theory as explained below, and will add the experiments and figures described below and in the additional rebuttal PDF file.

>

***New insights from the theory:***  The motivation may have been not clear enough in the first submission. There is actually a clear reason to consider our proposed theory, that we will add to the camera ready paper if accepted. **All past works about generalization and sampling in GNNs consider special assumptions on the data distribution, and often on the model. Our work provides upper bounds under no assumptions on the data distribution, and only mild Lipschitz continuity assumptions on the message passing functions. Hence, our theory shows how the bounds behave when all special assumptions from other papers are not met. We show that when all assumptions fail, MPNNs still have generalization and sampling guarantees, albeit much slower ones.** We see this as an important contribution, since the special assumptions from other papers, that are often simplistic, may not be satisfied in real data.

We will add paragraphs that explain this in detail in the introduction and in the generalization and sampling sections. We will also add a section to the supplementary material that discusses what special assumptions each past work considers. To summarize this comparison, see Table 1 to the rebuttal PDF.

>

***Tightness of bounds:*** We will clarify in the camera ready version the reason and meaning of the slow convergence rates. **We conjecture that our generalization bound is essentially tight.** Indeed, [1, Theorem 1.4] shows that the bound in the standard weak regularity lemma is tight up to the choice of the constant $c$ in $2^{c/\epsilon^2}$. Since in our graphon-signal theory the bound is also of the form $2^{c/\epsilon^2}$, and a graphon can be seen as special graphon-signal, where the signal is constant 1, our bound in the regularity lemma is also tight up to the constant $c$. Indeed, the graphon which does not allow a regularity partition with less than $2^{c’/\epsilon^2}$ intervals in the standard regularity lemma can also give a lower bound for the graphon-signal regularity lemma.

Moreover, we also conjecture that the error bound in Theorem H.3 is optimal up to dividing $\epsilon$ by $\log(1/\epsilon)$. Let us explain the motivation. Kolmogorov and Tihomirov gave an exponential upper and lower bounds on the covering number of the L-Lipschitz continuous space over a metric space with finite covering [2, Theorem 17]. The lower bound only differs by a log factor from the upper bound. The construction in [2] is similar in nature to our construction of Theorem H.3. We hence believe that we can also prove in future work that Theorem H.3 is essentially optimal, and as a result so is our generalization bound. **We will write this conjecture in the camera ready version, add the required references and background motivation, and leave the proof to future work.**

>

***To conclude,*** our bounds apply to any data distribution and any model, and hence are very pessimistic, but still conjectured to be essentially optimal (when no assumptions on the data are made). Real-life data is distributed in special unknown ways that make MPNNs perform better than our pessimistic bound. While other works try to model special distributions and restrict to special MPNNs to obtain “tighter” bounds, our goal was to give the “bound of all bounds,” that is guaranteed always, and conjectured to be essentially tight when no assumption on the data and only mild assumption on the model are made. **In this sense, our work complements past works: it gives an “insurance card” to special constructions, by giving guarantees for convergence even in case the special assumptions are not met in practice.**

>

***Experiments:*** We ran new experiments that we will include in the camera ready version if accepted. The only non trivial assumption in our model is that it uses normalized sum aggregation. To verify that such a model generalizes well in practice, we ran experiments on MPNNs with normalized sum aggregation. The table shows that the model generalizes well. See the rebuttal PDF for the result. In our theory, there are no assumptions on the other aspects of the learning setting (number of parameters, data distribution, MPNN architecture). This is the point in the paper - to find a bound (conjectured to be essentially tight) on generalization under no assumptions. Hence, there are no other aspects to directly check by experiments, and standard results from other papers, that did assume something on the data and the model, also apply in our case.

>

***Notations:*** We will provide a notation table in the supplementary material in the camera ready paper, and simplify notations where possible.

>


***Figures:*** We will add new figures that illustrate the cut distance and the regularity lemma. See rebuttal PDF.

>

[1] *D. Conlon, J. Fox. Bounds for graph regularity and removal lemmas. 2012.*

[2] *A.N. Kolmogorov, V.M. Tihomirov, ε-Entropy and ε-capacity of sets in functional space. 1961.*

---

### Comment · Area_Chair_51xX · 2023-08-13

Thanks to all reviewers and authors for their work on this submission.

As the discussion period starts, I want to make sure that reviewers have read the author's response.

This can be done either by communicating with authors, or in private conversation within the reviewing team.

---

### Decision · Program_Chairs · 2023-09-21

**Decision:**

Accept (poster)

**Comment:**

This paper is concerned with a theoretical analysis of GNNs through the lens of graphons. It was overall well received by the reviewers, and the rebuttal allowed to fix minor concerns.
Please incorporate the remaining reviewers' feedback for the camera-ready version.